# How energy determines spatial localisation and copy number of molecules in neurons

Cornelius Bergmann ®[1], Kanaan Mousaei[1], Silvio O. Rizzoli ®[2] & Tatjana Tchumatchenko ®[1] ✉

In neurons, the quantities of mRNAs and proteins are traditionally assumed to be determined by functional, electrical or genetic factors. Yet, there may also be global, currently unknown computational rules that are valid across different molecular species inside a cell. Surprisingly, our results show that the energy for molecular turnover is a significant cellular expense, en par with spiking cost, and which requires energy-saving strategies. We show that the drive to save energy determines transcript quantities and their location while acting differently on each molecular species depending on the length, longevity and other features of the respective molecule. We combined our own data and experimental reports from five other large-scale mRNA and proteomics screens, comprising more than ten thousand molecular species to reveal the underlying computational principles of molecular localisation. We found that energy minimisation principles explain experimentally-reported exponential rank distributions of mRNA and protein copy numbers. Our results further reveal robust energy benefits when certain mRNA classes are moved into dendrites, for example mRNAs of proteins with long amino acid chains or mRNAs with large non-coding regions and long half-lives proving surprising insights at the level of molecular populations.

Advances in biochemistry and microscopy now allow us to precisely map the location of individual mRNA transcripts and their corresponding proteins in situ in cells and to quantify the copy numbers for thousands of mRNA and protein species[1–3]. These advances now permit to probe the existence of macroscopic organisational principles governing spatial gene expression patterns and operating across different molecular species. On the microscopic level, pathways that regulate transcriptional and translational speed, mRNA localisation, and protein guidance to its target are complex. Recent years have shown that transcriptional and translational steps can be dynamic, interdependent, and spatially distributed, calling for more macroscopic, unified theories of gene expression and transcript localisation that generalise across molecular species[4–7].

In neural dendrites, establishing and maintaining specific copy numbers for each protein species across proximal and distal synapses is essential for a neuron's computational and physiological functions[3,8]. Like all cells, neuronal functions are affected by energy constraints, which are particularly acute in the brain since it consumes ~20% of the body's energy[9] despite being relatively small in size. Neural cells are an excellent starting point to probe the existence of global computational principles governing spatial protein and mRNA localisation for three reasons. First, neural dendrites can span hundreds of micrometres, allowing for an understanding of spatial patterns of mRNA and protein copy numbers. Second, high-throughput proteomics and mRNA screens are readily available to study thousands of molecular species across cell compartments. Third, we now know many dynamic mRNA

[1]Institute of Experimental Epileptology and Cognition Research, Medical Faculty, University of Bonn, Venusberg-Campus 1, 53127 Bonn, Germany. [2]Department for Neuro- and Sensory Physiology, University Medical Center Göttingen Center for Biostructural Imaging of Neurodegeneration, BIN Humboldtallee 23, 37073 Göttingen, Germany. ✉e-mail: tatjana.tchumatchenko@uni-bonn.de

and protein properties, such as translation rates, half-lives, transport mechanisms, and spine-dendrite interactions, allowing us to combine transcript properties, their expression, location, and copy numbers across different species within a common framework. For example, many diverse mRNA localisation patterns have been reported in neural dendrites. While some mRNAs are abundant in dendrites, others are primarily confined to the soma[10,11]. Intriguingly, even protein species with similar synaptic functions and synaptic copy numbers could have very different mRNA localisation patterns. Therefore, linking function to the corresponding mRNA and protein distributions has historically been challenging. Recent studies emphasised the importance of understanding the rules of spatial gene expression by showing that temporal and spatial differences in mRNA translation contribute to synapse specificity[8] and are critical for embryonic cell development[12].

Here, we show that mRNA transcripts localise preferentially to the dendrites if they have large non-coding regions or are abundant - features making them energetically costly. On the other hand, proteins with longer half-lives, shorter amino acid chains, or higher diffusion constants localise their mRNA transcripts preferentially to the soma to minimise energy. Interestingly, we could predict and validate experimentally that the total synaptic copy number does not influence mRNA transcript localisation. Overall, we verified eight major predictions using simulations, our data[3] and data from five further different proteomics and genomics screens conducted in neurons[13–17] and an experimental measurement of spatiotemporal protein dynamics[18].

We propose a computational framework to explain the spatial localisation profiles and copy number distributions of mRNAs and proteins in neurons. We show that the high metabolic costs associated with molecule synthesis, transport, and degradation constrain spatial localisation patterns and total transcript numbers to solutions that minimise energy expenditures. We formalise the molecular energy concept using model calculations and test our predictions using six distinct high-throughput mRNA and proteomics screens containing tens of thousands of molecular species[3,13–17] and more than thirty different experimental reports describing diffusive properties of neural proteins[19].

Our results shed light on the macroscopic organisational principles of gene expression in cells that operate across different molecular species and transcend individual transcriptional, translational, or transport regulation mechanisms. The computational framework we propose can aid in discerning how changes in gene expression are linked to energy budgets and will help make quantitative and experimentally testable predictions for future experiments studying developmental gene expression, cell type differences, or activity-dependent proteome regulation.

## Results

To understand how spatial mRNA and protein distributions in cells are organised, we described them using a set of dynamical equations. In the second step, we calculated the energy expenditures necessary to maintain these distributions by considering transcription, translation, degradation, and transport costs incurred per second. Our model is designed to capture the spatial distributions of mRNAs and proteins inside the soma, the dendritic shaft, and the adjacent spines along a closed, linear segment of length $L$ that starts at the soma ($x = 0$). The variable $m$ denotes the concentration per $\mu m$ dendritic length of dendritic mRNAs, $p$, and $p_{\text{spine}}$ the concentration of proteins in the dendritic shaft and spines, respectively. The following dynamical equations govern the distributions of mRNA and proteins across space and time

$$\frac{\partial}{\partial t} m(x,t) = D_m \frac{\partial^2}{\partial x^2} m(x,t) - \lambda_m m(x,t), \qquad (1a)$$

$$\frac{\partial}{\partial t} p(x,t) = \tau m(x,t) + D_p \frac{\partial^2}{\partial x^2} p(x,t) - u_p p(x,t)\left(1 - \frac{p_{\text{spine}}(x,t)}{\rho \eta_p}\right) \qquad (1b)$$
$$+ \nu_p p_{\text{spine}}(x,t) - \lambda_p p(x,t),$$

$$\frac{\partial}{\partial t} p_{\text{spine}}(x,t) = u_p p(x,t)\left(1 - \frac{p_{\text{spine}}(x,t)}{\rho \eta_p}\right) - \nu_p p_{\text{spine}}(x,t) - \lambda_p p_{\text{spine}}(x,t). \qquad (1c)$$

Here, $\partial_t$ and $\partial_x$ denote the derivatives in time and space, respectively. Next, we obtained the dynamical equilibrium for mRNA $m(x)$ and protein $p(x)$ distributions considering a closed dendrite of length $L$ ($dm/dx(L) = 0$, $dp/dx(L) = 0$), a somatic influx of mRNAs and protein synthesis taking place at a site $x$ in proportion to $m(x)$, both at the soma and the dendrite. The total amount of mRNAs and proteins is set such that the minimal copy number of $\phi \eta_p$ at each synapse is met, here $\phi$ is the filling ratio and $\eta_p$ the maximal spine capacity for proteins, for more mathematical details see section 'Obtaining spatial distributions of mRNAs and proteins' in the Supplementary Notes. Now, let us introduce the model parameters and their biological interpretation. The diffusive motion of a molecule in the cytoplasm or cell membrane, as well as motor-mediated bidirectional movement on microtubules along the longitudinal axis of the dendrite, is represented by the mRNA and protein diffusion constants are $D_m$ and $D_p$, respectively. Using experimentally reported average run durations[20–27], run velocities[22,23,25,28–38], and the ratio of transported mRNAs[22,29,30,32–34,39–42] and proteins[43–47] (Supplementary Table 5 and Supplementary Notes, section 'Capturing active mRNA transport in our model'), we mapped this transport process[48] mathematically to a diffusion process with diffusion constants $D_m$, $D_p$ (see Supplementary Notes). The degradation rates of mRNAs and proteins are denoted by $\lambda_m$ and $\lambda_p$. $\tau$ represents the translation rate, the number of proteins translated from one mRNA per second. We used a translation rate of 0.01 proteins per mRNA per second for dendrites from experimental reports[49–60] (see Supplementary Notes, section 'Baseline mRNA and protein distributions'). Following ref. 61, we assumed that somatic and dendritic translation speeds are similar. $u_p$ ($\nu_p$) are the rates of proteins entering (leaving) spines[18,62–73], and $\eta_p$ is the maximal occupancy per spine. $\rho$ represents the spatial spine density along the dendrite and will be 1 per $\mu m$ throughout this work[74–84]. Functionally, keeping the protein copy number in synapses within a specific range for each molecular species is essential[85]. Therefore, we required in our model that all spines be filled to ≥95% of their capacity for a protein of interest (≥0.95$\eta_p$). $\eta_p$ was chosen following our data on the experimentally reported distribution[3], see also Supplementary Fig. 1. Having discussed the model parameters, let us comment on the shape on the spatial shape of the emerging mRNA and protein distributions. Depending on parameters, the mRNA and protein distributions can cover a broad functional range[86], for example they can increase, decrease, exhibit local extrema or be close to constant across the dendritic space and are thus able to capture the variety of experimentally reported molecular distributions, see Fig. 1C and Supplementary Fig. 14.

Next, we calculated the energy cost (in ATP per second) necessary to maintain each protein distribution. To this end, we divided the costs into three categories: transcription, translation, and active transport. Since the energy costs in each category depend strongly on the properties of the specific molecular species (e.g., its half-life), we calculated the energy costs for each protein species individually and included the energy associated with its mRNA substrate. Here, we briefly summarise the costs for all three dimensions. First, the transcription cost in our model is 2.17 ATP per nucleotide[87,88]. Second, the cost for protein biosynthesis and degradation together amounts to 5 ATP per amino acid[87–89]. We collectively refer to these costs as 'translation cost'. When a specific number of mRNAs and proteins must be synthesised per unit of time, we multiplied the translation and

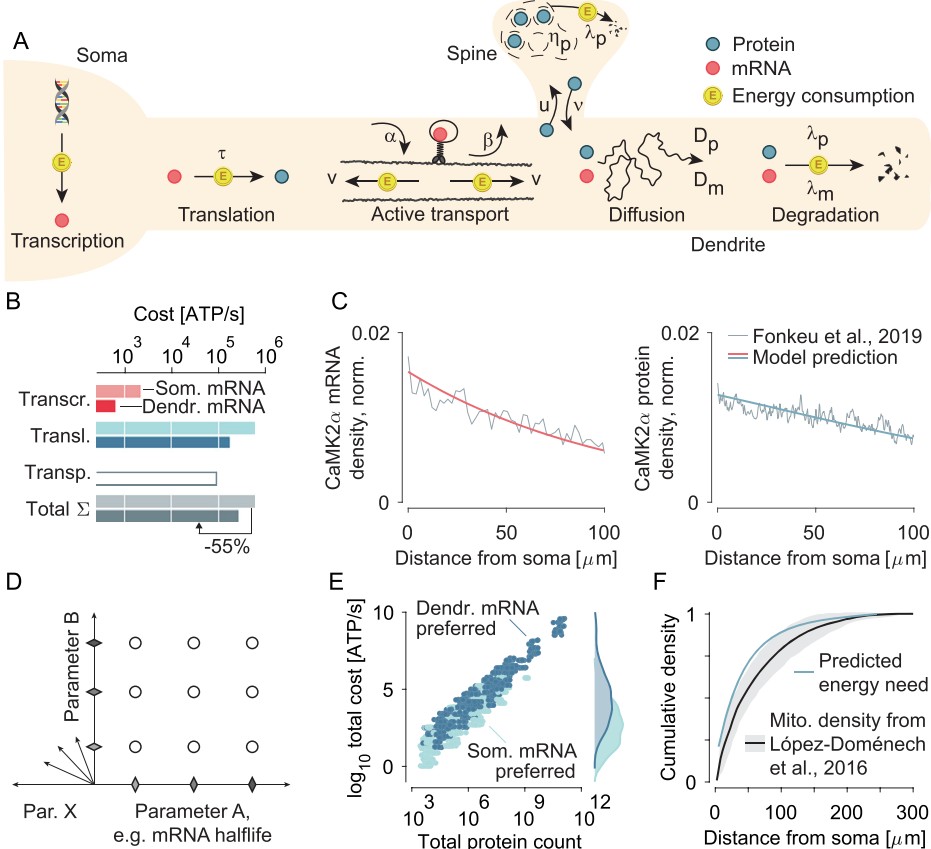

**Fig. 1 | Molecular processes and their individual energy expenditures.**
**A** Overview of the processes considered in our model. Yellow coins denote energy-consuming processes, processes lacking a coin are energetically free (e.g., diffusion). **B** Considering model parameters consistent with the prominent synaptic protein *CaMKIIα*, we find that dendritic protein synthesis is energetically more optimal (lower total cost) than the somatic mRNA confinement for this protein species. Our model computes the total energy cost and identifies minimal energy solutions. In the *CaMKIIα* inspired example, the mRNA transport costs (top white bar) are an investment worth making since the resulting lower total protein count drives down the dendritic translation cost (blue bar) and, as a result, minimises the total energy budget (dark vs. light grey bar). The dendrite length was 500 *μm*.
**C** Similarly, the predicted dendritic distributions of *CaMKIIα* mRNA and protein

align well with experimental data from[86]. Distributions were normalised to the integrated density within the first 100 *μm* of a 500 *μm* long dendrite. **D** Multiple parameter dimensions influence the energy budget of a protein. Therefore, we sampled three values for each parameter dimension and studied the energy budget associated with all parameter combinations. **E** Predicted total cost vs. total protein count per synthetic protein species sampled from our parameter space. All synthetic protein species utilise their preferred mRNA localisation, revealing a tendency of more costly and abundant proteins to localise mRNA in dendrites.
**F** Cumulative density of predicted total cost occurring in dendrites (blue) plotted vs. distance from soma. Costs were added across all our sampled synthetic protein species. Cumulative mitochondria distribution (black) from[128] is shown as mean with standard deviation.

transcription costs per molecule with the corresponding count, yielding the integrated energy costs per unit of time. Finally, transporting one cargo, i.e, one mRNA granule or protein vesicle, along the dendrite will cost 125 ATP per second, accounting for motor motion on the microtubules[36,90–97] (Supplementary Notes, section 'Calculating the energy cost'). We also analysed our transport model across different scenarios (Supplementary Figs. 2 and Supplementary Notes, section 'Capturing active mRNA transport in our model') and found that active mRNA transport is more energy effective compared to active protein transport (Supplementary Figs. 3, 4 and Supplementary Notes, section 'Capturing active protein transport in our model'), and can yield significant savings. For completeness, let us note that in the presence of an anterograde bias in active mRNA transport or protein transport results of Fig. 2 remain valid (Supplementary Figs. 3, 15). If all mRNAs of a protein species are confined to the soma, only transcription and translation contribute to the energy expenditures in our model, whereby translation will be the dominant contribution (Supplementary Fig. 5A-B left), in line with[88]. On the other hand, trafficking mRNAs from the soma into the dendrites reduces the spatial distances proteins have to cross, thereby lowering the cost of transcription and translation (the median reduction across our sampled parameters is

~ 15%) because fewer mRNA and protein copies need to be produced, but longitudinal mRNA transport along dendritic microtubules will contribute to the energy budget (Supplementary Fig. 5A, B right). Let us note that our energy calculation considers only the costs directly associated with protein and mRNA biosynthesis, degradation, or transport and excludes the energy needed to manufacture the supporting infrastructure, e.g., assembly of mitochondria providing ATP, synthesis of tRNAs delivering the amino acids, ribosome synthesis and transport, energy spent on regulatory pathways activating translation or co-translational modifications. We excluded these costs because they are already included in the cost estimates we use (e.g. co-transport of ribosomes and mRNA in a single granule does not incur additional cost[98]) or can be excluded because they are not unique to the biosynthesis or transportation of a specific transcript but instead can be considered general 'infrastructure' costs of a cell.

Let's now put the abstract calculations into action and study the energy budget of a specific molecule: a prominent neural protein implicated in synaptic plasticity, *CaMKIIα*. *CaMKIIα* proteins contain 478 amino acids, its transcripts have ~4900 nucleotides[99], and an average dendritic spine contains around 32336 *CaMKIIα* proteins[3].

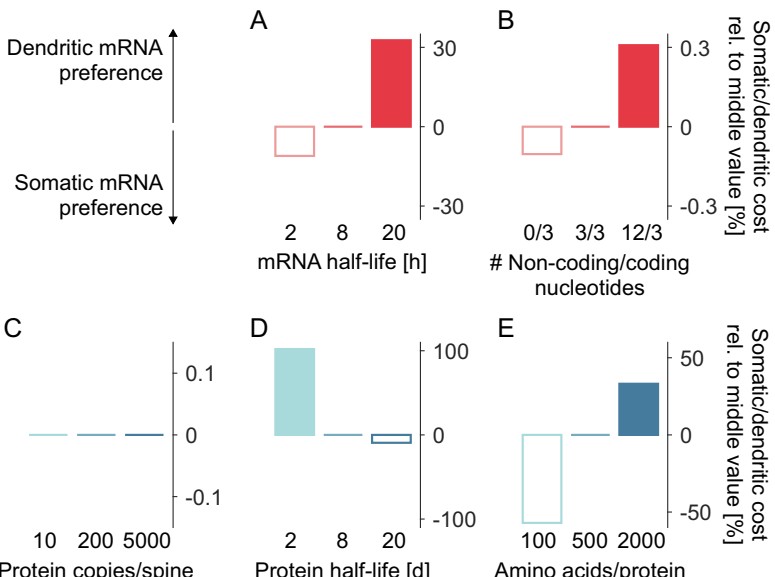

**Fig. 2 | Minimal energy principles predict differences in protein synthesis location along specific parameter dimensions. A–E** Effect of mRNA (red) and protein (blue) parameters on the predicted preference of synthetic protein species for dendritic mRNA with mRNA transport vs. somatic mRNA on a 1000 $\mu m$ long dendrite. In each panel, one parameter is varied along the *x*-axis: **A** mRNA half-life, **B** length of non-coding regions relative to coding sequence length, **C** maximal number of proteins per spine, **D** protein half-life, **E** protein size in amino acids. For each synthetic protein species, the total costs when using somatic mRNA were divided by those with dendritic mRNA and normalised with the corresponding ratio in the middle category. Bars show the population median.

*CaMKIIα* proteins have been shown to perform a rapid, non-directed two-dimensional diffusive motion in the dendritic cytoplasm without signatures of active transport[100] Its protein half-life is ~8 days[13,101], and its mRNA half-life is ~20 h[14,99]. Considering these parameters, our model suggests that local translation for this protein species is energetically ~55% more efficient; see Fig. 1B for an energy budget. This is consistent with the observation that *CaMKIIα* mRNAs associate with motor proteins in dendrites[39,40,53,102,103]. To examine the biological plausibility of our simulations further, we asked if our model could reproduce experimentally observed distributions of *CaMKIIα* mRNA and protein in dendrites. Using only experimentally reported *CaMKIIα* parameters (no fits), we simulated the energy optimal mRNA and protein levels and found them to be in line with the experimentally measured cell- and dendrite averaged profile in cultured rat hippocampal pyramidal cells reported in[86] (Fig. 1C). Considering this example, we wondered what the critical parameters are that make a protein prefer dendritic or somatic translation.

### Dendritic mRNA localisation minimises energy expenditure for specific classes of proteins

Are there any statistical rules that are valid across the diverse neural protein pool? This question can only be answered by dissecting the role of each parameter in the energy budget. Our model can explore the energy landscape and propose an energy-optimal strategy for each protein species of interest, analogously to our *CaMKIIα* example. We can study groups of proteins that share one common feature (e.g., a long mRNA half-life) but are otherwise diverse. In this scenario, we could answer whether these proteins as a group prefer a specific energy optimal configuration. This allows making statistical statements transcending individual mRNA and protein species and explaining how a given parameter value tilts the scale towards a particular energy optimal configuration. To this end, we generated synthetic protein species by drawing random, uncorrelated samples from the biologically plausible range for each model parameter individually ($N = 3^7 = 2187$; Supplementary Fig. 1 for experimental sources[13,14,18–20,22,28–30,40,42,43,50,52,60,88,99,101,104–127], Supplementary Table 6

for parameter overview and section 'Sampling of synthetic protein species' in the Supplementary Notes for more details).

We investigated the conditions making dendritic or somatic mRNAs energy-optimal by counting the fraction of our synthetic protein species preferring each strategy. We found that on a 250 $\mu m$ (500 $\mu m$, 750 $\mu m$, 1000 $\mu m$) long dendrite, dendritic mRNA is the energetically optimal solution for 34% (41%, 42%, 42%) of all species. We found that the total number of proteins is a strong predictor for the total energetic cost, which puts pressure on the molecular species with high protein numbers to opt for the dendritic mRNA localisation (Fig. 1E, see also Fig. 3). Additionally, the broad total cost distribution per species indicates that the overall energy budget for protein turnover is large and is dominated by few, very abundant protein species. Overall, our model framework shows that the predicted ATP amount per species is relatively high. Considering that many thousands of proteins operate in the dendrites, it is plausible that this energy need could imprint its spatial profile on the mitochondrial distribution. Therefore, we asked whether the distribution of dendritic mitochondria bears signatures of our predicted spatial energy consumption profile. Remarkably, we found that the local energy need predicted by our model is within the 1.4$\sigma$ interval of the experimentally measured mitochondria distribution in dendrites of hippocampal pyramidal cells of mice[128], see Fig. 1F. Let us also note that the range of predicted mRNA distributions aligns with experimental observations (Supplementary Fig. 6).

To investigate how the energetic preference for dendritic vs. somatic mRNA localisation varies across our parameter space, we defined three groups along each of the seven dimensions: small, medium, and large. We assigned each of the 2187 synthetic proteins to one of the three groups based on the parameters of interest, for which we sampled three values each. For each group member, we divided the costs arising in the somatic mRNA scenario by those of the dendritic mRNA scenario and normalised this ratio with the 'medium' middle group. The resulting quantity describes how the preference for dendritic vs. somatic mRNA changes as a function of the parameter of interest (Fig. 2), specifically mRNA half-life (Fig. 2A), the ratio of non-

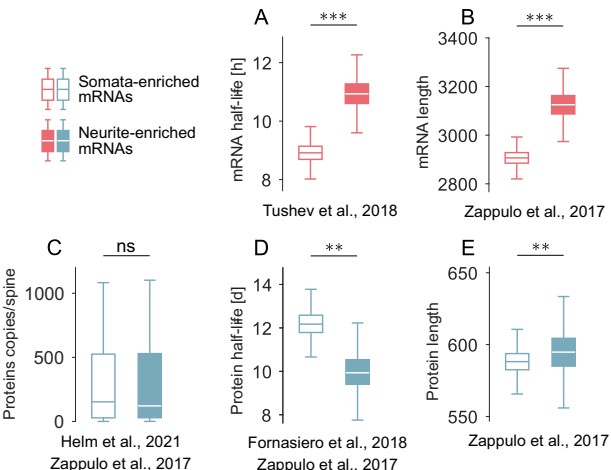

**Fig. 3 | Experimental validation of model predictions shown in Fig. 2.** mRNA (red) and protein (blue) parameter values and corresponding mRNA enrichment in neurites vs. somata were extracted from published databases. Based on mRNA enrichment, we labelled database entries as neurite-enriched (filled boxes) or somata-enriched (empty boxes). **A** 3'-UTRs isoform half-lives and corresponding enrichment scores from[14]. ($N_{somata}$ = 341, $N_{neurite}$ = 201, $p$ = 6.06 × 10$^{-4}$, bootstrapped to 10 k over mean); **B** Transcript length in nucleotides and mRNA enrichment of[16] ($N_{somata}$ = 5623, $N_{neurite}$ = 1406, $p$ = 2.14 × 10$^{-9}$, bootstrapped to 10k over mean); **C** Protein counts per spine from our data[3] and mRNA enrichment from[16] ($N_{somata}$ = 58, $N_{neurite}$ = 25, $p$ = 0.92); **D** Protein half-lives from[13] matched with mRNA enrichment scores of[16] ($N_{somata}$ = 418, $N_{neurite}$ = 151, $p$ = 1.9 × 10$^{-3}$, bootstrapped to 10k over mean); **E** Protein length in amino acids and mRNA enrichment scores from[16] ($N_{somata}$ = 5623, $N_{neurite}$ = 1406, $p$ = 5 × 10$^{-3}$, bootstrapped to 10k over mean). Boxplots indicate median, quartiles and 1.5x interquartile ranges. *$p$ < 0.05, **$p$ < 0.01, ***$p$ < 0.001, two-sided Wilcoxon ranksum test, pairwise comparison within each panel. For further details on the data analyses, see Methods.

coding vs. coding nucleotides per mRNA (Fig. 2B), protein copies per spine (Fig. 2C), protein half-life (Fig. 2D), and the amino acid number (Fig. 2E).

First, the total protein copy number in a spine strongly drives the total cost (Supplementary Fig. 7D) but was not crucial for the somatic vs. dendritic localisation decision (Fig. 2C). One could hypothesise that high spine copy numbers correlate with local translation, but the model does not support this. The reason is that the energy budget for each protein species depends critically on the relative energy expenditures between translation, transcription, and active transport and evaluates whether shifting between categories makes sense energetically. However, the value of the total energy budget is not essential for determining the optimal solution. Therefore, two proteins with the same model parameters differing only in their spine count will have the same energy optimal localisation strategy. Addressing our model's protein half-life dimension, we found that long half-lives correlate with a preference for somatic mRNA localisation (Fig. 2D). On the other end of the spectrum are proteins exploring a limited spatial range within their short lifetime; they prefer dendritic mRNA localisation. Considering the influence of the protein half-life on the total energy budget, we find that longer protein half-lives are correlated with lower energy expenditures (Supplementary Fig. 7E). Interestingly, the energetically optimal solution is strongly shaped by the length of the amino acid chain, Fig. 2E. Our model predicts that the longer the protein, the higher its translational cost, and as a result, the higher the energetic pressure for mRNA localisation. The underlying computational reason is the need for long proteins to keep their total copy number down by minimising excess translation cost (see also Supplementary Fig. 7F). Local synthesis of proteins reduces the time the proteins spend en route rather than at the destination. Notably, when considering the mRNA half-life in our model, we found the opposite behaviour (Fig. 2A)

relative to its protein counterpart in Fig. 2D. Our energy considerations predict that longer-lived mRNAs prefer dendritic localisation. Investing energy in active transport to move them out into the dendrite pays off more than doing the same with shorter-lived mRNAs. This strategy also drives down the total cost (Supplementary Fig. 7A). When studying the influence of the mRNA length on the energy budget and the optimal localisation strategy, we found that longer mRNAs, those with relatively long non-coding regions, favour local synthesis (Fig. 2B). We could explain this effect in our model via increased transcription cost for these molecules and, thus, higher energetic pressure to maximise their protein synthesis potential even though the transcription cost was small compared to the total cost (Supplementary Fig. 5). As a last step, we confirmed that our results in Fig. 2 hold across different dendritic lengths. When the dendrite grows, it will acquire more synapses and exponentially increase its total protein need with dendrite length (Supplementary Figs. 9, 10). However, we found the model predictions discussed above to be largely independent of dendritic length (Supplementary Fig. 3). For completeness, let us mention that results in Fig. 2 hold across different spine filling rates ($\phi$ = 0.7 − 0.95) and both in the presence as well as the absence of an anterograde bias in the active mRNA transport. For completeness, let us also note that in Fig. 2 we considered 1 mRNA per granule while we explored the effect of 2 and 10 mRNAs per granule in Supplementary Fig. 18. Since the number of mRNAs per granule is variable across the literature[22,27,30,32,33,103,123,124,129–133] (see section 'mRNA granule content' in the Supplementary Notes), we confirmed that our results hold across a broad experimentally plausible range.

## Model predictions match experimental evidence from high-throughput transcriptome and proteome screens

We were curious whether our predictions hold across neuronal protein species. Tens of thousands of distinct protein species have been experimentally observed in neurons, and many high-throughput mRNA screens have been conducted. This offers fertile ground to test our predictions (Supplementary Table 2). Here, we will focus specifically on experimental reports addressing the molecular composition of the soma and neural dendrites ('somata-enriched' or 'neurite-enriched' mRNA species, see Methods). As the first step, we focused on the model predictions made for the properties of mRNAs. First, we analysed mRNA transcript half-life and enrichment data from[14]. We reaffirmed their finding that neuropil-enriched mRNAs are significantly longer-lived (Fig. 3A), thereby confirming our model prediction for mRNA half-life (Fig. 2A). In addition, ref. 14 directly compared the half-lives of different isoforms of one mRNA and found that longer-lived isoforms are significantly more enriched in the neuropil, similar to results by ref. 134. Next, our model predicted longer mRNA transcripts within the neurite-enriched group (Fig. 2B), which we verified with experimentally reported transcript lengths and enrichment labels from[16] (Fig. 3B). Similarly, previous studies[14,80] highlighted that transcripts with shorter 3'-UTRs are constrained to the soma, while transcripts of the same mRNA with longer 3'-UTRs preferentially localise in dendrites. Another study also found longer 3'-UTRs among neurite-enriched transcripts[135].

Furthermore, we shift our focus to model predictions regarding protein properties. We can confirm our prediction that the protein copy number per spine is independent of the preference of mRNAs for somata or neurites (Fig. 2C) by combining our data on spine copy numbers[3] with mRNA enrichment scores from[16] (Fig. 3C). To test our next model prediction (Fig. 2D) that proteins with a longer half-life preferentially localise their mRNAs in dendrites, we combined protein half-lives from[13] and mRNA enrichment categories from[16]. Let us briefly note that other mRNA screens[14,61], could also be suitable candidates; however, they contained less overlapping mRNA and protein species with[13]. Therefore, we favoured pairs of databases with the most extensive overlap throughout the analyses shown in Fig. 3. Sorting

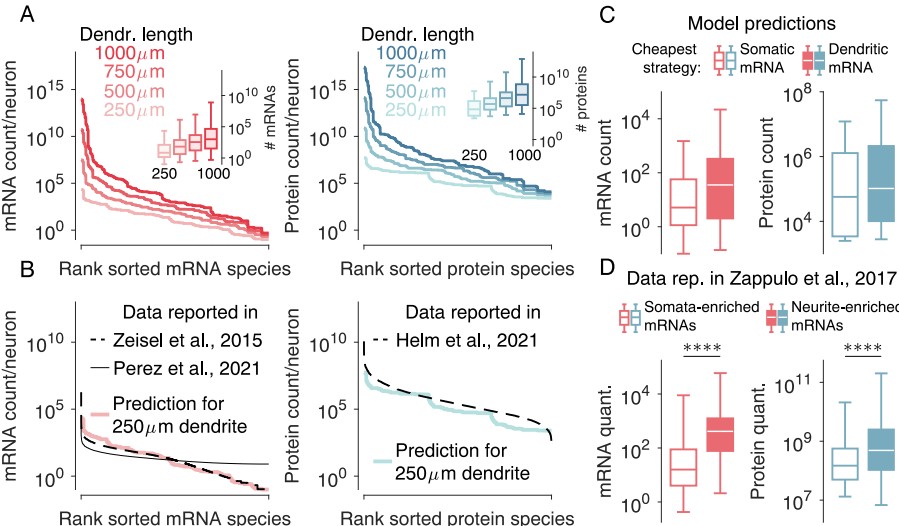

**Fig. 4 | Energy-based model predictions explain experimentally reported mRNA and protein copy numbers in neurons. A** Predicted total mRNA (left) and protein (right) copy numbers across different dendrite lengths, colour-coded for length. mRNA and protein species are rank-sorted for abundance in each panel independently. Insets: Corresponding copy number statistics vs. dendritic length. **B** Validation of predicted abundances on a 250 $\mu m$ long dendrite with experimental data. Simulated model results are shown alongside mRNA abundance data (left) published by ref. 15 ($N = 14248$), ref. 17 ($N = 19184$) and protein abundance data (right) reported by us[3] ($N = 6153$). Experimental protein and mRNA data were sorted independently based on abundance. **C** Predicted mRNA (left) and protein (right) protein species in the somata vs. neurite categories based on their mRNA localisation[16] and calculating their half-life from[13], we found that species with neurite-enriched mRNAs indeed exhibit significantly lower protein half-lives (Fig. 3D). This finding agrees with our model prediction (Fig. 2D). For our last model prediction that proteins with a longer amino acid chain tend to prefer local rather than somatic translation, we used mRNA enrichment and protein size data from[16]. We found that proteins of neurite-enriched mRNAs indeed possess more amino acids than those associated with somata-enriched mRNAs (Fig. 3E), in line with our energy minimisation prediction (Fig. 2E).

abundances for synthetic protein species grouped along the energetically preferred localisation pathway, either dendritic mRNA with mRNA transport (filled box) or somatic mRNA (empty box). Simulations were performed on a 250 $\mu m$ long dendrite. **D** Validation of C with experimental data. mRNA (left) and protein (right) abundance data were extracted from ref. 16 and labelled as neurite-enriched (filled boxes) or somata-enriched (empty boxes) based on their mRNA enrichment (see Methods; mRNA: $N_{somata} = 5623$, $N_{neurite} = 1406$, $p = 0$; protein: $N_{somata} = 852$, $N_{neurite} = 622$, $p = 5.12 \times 10^{-21}$). Synthetic protein species use their respective energetically cheapest localisation pathway in A, B. Boxplots indicate median, quartiles and 1.5x interquartile ranges. ****$p < 0.0001$, two-sided Wilcoxon ranksum test.

For completeness, let us note that, the predictions in Fig. 2 are valid also in the presence of an anterograde bias that results in a net active mRNA velocity ($v_m = 0.001 \mu m/s$), see Supplementary Fig. 15. Finally, let us note that our model makes predictions about energy-optimal strategies to achieve stable protein numbers across synapses (proteostasis), which may not be the only functional goal of a neuron. Neurons must modulate temporally and spatially protein copy numbers to respond to growth, activity or plasticity cues. Therefore, it is surprising to see model predictions derived solely based on proteostasis in the dendrites being a strong predictor for molecular and mitochondrial localisations.

## Minimal energy principles are in line with experimentally reported mRNA and protein abundances

Next, we used minimal energy principles to predict a single neuron's total protein and mRNA copy numbers per transcript. To this end, we used again the 2187 synthetic model proteins we sampled from biologically reported parameter values (Supplementary Fig. 1). We predicted the total mRNA and protein numbers for each synthetic protein, assuming each protein implements an energetically optimal mRNA localisation strategy. In other words, we calculated the corresponding metabolic costs associated with either somatic or dendritic mRNA and chose the least costly option for every parameter combination in our 7-dimensional parameter space. Then, we

integrated the mRNA and protein counts in the soma, dendrite, and spines and obtained the total overall counts after summing up. Finally, we ranked mRNAs and proteins based on their abundance and repeated the procedure for dendrite lengths from 250 to 1000 $\mu m$ (Fig. 4A). Longer dendrites need more proteins, wherefore both mRNA and protein counts increase strongly with dendrite length (see also Supplementary Fig. 10).

When comparing our simulation results with experimental data, we first found that the mRNA count distribution per neuron reported by Perez et al.[15] is on the same order of magnitude as those predicted by our model (Fig. 4B, left). To ensure that our choice of a 250 $\mu m$ model dendrite reflects the actual size of the cell type for which[15] reported mRNA copy numbers, we consulted morphologies of neurons published on neuromorpho.org. We found good agreement (Supplementary Fig. 11). Slight deviations between the predicted and measured mRNA copy numbers can be explained by our parameter sampling strategy, which may have over-represent high and low values. Similarly, our predicted protein abundances in a 250$\mu m$ long dendrite matched our experimental reports[3] (Fig. 4B, right). Having found a good correspondence between our model and the experimentally reported absolute mRNA and protein copy numbers per cell, we investigated whether mRNA and protein counts vary among mRNA and protein species that prefer somatic vs. dendritic mRNA localisation. For this purpose, we split the space of synthetic proteins into two groups, those energetically favouring somatic vs dendritic mRNA localisation. In our simulations, the latter group showed higher mRNA and protein counts than the first, independent of dendrite length (Fig. 4C, Supplementary Fig. 10). We confirmed this model prediction using mRNA localisation patterns and mRNA and protein quantifications from[16] (Fig. 4D). The predicted difference in mRNA and protein abundance between species that energetically prefer somatic vs. dendritic mRNAs can also be intuitively understood as follows: Molecular species generating high energy costs are those with short half-lives, small diffusion constants, and big protein and mRNA molecules.

These tend to profit stronger from dendritic mRNA localisation (Fig. 2). At the same time, these 'cost intensive' species tend to have overall higher mRNA and protein numbers. Employing active mRNA transport can save some of the cost by reducing the mRNA and protein levels. However, even after all energy optimisation strategies are factored in, the absolute numbers are still higher for these molecular species than those preferring somatic mRNA localisation. Let us note that beyond the total mRNA and protein copy numbers, our model predicts that protein numbers correlate with mRNA numbers (Supplementary Fig. 12), and this correlation increases as a function of dendritic length considered in the model. Experimentally-measured baseline mRNA and protein levels are known to correlate (reviewed in[136]). Because mRNA and protein levels are subject to variability, e.g., cell state or activity fluctuations, the experimentally observed correlations can be lower than those predicted by our model. For completeness, let us mention that we confirmed all five experimental confirmations shown in Fig. 3 using at least three different database pairings per result considering only data from the same species in each database pair (Supplementary Fig. 19).

## Integration of proteins into spines across time and space

Up to this point, we have focused on the equilibrium distributions of mRNA and proteins. However, many of the processes covered by our model, such as trafficking and protein exchange between the dendrite and spines, are inherently dynamic. These dynamics are crucial for the energy budget because the time-to-destination for proteins directly affects the number of proteins that must be synthesised (and thereby the total metabolic cost) to supply all synaptic sites. Therefore, we wanted to test the dynamic aspects of our model and chose a paradigm focusing on the spine integration of somatically synthesised *Shank3* proteins at varying distances from the soma[18]. Energy consumption is not directly assessed in this paradigm; yet, it is a

valuable test for the underlying dynamics used in our model, e.g., the spine-dendrite coupling, which are fundamental for the energy calculations presented in the previous paragraphs. The availability of quantitative measurements of photoactivated *Shank3* proteins makes it possible to directly compare the experimentally measured time scales with the space and time dynamics emerging from the mobility of dendritic protein and the dynamic coupling of the dendritic shaft with the spines in our model. To replicate the experimental paradigm in silico, we carefully researched the necessary parameters of *Shank3* mRNA and protein from the literature[3,10,13,14,16,18,61,99,104,105,137–141] (see Supplementary Notes, section 'Simulating the spread and synaptic integration of somatic proteins'). Following[18], we assumed that from $t = 0$ on, all somatic proteins at $x = 0$ get labelled ('photoactivated'), and as time progresses, the photoactivated proteins spread throughout the dendritic shaft. With some delay, the photoactivated proteins also enter dendritic spines (Fig. 5A, D). Let us emphasise that the overall distribution of proteins in the dendrite and spines, i.e., the sum of photoactivated and non-photoactivated molecules, remains constant[18]. Recorded the increase in fluorescence due to the integration of somatically photoactivated *Shank3* in spines located on average 25, 50, 75, 105 $\mu m$ from the soma (Fig. 5B, E). We repeated this in silico by tracking the number of photoactivated proteins in the spines of interest (Fig. 5A, D). Similar to[18], the proximal-most spine starts to saturate after around 4 h. In contrast, the fluorescence of the remotest spine is still low after 8 h (Fig. 5D, E). To allow a more quantitative validation of our predictions, we normalised simulation results and experimental recordings with the proximal-most spine's data (Fig. 5C, F), showing a good agreement of our simulations with the experimental measurements.

To summarise, our results show that the experimentally reported temporal protein dynamics can be captured by our minimal energy model framework.

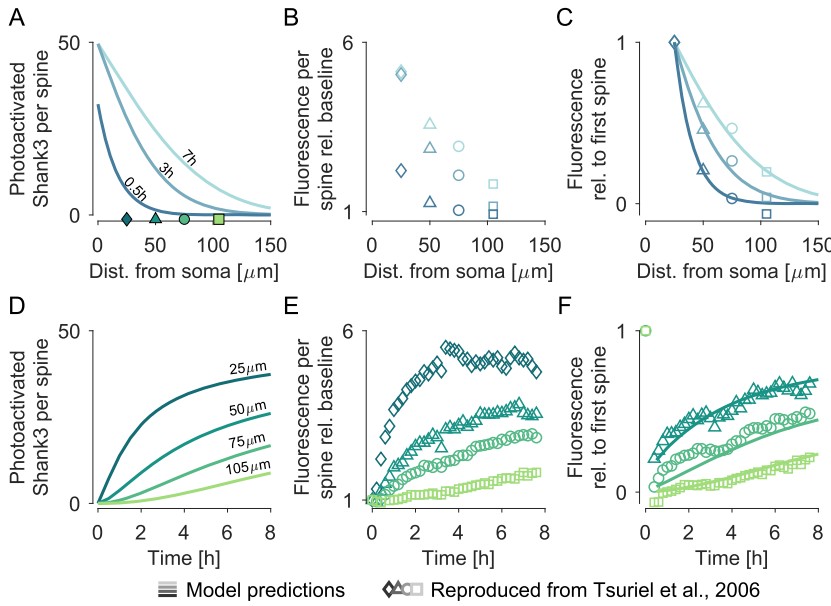

**Fig. 5 | Predicted model dynamics reproduces experimentally observed spread of somatic proteins in space and time. A** Simulated spatial spread of somatically photoactivated *Shank3* proteins in dendritic spines along a linear dendrite after 0.5, 3, 7 h. Proteins are continuously photoactivated at the soma. The first 150 $\mu m$ of the dendrite are shown and the spatial locations of four spines of interest (25, 50, 75, 105 $\mu m$ distance from soma) are highlighted in green. **B** Experimentally observed increase in fluorescence after 0.5, 3, 7 h at the imaged spines (on average 25, 50, 75, 105 $\mu m$ distance from soma) as reported from ref. 18. **C** Overlay of predicted A and experimentally observed B spatial spread of fluorescence across spines. Data were normalised with those of the proximal-most spine to allow comparison of simulated and experimentally observed data. **D** Predicted temporal integration of photoactivated *Shank3* in the four spines of interest. **E** Experimentally observed time course of fluorescence during the whole imaging period of 8 h at the imaged spines as reported from ref. 18. **F** Overlay of predicted D and experimentally observed E time course of fluorescence in spines of interest. Data were normalised with those of the proximal-most spine to allow comparison of simulated and experimentally observed data.

## Discussion

Here, we presented an original modelling framework quantifying how energy shapes spatial distributions and counts of mRNAs and proteins in neural dendrites. We harnessed numerous experimental reports describing the energetic costs associated with transcription, translation, and transport, as well as experimentally reported physical properties such as half-lives, diffusion constants, active transport, and spine-dendrite coupling. By linking energy expenditures to molecular dynamics, we derived energy-optimal spatial distributions as a function of the dynamical properties of the corresponding molecular species.

Our results revealed that mRNA and protein localisation patterns significantly influence the energy needed to maintain them. Specifically, we found that protein species with low half-lives benefit energetically when their mRNAs localise in dendrites rather than somata. Similarly, our model calculations predicted that mRNA species with long non-coding regions or long half-lives profit energetically from moving into dendrites and translating locally. We could experimentally confirm eight major model predictions using screens of transcriptome and proteome[3,13–18]. Interestingly, our predictions matched quantitatively the experimentally reported distribution of total mRNA and protein counts across thousands of species. Our results revealed that mRNA and protein copy numbers and spatial arrangement of tens of thousands of different molecules inside neural dendrites can be understood with a minimal energy principle for each molecular species.

We represented each (synthetic) mRNA and protein species in our model via seven dynamical parameters, including half-lives and diffusion constants. Experimentally measuring all seven parameters for each of the tens of thousands of molecular species reported in the neurons has not yet been attempted due to the challenging nature of quantitative measurements in sub-cellular compartments. To circumvent this challenge, we devised a strategy sampling from the parameter distributions that were available experimentally and computer-generated surrogate molecular species. For some model parameters, multiple different values were readily available (Supplementary Fig. 1), while parameters characterising diffusion and transport properties have yet to be measured for most molecules in neurons. Let us note that the majority of the available data on molecular parameters entering our model came from hippocampal pyramidal cells. Since this is a commonly studied cell type in the context of molecular turnover, we drew on this available experimental literature to build our model and test its predictions. However, future datasets characterizing molecular turnover and trafficking across cell types could help generalise our model findings across the diversity of neural cell types. The computer-generated $3^7 = 2187$ synthetic molecules which obtained by sampling a lower, an upper, and an intermediate value for each model parameter from the available data sets allowed us to overcome experimental limitations and calculate the minimal energy solutions not only for individual proteins of interest but at the level of protein populations. To benchmark our model predictions not only in the spatial domain (Figs. 2, 3, 4) but also test them in the time domain, we have considered the temporal dynamics of photoactivated *Shank3* protein (Fig. 5) reported in[18]. Here, we found that the evolution of the *Shank3* occupancy across spines that were 25, 50, 75, or $105\,\mu m$ from the soma was quantitatively in line with the time dynamics of our model confirming not only our optimal energy predictions but also the protein dynamics in space and time.

Notably, we could confirm our model predictions not only at the level of molecular classes but also for selected molecular species. Our model predicted that mRNA classes with long non-coding regions will be preferentially localised in the dendrites. Intuitively, this can be understood by considering the increased transcription cost these transcripts incur which makes them energetically more valuable and

therefore require reducing the degradation probability for the corresponding proteins on route to optimise the energy budget. Interestingly, this model prediction can be observed not only at the population level (Figs. 2, 3) but also in mouse hippocampal neurons within the mRNAs coding for the same protein. An et al.[80] reported that the mRNAs of the prominent synaptic protein BDNF preferentially localise to the soma if they have a short 3'-UTR sequence. In contrast, BDNF mRNAs with long 3'UTR sequences preferentially localise in dendrites, confirming our model predictions at the level of individual protein species. This finding has been further corroborated across genes[14].

Interestingly, our simulations also suggested that proteins with higher diffusion constants preferentially localise mRNA in the soma (Supplementary Fig. 8A). We curated and expanded a previously published list of diffusion constants[19] ($N = 32$) matching each reported protein with the corresponding mRNA enrichment scores[16] (Supplementary Table 1) revealing a trend in line with our prediction (Supplementary Fig. 8B). As future studies provide more diffusion records, this model-predicted relation could be further tested.

Our model also offers an interesting perspective on active protein transport in dendrites. In comparison to mRNA transport, which has been researched quite extensively, reports on active protein trafficking in dendrites are rather sparse (e.g., refs. 43–47,142). In our modelling framework, mRNA transport is predicted to be almost always energetically more efficient than protein transport to maintain the baseline protein distribution across spines (Supplementary Figs. 3, 4), particularly in shorter dendrites. Future experiments targeting the population of actively trafficked proteins can shed light on the statistics of actively trafficked proteins in relation to the locally synthesised protein group.

Surprisingly, we found that a handful of very abundant protein species dominate the metabolic costs with ~$10^9$ ATP/s per species (Supplementary Fig. 7). Yet, even when we excluded the most cost-intense 5% of molecular species, we still found that the energy demand integrated across the remaining 95% of proteins is high, ~$6.5 \times 10^8$ ATP/s (assuming $N = 10{,}000$ in the 95% group). Interestingly, these numbers are substantially higher than previously estimated and are compatible with the energy demand reported for spiking activity[143–145]. Our calculations suggest that the maintenance of the neuronal proteome consumes a significant share of the energy budget. These results provide a quantitative link between protein synthesis-dependent synaptic plasticity and cellular energy metabolism[146–149].

While metabolic costs were our primary focus, our model also provided predictions about microtubular transport, intracellular molecule localisation and distribution, protein dynamics in time and space, and total mRNA and protein counts. By integrating the main processes driving mRNA and protein distributions in neurons into a single mathematical model we combined biological plausibility with mathematical tractability. While most previous studies focused on one or two aspects of a cell's molecular landscape, our model allowed for the first time to study the interplay of different dynamic processes and energy consumption within one common framework. Additionally, our model could serve as a powerful platform addressing how differences in branching translate into molecular localisation preferences and different energy demands as more experimental evidence becomes available about which cell types do or do not preserve the number of microtubular tracks between the mother and daughter branches, on how the area/circumference ratios at branch points[19] determine the distribution of microtubules, mRNAs and proteins between the mother and daughter branches. Future studies can also adapt our model framework to investigate local and global correlations between individual parameters and integrate additional intracellular processes, e.g., temporal and spatial dynamics and metabolic costs associated with the temporal changes in synaptic copy numbers during plasticity, opening doors for a comprehensive view on molecular dynamics. Similar to the computational studies at the neural network level

emerging from the sparse coding hypothesis that was rooted in the energy restrictions at the activity level[150–153] future research addressing to what degree energy optimisation strategies play a role at the molecular level across the evolutionary, developmental, and cell type-specific dimensions will help us gain a deeper understanding of the functional architecture of a cell at the subcellular level and understand the role of the neuronal metabolism in disease[154–157]. For example, single cell transcriptomics and spatial transcriptomics analyses[158,159] revealing differences in molecular copy numbers and transcript localisation across neural compartments, neural cell types or disease states can now be interpreted with the concept of energy minimisation in mind.

## Methods
### Mathematical methods
Here, we give a brief overview of our model. The mathematical derivations with their underlying motivations and all parameters are discussed in much greater detail in the Supplementary Notes.

To model mRNA and protein distributions in dendrites ($m$, $p$) and spines ($p_{\text{spine}}$) along a linear dendrite of length $L$, we solved equation 1 with the following associated boundary conditions:

$$0 = \frac{\partial}{\partial x} m(L), \tag{2a}$$

$$0 = \frac{\partial}{\partial x} p(L), \tag{2b}$$

$$0 = \begin{cases} D_p \frac{\partial}{\partial x} p(0) - \frac{r_{\text{soma}}}{1 - r_{\text{soma}}} \frac{\tau \bar{D}_m}{\lambda_m} \frac{\partial}{\partial x} m(0), & r_{\text{soma}} < 1, \\ \frac{\partial}{\partial x} m(0), & r_{\text{soma}} = 1, \end{cases} \tag{2c}$$

$$0 = p(L) - \frac{1}{\pi_p} \frac{\phi}{(1 - \phi)} \rho \eta_p \tag{2d}$$

Conditions 1 and 2 close the dendrite at the tip. Condition 3 implements the share of mRNAs in the soma $r_{\text{soma}} \in [0, 1]$. Condition 4 ensures that all spines are filled with a share $\phi \in [0, 1]$ of the maximal protein copy number per spine $\eta_p$. Here, $\pi_p$ is the permeability of the spine, i.e., the rate of proteins entering spines divided by the rate of proteins leaving them.

$\bar{D}_m$ is the ensemble mRNA diffusion constant, which we obtained by fitting it to a 3-state model of dendritic transport[48]:

$$\frac{\partial}{\partial t} m_+ = -v \frac{\partial}{\partial x} m_+ - \beta m_+ + \alpha m_0, \tag{3a}$$

$$\frac{\partial}{\partial t} m_- = v \frac{\partial}{\partial x} m_- - \beta m_- + \alpha m_0, \tag{3b}$$

$$\frac{\partial}{\partial t} m_0 = \beta m_+ + \beta m_- - 2\alpha m_0 + D_m \frac{\partial^2}{\partial x^2} m_0 - \lambda_m m_0 \tag{3c}$$

with appropriate boundary conditions. Here, $m_+$ and $m_-$ are the anterogradely and retrogradely transported populations, moving with instantaneous velocity $v$. Molecules switch from these states to the diffusive, resting state $m_0$ with rate $\beta$ and re-enter each transported state with rate $\alpha$. $D_m$ is the diffusion constant of the non-transported state $m_0$.

### Energy costs
We assigned energetic costs to the resulting mRNA and protein distributions for active transport, synthesis and degradation.

The cost per second to transcribe the mRNAs of a species with $N_{\text{nt}}$ nucleotides is given through

$$C_{\text{transcr}} = C_{\text{nt}} N_{\text{nt}} \underbrace{\lambda_m \left( m_{\text{soma}} + \int_0^L m(x) \, dx \right)}_{\text{total mRNA decay}}, \tag{4}$$

where $C_{\text{nt}} = 2.17 \text{ATP}$ is the cost of chain elongation and posttranscriptional modifications in eukaryotes[87,88]. Similarly, the cost of protein synthesis and degradation (subsumed under 'translation cost') is

$$C_{\text{transl}} = C_{\text{aa}} N_{\text{aa}} \underbrace{\tau \left( m_{\text{soma}} + \int_0^L m(x) \, dx \right)}_{\text{total synthesised protein}} \tag{5}$$

for a protein with $N_{\text{aa}}$ amino acids. $C_{\text{aa}} = 5 \text{ATP}$ covers the cost to elongate and later degrade each amino acid from the chain[87–89]. Note that, in the baseline mRNA and protein distributions, the numbers of synthesised and degraded proteins always match, wherefore we can combine synthesis and degradation costs in one term. For the transport cost, we combine the cost $C_{\text{cargo}}$ to transport a single cargo (e.g., an mRNA granule) at a given speed $v$ (see equation 3 in the Supplementary Notes) with the total number of cargoes (here: mRNA granules):

$$C_{\text{transp}} = C_{\text{cargo}} \underbrace{\theta_m \int_0^L m(x) \, dx}_{\text{total transported mRNA}}, \tag{6}$$

The total number of currently transported mRNA granules is determined by the total number of granules and the share of transported granules $\theta_m$. $\theta_m$ is directly related to the switching rates $\alpha$, $\beta$ in equation 3 (see the Supplementary Notes, section 'Capturing active mRNA transport in our model', for details). We assume that the cost to transport a single cargo with 1 $\mu m/s$[22,23,25,28–38,41] amount to 125ATP/s[23,34–36,90–97]. For a thorough discussion of the equations and the associated parameters we refer the reader to our detailed Supplementary Notes.

### Experimental data analysis
Related to Fig. 3A: mRNA half-lives of somatic vs dendritic transcripts were available from ref. 14. We adopted their classification of transcripts as somata-enriched or neurite-enriched, which was based on a differential expression analysis (see Methods of ref. 14).

Related to Fig. 3B: We used data published by ref. 16 to analyse differences in transcript length between somata-enriched and neurite-enriched mRNAs. Single mRNAs were classified as somata-enriched or neurite-enriched if enriched at least 2-fold in somata or neurites. We assessed the significance of the difference in mRNA length between the two groups with a two-sided Wilcoxon rank-sum test. We bootstrapped each group 10,000-fold to capture its statistics more accurately.

Related to Fig. 3C: We used a published a resource[3] containing protein copy numbers per spine for ~100 proteins. This dataset did not record mRNA enrichment in neurites; therefore, we cross-matched it with mRNA enrichment scores by ref. 16. We chose ref. 16 because their data showed the most extensive overlap with the protein species we analysed[3]. The matching was done based on gene names. Because we[3] imaged mouse brains, whereas ref. 16 used rats, we used additionally a mouse-rat homologue table from ref. 160. Classification of entries as somata-enriched and neurite-enriched was based on mRNA enrichment. In contrast to our further analyses, we considered mRNAs somata-enriched (neurite-enriched) in the case of 1-fold enrichment, not 2-fold. We applied this less strict criterion due to our relatively few data points[3] compared to the other databases (Supplementary

Table 2). Significance was confirmed using a two-sided Wilcoxon rank-sum test. No bootstrapping was performed.

Related to Fig. 3D: Protein half-lives in rat cortex homogenate were extracted from ref. 13 and cross-matched with mRNA enrichment in neurites vs. somata from ref. 16 based on gene names. We used protein half-lives from ref. 13 because they measured lifetimes in vivo and reported more data points than an earlier study[101]. ref. 13 showed that the results of the two studies were not identical but highly correlated. We took the mRNA enrichment data from ref. 16 because they provided the highest number of cross-database matches. After the matching procedure, we classified entries with at least 2-fold mRNA enrichment in somata (neurites) as somata-enriched (neurite-enriched). Lastly, we performed a two-sided Wilcoxon rank-sum test on the protein half-lives of the two resulting categories. We bootstrapped the protein half-lives of both groups 10,000 times.

Related to Fig. 3E: To evaluate the length of proteins associated with different mRNA localization patterns, we used the database from ref. 16. It reported mRNA enrichment and the sizes of the corresponding mRNA coding sequences, measured in nucleotides, which equals three times the amino acid number per protein. We categorised entries again as somata-enriched (neurite-enriched) if the mRNAs were enriched at least 2-fold in somata (neurites). Statistical differences in protein length between the somata-enriched and neurite-enriched entries were evaluated with a two-sided Wilcoxon rank-sum test. Finally, we performed 10,000-fold bootstrapping to visualise the underlying statistics better.

Related to Fig. 4B: Total mRNA counts were derived from data published by Perez et al.[15] and Zeisel et al.[17]. Both studies performed single-cell mRNA sequencing with unique molecular identifiers, allowing them to quantify the number of mRNA molecules in a cell. Perez et al. report a detection probability of 25%, Zeisel et al. 20%. Both values align with another study examining RNA sequencing sensitivity[161]. Therefore, we multiplied all mRNA counts with 5 or 4, respectively. Perez et al. provide data on the mRNA count in somata and dendrites, which we added to obtain a 'per cell' count. To compare the mRNA counts per neuron with our simulations, we decided to use cells that best fit the assumptions of our simulations: we restricted our analysis to glutamatergic neurons (Perez et al.) and CA1 pyramidal cells (Zeisel et al.). We finally computed the mean count of mRNAs per gene within each cell group. Protein counts per neuron for >6000 protein species were taken from our own measurements[3].

Related to Fig. 4D: For the abundance comparison between species with neurite- or somata-enriched mRNAs, we used the label-free quantification of mRNA and protein provided by ref. 16 and the same studies' classification of neurite- and soma-enriched mRNAs based on at least 2-fold enrichment in neurites or somata. For visual consistency with Fig. 4C, we transformed the total abundance (somata plus neurites) to a $log_{10}$-scale.

A summary of the databases used in this study and their respective sizes are shown in Supplementary Table 2.

### Reporting summary
Further information on research design is available in the Nature Portfolio Reporting Summary linked to this article.

## Data availability
All experimental data used in this study have been previously published and their online source links are incorporated into the analysis code available here https://github.com/CompNeuroTchuGroup/energyDeterminesMoleculesCode. No new data have been acquired for this study.

The mRNA data from ref. 14, Table S1, are available under https://www.cell.com/cms/10.1016/j.neuron.2018.03.030/attachment/62fccf6a-ed86-4dfa-bc7d-13955e782a68/mmc2.xls.

The mRNA and protein data from ref. 16, Supplementary Data 2, are available under https://static-content.springer.com/esm/art%3A10.1038%2Fs41467-017-00690-6/MediaObjects/41467_2017_690_MOESM3_ESM.xlsx.

The mRNA data from ref. 15, Supplementary file 3, are available under https://cdn.elifesciences.org/articles/63092/elife-63092-supp3-v2.xlsx.

The mRNA data from ref. 17 are available under https://storage.googleapis.com/linnarsson-lab-www-blobs/blobs/cortex/expression_mRNA_17-Aug-2014.txt.

The mRNA data from ref. 105, Supplementary file 1, are available under https://doi.org/10.7554/eLife.34202.029.

The mRNA data from ref. 99, Supplementary Table 2, are available under https://oup.silverchair-cdn.com/oup/backfile/Content_public/Journal/dnaresearch/16/1/10.1093/dnares/dsn030/2/dsn030-dsn030_Table2.xls?Expires=1740158069&Signature=UkEZXUbJXI7CGtDsXPGxqVb7fmbHrQEeSQiIL7HxcK3k1onrScuWvIXw5Jt0JNacytqpy1Aq1AE09m4vOVcywdjJILEaLGmjAqQHaMBTFKYvJxLOQ0sJOf6osr2G-a1ob62ZSgkXyK-6Z8VqOGVpCiUcVB8QQ669wAhTMM-wQZCxbYxWcHh7kLQpIB-HtYraddBeo-hWQ2Y8tEtp2ZopXGWhuX1OLrEhndV5mP8oRJoIvO7aFsKD-hsXvaphXJ--7L1XJx7R-YvSuyUG419GXpDgFzhgZ4IdQPQqmxpoeVynkUiGXufZDlhGfjmHk6cBQMkfgZcGWRaT2lLfiWbQ_&Key-Pair-Id=APKAIE5G5CRDK6RD3PGA.

The mRNA data from ref. 60, Supplementary Table 3, are available under https://static-content.springer.com/esm/art%3A10.1038%2Fnature10098/MediaObjects/41586_2011_BFnature10098_MOESM304_ESM.xls.

The mRNA data from ref. 120, Supplementary Table 9, are available under https://genome.cshlp.org/content/suppl/2006/06/14/13.8.1863.DC1/yang.pdf.

The mRNA data from ref. 61, Dataset S3, are available under https://doi.org/10.1073/pnas.2113929118.suppl_file/pnas.2113929118.sd03.xlsx.

The mRNA data from ref. 10, Table S5, are available under https://ars.els-cdn.com/content/image/1-s2.0-S0896627312002863-mmc2.xlsx.

The mRNA data from ref. 134, Table S2, are available under https://ars.els-cdn.com/content/image/1-s2.0-S1097276523004689-mmc3.xls.

The mRNA data from ref. 162, Table S3, are available under https://ars.els-cdn.com/content/image/1-s2.0-S2211124719311556-mmc5.xlsx

The protein data from ref. 13, Supplementary Data 1, are available under https://static-content.springer.com/esm/art%3A10.1038%2Fs41467-018-06519-0/MediaObjects/41467_2018_6519_MOESM3_ESM.xlsx.

The protein data from ref. 3, Supplementary Table 2, are available under https://static-content.springer.com/esm/art%3A10.1038%2Fs41593-021-00874-w/MediaObjects/41593_2021_874_MOESM4_ESM.xlsx.

The protein data from ref. 104, Supplementary Data 2, are available under https://static-content.springer.com/esm/art%3A10.1038%2Fs41467-018-03106-1/MediaObjects/41467_2018_3106_MOESM5_ESM.xlsx.

The protein data from ref. 101, Table S2, are available under https://doi.org/10.1073/pnas.1006551107/suppl_file/sd01.xls.

The protein data from ref. 163, Dataset S2, are available under https://doi.org/10.1073/pnas.1720956115/suppl_file/pnas.1720956115.sd02.xlsx.

## Code availability
The code to run the simulations, analyze the simulated and experimental data and generate the figures in this manuscript is deposited on GitHub (https://github.com/CompNeuroTchuGroup/energyDeterminesMoleculesCode)[164].

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

## Acknowledgements
This project has received funding from the European Research Council (ERC) under the European Union's Horizon 2020 research and innovation program ('MolDynForSyn,' grant agreement No. 945700). We also acknowledge support from the University Hospital Bonn (T.T.), the Ger-man Research Foundation via CRC 1080 (T.T.), the Forschungszentrum Translationale Neurowissenschaften Mainz (K.M., T.T.), the University of Frankfurt Interdisciplinary Neuroscience program (C.B.), the German Academic Scholarship Foundation (C.B.), and SFB1286/A03 (S.O.R.). We thank N. Ziv for sharing the Shank3 protein data published in[18]. We thank all members of the Tchumatchenko lab, D. Schmucker and E. Schuman for fruitful discussions. We thank L. Wenning, A. Nold, S. Wagle and J. Petkovic for valuable comments on the previous version of the model.

## Author contributions
T.T. conceived the study, C.B. and T.T. designed the model framework. C.B. built and analysed the mathematical model, K.M. prepared the experimental dataset pairings, C.B. and K.M. analysed experimental data, S.O.R. contributed protein data analysis. C.B., K.M., S.O.R. and T.T. wrote the manuscript.

## Funding

## Competing interests
The authors declare no competing interests.
