## [Transparent Peer Review file · Nature Communications]

How energy determines spatial localisation and copy number of molecules in neurons

Corresponding Author: Professor Tatjana Tchumatchenko

Version 1:

Reviewer comments:

Reviewer #1

(Remarks to the Author)

This paper explores whether energy arguments can explain why some neural proteins are locally synthesized whereas others are somatically synthesized. It concludes that energy might be able to explain a range of phenomena. The insight is novel and might interest a wide range of cellular neuroscientists and computational modellers.

Overall the study uses an original but well justified diffusion approximation approach and simulations.

Major comments.

1. It is assumed that there is as much motor movement towards as away from the soma. In other words, the whole effect of motor transport is to speed up the diffusion constant (as is shown in one of the figures, btw the relation plotted there should be easy to derive using jump moments). However, diffusive transport is highly inefficient for long distances and a small amount of net transport towards the synapse, to be included as a drift term in the diffusion eq, could make a big difference. So I think it is important to check that.

Less important but perhaps good to point out explicitly is that when there is a concentration gradient, a net current will be present which might explain some experiments.

2. Also the 95% percent protein requirement might play a critical role. Re-running for a few other values would be useful.

3. Presentation could be sharper. The maths is actually straightforward but the informed reader will need to dig through a lot of supplement to find what is done. It could be much simpler: "We solve the steady state diff eq, subject to $m(0)=...$, etc". The manuscript mentions 8 phenomena and 7 parameters. A table/ list specifying them in the main text would help.

4. Fig 5 is presented as in line with the model, but I didn't see how it is related to energy. I think it simply verifies the diffusion model (which is useful of course). The data might restrict the amount of drift mentioned above.

5. The discussion mentions that the total energy for protein synthesis and transport is comparable the spiking cost. This is an important claim and would be worth mentioning in the abstract.

Minor:

Data analysis (Fig 2) is a bit odd with this middle bar. Why not simply split the simulation data in two?

Is there evidence that protein synthesis wears out the mRNA, speeding up degradation? If so, that would be worth mentioning and perhaps including.

p2 per mRNA and second => per mRNA per second

p2 what is a 'cargo'?

p2 the prominent => a prominent

(Remarks to the Author)

Neurons are cells with extensive branched processes which form synaptic connections. The proteins within synapses determine their function and it is known that synapses are molecularly complex and diverse. It is therefore of interest to understand the basic cellular mechanisms that regulate the spatial distribution of the mRNAs and synaptic proteins in the dendrites and axons of neurons. Numerous studies (cited by the authors) have isolated mRNAs and proteins from either the soma or dendritic fraction and shown differential molecular compositions, lifetimes and other parameters. In the current paper the authors address the issue of energy constraints on the spatial distribution of mRNAs and proteins in neurons using a computational modelling approach. The authors then draw up on previously published datasets for use in their models and for validation. In general, I like this paper as it addresses the role of energy metabolism and generates interesting hypotheses that at some future time could be tested.

Issues that need to be addressed:

1. A major claim of this paper is “We found that energy minimisation principles are able to explain experimentally reported exponential distributions of mRNA and protein copy numbers and their spatial profiles”. This is an overstatement that needs to be fixed. The “exponential distributions” the authors present are shown as values plotted as a function of distance from the soma (see for example, Figs 1, 4, 5). What these figures have in common is that they present gradients from soma to distal dendrite of molecular features. The authors repeatedly claim to have validated these gradients and refer to publication numbers 3, 13-18. I have read these papers and in none of them do I find any quantification of mRNAs or proteins along dendrites. These papers all contain data from either soma or dendrites, not gradients. So how is it possible to use these data to claim that the predicted gradients have been validated. To remedy this situation the authors can take one of two courses of action: (i) make it very clear in the text that they have not validated their predicted gradients, (ii) use datasets from the published literature that show real gradients of molecular distributions along dendrites.
2. It would help if the authors could refer to and describe manuscripts where dendritic gradients have been measured at the single-synapse resolution level and note whether they are exponential or not. Furthermore, it is my understanding that some molecular gradients from soma to the distal dendrite can be either diminishing (as modelled in this paper) or increasing (e.g. AMPA receptors). How can their model accommodate gradients where the protein concentration increases as a function of distance from the soma? It is also the case that observed gradients of synapse proteins do not follow an exponential and have more complex distributions, such as input-specific differences. How does the model deal with the observation that certain synaptic proteins are more homogeneously expressed in synapses across the dendrite while others have very heterogeneous patterns of expression?
3. Relevant to the above points, in Figure 5 the title reads “Simulations reproduce experimentally observed etc”. Fig.5 A-C have no experimental data for support, but Fig.5E does show experimental data for comparison. But Fig.5E only shows fluorescence increase within 4 spines over time and not “spatiotemporal spread of proteins in dendrites” that the title of the figure states. It would be desirable to have experimental data accompanying Fig.5A and B to convince the reader that their model indeed reproduces protein distributions in dendrites and spines, and to show the comparison between the model and the experimental data.
4. The authors use data from a 100 um segment of a dendrite to model distributions up to 1000 um (Figure 1C). The authors should make it clear that this is the assumption behind their model and note that dendrites have more complex gradients and distributions that will not be accommodated by their model.
5. The authors state: “Functionally, keeping the protein copy number in synapses within a specific range for each molecular species is essential (Wang et al)”. Therefore, we required in our model that all spines be filled to $\geq 95\%$ of their capacity for a protein of interest”. Reading Wang et al I could not find reference to the statement about specific ranges being essential. It also doesn't make sense because labelling of synaptic proteins using immunolabelling typically shows considerable heterogeneity (as shown in papers cited in this manuscript including those from Dr Rizzoli). Furthermore, what does it mean to say that spines are filled to 95% capacity? Does that mean any one protein occupies 95% and other proteins the remaining 5%?
6. The authors use protein abundance data from Figure 4C of Helm et al for the copy number of proteins in individual dendritic spines and in particular the copy of number for CamKII of 32,336. The same figure notes that PSD95 copy number is $\sim 1,000$ and NMDA GluN2B subunit is several hundred. It is widely thought, based on electrophysiological data, that there is ~ 10 NMDARs/synapses which is about 20 copies of GluN2B (see work of Roger Nicoll). Similarly, the estimates for PSD95 based on MS and super-resolution studies is about 10-fold lower than those in Helm. Indeed, in Helm et al Extended Data Fig 8 the graph indicates their estimated copy number is up to 10-100 fold higher than in other studies. It should also be noted that Helm et al did not make direct measurements of copy number per synapse. Furthermore, they reported only two types of spines and not the whole range of spines (which will have implications for the computational modelling). While the current manuscript is not one that might settle these differences in the datasets used for the model, the authors should present results using these lower numbers alongside their existing models based on Helm.
7. What type of neuron (excitatory/inhibitory, pyramidal/purkinje/other neuron type) is being used as a model for this study as the cells have largely different morphologies, spine densities, number of main branches stemming from the soma and all of these would significantly influence the derived mRNA and protein distribution calculations.

8. Please justify (in main text) why the passive diffusion-mediated molecule movement and microtubule-mediated active transport are reduced into a single variable. The authors should justify this by providing examples of experimental data that demonstrate the validity of this approach. Can the authors justify why they can treat microtubule-mediated active transport as random walk and refer to experimental data.

9. The Methods section contains a single subheading “Experimental data analysis”. This only really refers to the datasets that were used and provides very little detailed methods information and certainly not enough to replicate the study. Some of this information is in the supplemental information section and should be in methods.

10. In results section (page 2) the authors refer to “using experimentally reported average run duration, run velocities, and the ratio of transported cargoes” to mathematically map the transport process to a diffusion process. Please create and, in main text, refer to a table outlining the references for the “experimentally reported” measurements, the values associated with each publication and whether the values refer to mRNA or protein species.

11. How different are the calculated transcription costs for the following scenarios: (a) most mRNA is soma-localised and protein is synthesized in soma and then transported to target synapses, (b) mRNA is preferentially localised in dendrites and most protein is synthesized locally. Please discuss in main text.

12. On page 3, the authors state that “active mRNA transport is more energy effective compared to active protein transport”. In the ‘Transport cost’ section of the ‘Supplemental Information’ the authors do not refer to any differences or factors that influence the costs of mRNA vs protein trafficking differently. The supplementary figures S3 and S4 they refer to show stacked bar plots and box plots with no context and no intuitive way to help the reader understand how their statement is supported by data. In the same paragraph on page 3, it says that “trafficking mRNAs from the soma into the dendrites can reduce the cost of transcription and translation because fewer mRNAs and protein copies need to be produced”. Please explain where this statement is coming from and, if it aids the naïve reader, add an illustration.

13. The “cell’s general ‘infrastructure’ costs” (page 3). These costs will scale with the location in the cell (close or in the soma vs distal dendrites). This is especially crucial for protein translation-related infrastructure. A calculation of the average costs for such infrastructure in different parts of the cell should be estimated and then the workload these infrastructure components get could be used to split the costs per protein species and show how substantial the costs end up being. If they are minor, they can stay as a supplementary figure, if major, they should be included and integrated into the main model.

14. For the statement “CaMKIIalpha proteins are not actively transported in dendrites”, I can’t find in the Shen et al, 1998 referenced paper any data supporting this statement. Shen only refers to CaMKIIalpha capture and transfer from dendrite to the dendritic spine and not the trafficking from soma to and along the dendrites. Moreover, the reference to Hirokawa and Takemura, 2005 paper only speaks about CaMKIIalpha mRNA trafficking and not protein.

15. For the sentence “Remarkably, we found that the experimentally measured mitochondria distribution in dendrites is consistent with our model predictions.” Please clarify which exact predictions. To claim ‘consistency’ you would require support from a statistical test.

16. In Figure 1F, the graph of energy needs indicates that beyond 200 um there is very little need for energy, which is implausible. Please clarify.

17. Figure 4a. The graphs show that the most abundant mRNA has a log10 value of 15, which is 1.176 mRNAs for a dendrite of 1000um. For the right panel showing protein, for the most abundant protein, the total predicted copy number is less than log10 20, which equals ~1.3 proteins across the entire dendrite. How can that be correct?

18. To estimate and predict a single neuron’s total protein and mRNA copy number (in absolute numbers), you may have to know how big your model neuron is, the total length of its dendrites, and the number of spines/synapses as all those measurements directly influence how much protein and mRNA will be needed to fill up the relevant synapses. Please discuss and provide some numbers.

19. Figure S12: the correlation between total mRNA and protein numbers appears unrealistically high given previous literature which typically shows lower values (see Liu et al, 2016 cited by the authors). The authors should discuss why their values appear high and the implications of that for their model.

20. In the first paragraph of the ‘Results’ section the authors refer to their model capturing the mRNA and protein density inside the soma and other parts of the neuron. Are you referring to absolute number of molecules or number of molecules per some unit area? If per unit area, what units are used? Please specify. Straight after they mention protein concentration. Is it meant to be the same as protein density? Or is this a different measure? What are the units for concentration? Is it 2D or 3D spine/shaft/soma?

21. Throughout the text the authors refer to “Supplemental Information” without directing the reader to the actual part of the Supplemental information, which contains 13 figures, 4 tables and 14 pages of “Additional Details on the Mathematical Model and Parameters”. The authors must specify the precise figure/table/section of the Supplemental information at each point in the text.

22. The synthesis and degradation rates are denoted by a single variable. Given existent evidence that protein synthesis

and lifetime are synapse-specific, how accurately does a single variable per cellular process capture the diversity? Please discuss.

23. Since the proteins are not uniformly distributed among all synapses along the dendrite, please discuss the known mechanisms involved in active capture of the protein into specific spines and their associated costs. Are these costs substantial to affect the calculations in the model?

24. The authors should discuss how the model will be affected by branching in the dendrites which obviously could affect the distributions of mRNA and proteins in more distal parts of the neuron and will likely significantly depend on the neuron type.

25. The datasets that the authors refer to are derived from many different sources including primary neuronal cultures, whole brain homogenates, different species, different brain areas, different cell types and different parts of the dendritic tree. This information could be added to Table S1 and S2. The authors need to specify which dataset and its source material is being referred to in the main text at the point the data is being used/described. For example, Figure 1C shows data from Fonkeu et al but it is not clear if the measure is from a single dendrite or population, which part of a dendrite, from primary cell culture or in vivo, and what cell type.

26. The “protein diffusion constants” presented in Table S1: are they passive transport, microtubule-associated transport or a mix? Should be specified where possible.

27. On page 3, where the authors explain what they consider when calculating the “energy cost (in ATP per second) necessary to maintain each protein distribution”, they do not mention protein degradation. Protein degradation is an essential component of maintenance of stable protein distribution and proteasome-mediated degradation does cost ATP and really should be included in the considerations and should be well explained in main text.

28. When introducing the calculations of energy costs in ATP for the different processes, including transcription, translation and active transport, it would be helpful to include a figure or a table with values for an example protein and its distribution in an example dendrite of some unit length. It would help put those numbers into perspective and more intuitively visualise the most costly processes.

29. For fig. 1A, please expand the description and remove the passive diffusion from the illustration since it does not require energy.

30. On page 4, the authors say that they “defined three groups along each of the seven dimensions: small, medium, and large”. Were the three groups defined by percentile or other measure of the distribution? Since the 3 groups are defined not by a range but by a single point value, how was the range for the 3 groups defined? Please discuss and clarify.

31. In Fig.2 the authors present how the “preference for dendritic vs. somatic mRNA changes as a function of the parameter of interest” and present the plots for 5 parameters out of the 7 that were mentioned earlier in the paragraph. Where are the other two and why not include and discuss them?

32. Page 4-5, authors refer to Fig. S8 saying that their “results hold across different dendritic lengths”. Fig.S8 shows nothing to do with dendritic lengths. Please fix.

33. Page 5, authors refer to ‘somata-enriched’ and ‘neurite-enriched’ mRNA species that were used in their analysis but it is unclear what study the data was taken from (please reference in main text), whether the data is from intact tissue or cell culture, what parts of neurites (i.e. apical/basal/ proximal/distal) were included in the analysis and what cell type was analysed. Please clearly state in text.

34. On page 5, authors state “we found that species with neurite-enriched mRNAs indeed exhibit significantly lower protein half-lives (Figure 3C)”. Figure 3C shows data for protein copies per spine and there is no mention of protein half-lives in this figure. Please fix.

35. For ‘soma-enriched’ and ‘neurite-enriched’ mRNA species that are discussed and analyzed in the paper, what fraction of all those mRNA species code for synaptic proteins? Did the authors filter to only analyze the synaptic proteins? Please clarify.

36. In text, the authors refer to figure 3E when saying the following: “We could also confirm our third prediction that the protein copy number per spine is independent of the preference of mRNAs for soma or neurites”. Figure 3E refers to protein length in amino acids and has no mention of protein copy numbers. Please fix.

37. On page 5, where authors aim to “predict a single neuron’s total protein and mRNA copy number per transcript” and use the synthetic model proteins for this, I can see how you can aim to predict the relative localization preferences of mRNAs and proteins across the whole dendritic tree, but how do you work out the total numbers of mRNAs and proteins per cell? Do we know what determines the total numbers of any given mRNA or protein species per any given cell? Please expand.

38. Figure S3. The legend refers to “Blue, empty” bars but none are visible in the figure.

39. Figure S6. There is no scale for Y-axis.

Reviewer #3

(Remarks to the Author)

The manuscript by Bergmann et al. "How energy determines spatial localization and copy number of molecules in neurons," addresses important questions in the field concerning the parameters that affect the localization of RNAs and proteins in neuronal cells. It presents a computational framework to explain the spatial localization profiles and copy number distributions of mRNAs and proteins in neurons. The core proposition of the manuscript is that molecular species that generate high energy costs benefit more from dendritic mRNA localization. I found reading the manuscript enjoyable. The conclusion that RNA transport and local translation can be energetically more efficient than protein transport is insightful and holds significant importance for the field.

I have a few comments and questions:

1. What literature data are available on how many proteins are localized primarily through RNA transport and local translation compared to those localized primarily through protein transport? A summary would be useful.
2. In calculating transport costs, is it assumed that each molecule is transported individually? Recent studies suggest that mRNAs are localized as parts of larger granules with multiple molecules co-transported together. How would this affect the calculation of energy costs and the model?
3. Figure 2 demonstrates that longer proteins favor dendritic RNA localization (2E), but lower non-coding/coding RNA ratios favor somatic localization (2B). How were the energy costs for the data in 2B calculated? Is the difference in energy costs for different non-coding/coding ratios due to varying protein lengths (e.g., longer proteins lead to a lower non-coding/coding ratio) or other factors? How does the plot (2B) look when UTR length is analyzed instead of different non-coding/coding ratios?
4. The labeling of panels in Figure 3 does not match the text, and the panels are not presented in the same order as mentioned in the text.
5. There is more than a dozen datasets from the literature pertaining to RNA localization and stability, yet the manuscript uses only a few for model validation. Other datasets should be included or an explanation for their exclusion should be provided. Moreover, data from all relevant datasets should be applied to test multiple parameters, i.e., in different panels of Figure 3.
6. The authors discuss previous studies that show transcripts with shorter 3' UTRs tend to remain in the soma, while those with longer 3' UTRs are preferentially localized in dendrites, in relation to Figure 3B. However, Figure 3B shows mRNA length. What are the findings when 3' UTR length is considered?

Version 2:

Reviewer comments:

Reviewer #1

(Remarks to the Author)

I'm satisfied with the corrections. Nice paper!

(Remarks on code availability)

Reviewer #2

(Remarks to the Author)

We thank the authors for their careful and systematic responses to our questions and for the corrections to the manuscript. This is a very interesting and timely study.

(Remarks on code availability)

I cannot comment on this aspect.

Reviewer #3

(Remarks to the Author)

I thank the authors for addressing my comments. This is an important manuscript that could make better use of the available datasets. Please see my specific comments below.

2. In calculating transport costs, is it assumed that each molecule is transported individually? Recent studies suggest that mRNAs are localized as parts of larger granules with multiple molecules co-transported together. How would this affect the calculation of energy costs and the model?

Answer: In calculating transport costs, is it assumed that each molecule is transported individually? Recent studies suggest that mRNAs are localized as parts of larger granules with multiple molecules co-transported together. How would this affect the calculation of energy costs and the model?

Answer: Thanks for pointing this out. Based on our review of the existing literature we concluded that most mRNAs do indeed travel alone along microtubules (see section 'mRNA granule content' in the Supplement). Still, following the referee's suggestion, we wanted to check the effect of bigger granules on our main findings. We did so by assuming that always two mRNAs are trafficked together per granule and obtained similar results as shown in Figure 2 (Reply Figure 15). We here assume 2 mRNAs per granule, because, to our knowledge, bigger granules are observed only extremely sparsely.

I feel the manuscript presents a somewhat selective interpretation of published data and a narrow assumption that mRNA granules contain only one or at most two mRNA molecules, despite studies both supporting and opposing this view. The authors should at least discuss this possibility in the text.

3. Figure 2 demonstrates that longer proteins favor dendritic RNA localization (2E), but lower non-coding/coding RNA ratios favor somatic localization (2B). How were the energy costs for the data in 2B calculated? Is the difference in energy costs for different non-coding/coding ratios due to varying protein lengths (e.g., longer proteins lead to a lower non-coding/coding ratio) or other factors? How does the plot (2B) look when UTR length is analyzed instead of different non-coding/coding ratios?

Answer: Thanks for this comment. We will answer this comment in two ways. First, we follow the referee's suggestion and show a version of Figure 2 using different UTR lengths per transcript and not the non-coding/coding nucleotide ratio (Reply Figure 16). Second, we will argue that the non-coding/coding nucleotide ratio represents the same effect compared to adding UTRs of different lengths.

First, we re-simulated Figure 2 using UTR lengths of 100, 1000, 5000 (Reply Figure 16). This means that, for a given protein, we have three transcript lengths. Their lengths are given by three times the amino acid number (the coding sequence) plus the

UTR, which is either 100, 1000, or 5000 nucleotides long. Our findings (Figure 2) are preserved.

Second, we want to argue that our sampling approach based on the non-coding/coding nucleotide ratio is conceptually the same as sampling additive UTR lengths. For a given protein with a given number of amino acids, we sample three noncoding/coding nucleotide ratios. This gives three total transcript lengths for this protein, which contain the same coding region (three times the amino acid number) and three differently sized non-coding regions. The referee is correct insofar as bigger proteins have larger non-coding regions than small ones - but in Figure 2B we directly compare the localization of the three transcript lengths per underlying protein, so there should not be a distorting effect of the protein size itself. We acknowledge that this might not be clear at first glance, wherefore we visualized the results in Reply Figure 16 below.

The new figure suggests that mRNAs with longer half-lives (16A) also have longer UTRs (16B). Is that actually the case? Which dataset(s) were used to generate this figure? All prior research in this field has shown the opposite: transcripts with longer 3' UTRs tend to contain more destabilizing elements and are generally less stable.

4. The labeling of panels in Figure 3 does not match the text, and the panels are not presented in the same order as mentioned in the text.

5. There is more than a dozen datasets from the literature pertaining to RNA localization and stability, yet the manuscript uses only a few for model validation. Other datasets should be included or an explanation for their exclusion should be provided. Moreover, data from all relevant datasets should be applied to test multiple parameters, i.e., in different panels of Figure 3.

Answer: Thank you for this comment. In selecting the databases and matching proteins and mRNAs across databases, we have indeed evaluated multiple database pairings whereby some database pairings had much larger overlaps than others. In

selecting databases for Figure 3 we gave preference to the databases which met two criteria: 1) had a large number of entries

and 2) had the largest overlap with other databases, e.g., a database reporting half-lives vs a database reporting soma/dendrite

localisation ratios needed to have a large overlap in molecular species. A summary of the datasets we considered can be found in Reply Table 4. We have cross-matched genes from datasets reporting one of the parameters of interest with datasets

that reported other parameters of interest for the same gene (e.g., mRNA neurite-to-soma enrichment scores vs half-lives). To

maintain consistency, we have used the same dataset for analysis whenever possible and gave preference to dataset combinations

with the highest overlap. Shown in grey in Reply Table 4 are the cross-matched datasets we selected for Figure 3 based on the database overlap and total sample size.

I thank the authors for their explanation. First, this explanation should be included in the manuscript. Second, some of the presented datasets show equal or almost equal overlap with multiple other datasets. In such cases, the authors should conduct an analysis of these multiple overlaps. Drawing conclusions based on multiple datasets will strengthen the reliability of those conclusions. Third, the list of datasets available for analysis in Table 4 is incomplete.

6. The authors discuss previous studies that show transcripts with shorter 3' UTRs tend to remain in the soma, while those with longer 3' UTRs are preferentially localized in dendrites, in relation to Figure 3B. However, Figure 3B shows mRNA length. What are the findings when 3' UTR length is considered?

Answer: We thank the referee for mentioning this point. In case the referee is interested in the model predictions when considering 3'-UTR length instead of mRNA length, we kindly refer to our answer to comment 3.3. Here, we want to comment on the experimental findings concerning the localization of transcripts with different 3'-UTR lengths. To clarify that our statements on 3'-UTRs are indeed valid, we want to point to the findings of Erin Schuman's lab on the localization of transcripts in neurons. Tushev et al., 2018¹⁰ showed that transcripts localizing in dendrites are generally longer than those in the soma (Reply Figure 17A), and, when comparing transcripts for a given gene (which is equivalent to keeping the coding sequence constant and varying the UTR length), the transcript lengths in the dendrite are longer than those in the soma (see Reply Figure 17B). In addition, similar results have been independently obtained by 100 for BDNF. To highlight this in our manuscript, we added the following lines to the 'Discussion' section: "Interestingly, this model prediction can be observed not only at the population level (Figures 2, 3) but also in mouse hippocampal neurons within the mRNAs coding for the same protein. An et al., 2008¹⁰⁰ reported that the mRNAs of the prominent synaptic protein BDNF preferentially localise to the soma if they have a short 3'-UTR sequence. In contrast, BDNF mRNAs with long 3'UTR sequences preferentially localise in dendrites, confirming our model predictions at the level of individual protein species. This finding has been further corroborated across genes¹⁰."

The authors are working with multiple RNA localization datasets, and for each of these, the length of 3' UTRs is an easily calculable parameter. The analysis should not be restricted to just one study.

(Remarks on code availability)

Version 3:

Reviewer comments:

Reviewer #3

(Remarks to the Author)

The dataset usage remains limited, and the authors combine data from different models in a suboptimal way. Despite the availability of over a dozen recent neuronal RNA localization datasets with high coverage, the manuscript relies on only three (Table S7: Zappulo 2017, Glock 2021, and Tushev 2018). Additionally, data from different models are combined. For example, the authors use mRNA localization scores from mESC-derived neurons but pair them with mRNA half-lives from rat hippocampal neurons. This limits the overall conclusions of the paper. The authors should address the limited dataset usage and its impact on their model in the manuscript.

The question of how 3'-UTR length affects localization is also presented in a limited manner. The figures show mRNA lengths for somata-enriched and neurite-enriched transcripts, but the text attributes the difference in mRNA length to variations in the length of the 3'-UTRs, implying this is a general trend. However, the authors do not provide an analysis of multiple datasets to support this claim, such as distributions of 3'-UTR lengths for somata-enriched and neurite-enriched transcripts within individual localization datasets, which is essential to back up the authors' claim. A reference to the conclusions of the Tushev et al. manuscript is insufficient, as it is a literature reference, not a finding of the current study, and it represents a single study.

(Remarks on code availability)

Version 4:

Reviewer comments:

Reviewer #4

(Remarks to the Author)

(Remarks on code availability)

Dear Reviewers,

thank you for forwarding us the many encouraging and helpful comments which helped us improve the accessibility of our work. Below we present a point-by-point reply. In *italics* are the comments of the referees followed by our respective answers.

1 Reviewer #1

This paper explores whether energy arguments can explain why some neural proteins are locally synthesized whereas others are somatically synthesized. It concludes that energy might be able to explain a range of phenomena. The insight is novel and might interest a wide range of cellular neuroscientists and computational modellers. Overall the study uses an original but well justified diffusion approximation approach and simulations.

Major comments

1. It is assumed that there as much motor movement towards as away from the soma. In other words, the whole effect of motor transport is to speed up the diffusion constant (as is shown in one of the figures, btw the relation plotted there should be easy to derive using jump moments). However, diffusive transport is highly inefficient for long distances and a small amount of net transport towards the synapse, to be included as a drift term in the diffusion eq, could make a big difference. So I think it is important to check that. Less important but perhaps good to point out explicitly is that when there is a concentration gradient, a net current will be present which might explain some experiments.

Answer: Thank you for suggesting investigating the role of anterograde transport bias more closely. To address this question, we employed a multi-step approach. We first conducted a comprehensive literature review to identify evidence for or against a net anterograde velocity bias. A summary of this literature research can be found in the Supplement, section “Symmetry of mRNA transport” and it can be summarized by the following sentence “A small anterograde transport bias has been reported, but its magnitude, temporal stability, and spatial homogeneity along the dendrite are now the topic of active research¹⁻⁵”. In the available literature, we could neither identify a specific net-velocity value nor decisive evidence for its existence. All velocity values we found were reported as part of the tug-of-war motor motion and the presence/absence of a net bias seemed to be a topic of open debate.

In the next step, we incorporated an anterograde bias and just needed to determine its amplitude. Since the large majority of literature on active transport focuses on active mRNA granule transport, we decided to include a net velocity in our mRNA model and explore its computational consequences. First, we considered three velocity values $\nu_m = 0.0001\mu\text{m}/\text{s}$, $\nu_m = 0.0001\mu\text{m}/\text{s}$ and $\nu_m = 0.0003\mu\text{m}/\text{s}$, all of them are consistent with the available literature on active transport. Next, we evaluated their computational consequences in the context of mRNA localisation. We discovered that values as small as $\nu_m = 0.001\mu\text{m}/\text{s}$ (as well as all bigger values) would lead to a build-up of mRNAs at dendritic tips that can quickly become quite large (see Reply Figure 1). Mathematically, this is due to the zero-flux boundary condition at the end of the dendrite that prevents particles from leaving the dendrite. Intuitively, one can consider particles being transported by an anterograde bias quickly toward the “wall” at the end where they accumulate. Next, we checked whether the presence of mRNA build-up at the dendritic tips is consistent with experimental reports of long-range mRNA distributions in the cortex. To this end, we studied the mRNA expression data from the Allen Institute Mouse Brain Atlas (<https://mouse.brain-map.org/>).

For our analysis, we selected three mRNA species that are known to be dendritic⁶ (CaMK2a, Dlg4, Eef2) and examined whether model-predicted distal build-up is observed. We found that none of the mRNA species we studied in the Allen atlas exhibited an accumulation of mRNAs at the dendritic tips in hippocampal CA1 pyramidal neurons or DG granule cells (see Reply Figure 2). Therefore, we concluded that if an anterograde bias and the resulting net velocity is present, its value could be at most $\nu_m \approx 0.0001\mu\text{m}/\text{s}$.

Next, we included this anterograde velocity in our model and simulated the energy budgets for all synthetic protein species in our parameter space as described in the manuscript, but now consider a small anterograde velocity. Note that we made sure that, across the whole parameter space, all spines reach a given filling rate $\phi = 0.95$. Interestingly, we found that the results we present in Figure 2 remain valid (Reply Figure 3) also in the presence of an anterograde bias. To highlight this finding, we have now included the Reply Figure 3 in the supplement and added the following sentence in the ‘Results’ section ‘Dendritic mRNA localisation minimises energy expenditure for specific classes of proteins’: “For completeness, let us note that the predictions in Figure 2 are valid also in the presence of an anterograde bias that results in a net active mRNA velocity

($\nu_m = 0.0001\mu m/s$), see Figure S15”.

As a next step, we investigated the possibility that a bias in active protein transport could alter our model predictions, we re-simulated our model with a net protein velocity. For proteins, much less is known about the transport modalities in dendrites. Therefore, we decided to use a much bigger net velocity for proteins than for mRNA to take this possibility into account. In the following, we use $\nu_p = 0.001\mu m/s$ (which is possibly higher than the biological range, just to be sure there is no systematic effect we are missing). Again we made sure that, across the whole parameter space, the least filled spine still receives $\phi = 0.95$ of its maximal protein need η_p . Our results (Reply Figure 4A) suggest that this is not the case. To be extra sure, we considered the possibility that multiple proteins are transported together in a granule. Doing so, we found that, for a granule size of 5 proteins, our predictions remain unchanged (Reply Figure 4B). As a side note, we observed that a lot of proteins have to be packed into each granule to make some proteins favour protein transport over mRNA transport (~ 90 in the absence of a protein transport bias, ~ 60 with a protein net velocity of $0.001\mu m$). To show these cross-checks, we replaced Figure S3 with Reply Figure 4 and added the following sentence to the ‘Results’ section: “For completeness, let us note that in the presence of an anterograde bias in active mRNA transport or protein transport results of Figure 2 remain valid (Figures S3, S15)”.

Reply Figure 1: Even a tiny anterograde bias in mRNA transport leads to build-ups at dendritic tips.

Adding a net drift velocity leads to an accumulation of mRNAs at the dendritic tips. For demonstration, we show here mRNA distribution corresponding to the following parameters: half-life 8h, mRNA passive diffusion constant $10^{-3}\mu m^2/s$, protein half-life 8d, protein passive diffusion constant $0.01\mu m^2/s$, maximal spine content (η_p) 1000. All other parameters are set to the default values used in our manuscript (mean values in supplementary table S6).

Reply Figure 2: Dendritic tips in CA1 and DG do not show mRNA build-ups. In situ hybridization data in a coronal section downloaded from Allen Institute's Mouse Brain Atlas for **A** CaMK2a (<https://mouse.brain-map.org/gene/show/12107>, slice 30/59) **B** Dlg4 (<https://mouse.brain-map.org/gene/show/13164>, slice 27/59) and **C** Eef2 (<https://mouse.brain-map.org/gene/show/13407>, slice 28/58). CaMK2a, Dlg4 and Eef2 mRNAs are reportedly highly abundant in dendrites⁶. Yellow boxes are shown with 10-fold magnification on the right side. The orientations of CA1 pyramidal cells (orange) and DG granule cells (blue) together with their dendrites are highlighted. The outer boundaries of the dendritic layers are denoted with arrows (stratum moleculare of the DG, blue; stratum oriens of the CA1, upper orange; stratum lacunosum-moleculare of the CA1, lower orange). No local increases of RNA at these layer boundaries are detectable.

Reply Figure 3: (added as Figure S15) Results in Figure 2 remain valid in the presence of an anterograde active transport bias (net forward mRNA velocity). We re-simulated the mRNA and protein distributions with their associated energetic costs assuming an anterograde net velocity of $0.0001 \mu\text{m/s}$ for dendritic mRNAs. All other parameters were chosen as in Figure 2 in the main text. This shows that the results shown in Figure 2 can be obtained equivalently in the presence of a net anterograde velocity in the active mRNA transport.

Reply Figure 4: (new version of Figure S3) Protein transport in small granules is energy efficient only as an ‘add-on’ to mRNA transport and solely in long dendrites. For each synthetic protein species within our parameter space we computed the total cost for four trafficking options: 1) somatic mRNA with no active transport at all (red, empty), 2) somatic mRNA with protein transport in absence of mRNA transport (blue, empty), 3) dendritic mRNA with mRNA transport but no protein transport (red, filled), and 4) dendritic mRNA with mRNA and protein transport (violet, filled). In dendrites of increasing length, we then computed which of the four possible transport schemes was energetically optimal for each synthetic protein species. In **A** we show the summary distribution per dendrite length. Somatic mRNA with protein transport but no mRNA transport was never optimal, therefore we omit the associated bar (blue, empty). To evaluate the effect of anterogradely biased protein transport, we added a net velocity of $\nu_p = 0.001\mu\text{m}/\text{s}$ to our simulations and found that the preference for strategies remains largely unchanged (right bar). In **B**, we show that transporting more proteins in each transport-competent granule (5 proteins/granule) reduces the transport cost but leaves our model predictions largely the same. For completeness, let us note that when increasing the number of proteins per granule (to $\approx 60\text{-}90$) and growing the net anterograde protein transport bias such that the net protein velocity is $\geq \nu_p = 0.001\mu\text{m}/\text{s}$, an additional (4.) optimal strategy can occur where protein transport in the presence of somatic mRNA is favoured by a very small fraction of proteins.

2. Also the 95% percent protein requirement might play a critical role. Re-running for a few other values would be useful.

Answer: Thank you for this suggestion. We now evaluated our model predictions for varying spine filling rates. We found that protein requirements (in our nomenclature ϕ) of $\phi = 0.85$ and $\phi = 0.7$ instead of the $\phi = 0.95$, which we use throughout our manuscript lead to the same results for the energy efficiency of somatic vs dendritic mRNA, see Reply Figure 5 (added as Figure S16) below. To indicate this we have added the following sentence to the results section “Dendritic mRNA localisation minimises energy expenditure for specific classes of proteins” when discussing Figure 2 “For completeness, let us mention that results in Figure 2 hold across different spine filling rates ($\phi = 0.7 - 0.95$).”

Reply Figure 5: (added as Figure S16) Variations of the required spine supply rate ϕ do not affect our main results shown in Figure 2. We simulated the mRNA and protein distributions with their associated energetic costs assuming a spine supply rate ϕ of 0.85 (A1-E1) or 0.7 (A2-E2) instead of 0.95 which we used throughout the manuscript. Other than that, all parameters were chosen identically to Figure 2 in the main text. These results show that varying ϕ does not affect the results we show in Figure 2 in our manuscript.

3. *Presentation could be sharper. The maths is actually straightforward but the informed reader will need to dig through a lot of supplement to find what is done. It could be much simpler : “We solve the steady state diff eq, subject to $m(0)=...$, etc”. The manuscript mentions 8 phenomena and 7 parameters. A table/ list specifying them in the main text would help.*

Answer: Thank you for the suggestion to condense the model dynamics and its boundary conditions after the introduction of the core dynamics in Eq. 1. We now added the following sentences in ‘Results’ section to make clear that we are aiming for the dynamical equilibrium of the dynamics while meeting certain boundary conditions: “Here, ∂_t and ∂_x denote the derivatives in time and space, respectively. Next, we obtained the dynamical equilibrium for mRNA $m(x)$ and protein $p(x)$ distributions considering a closed dendrite of length L ($dm/dx(L) = 0$, $dp/dx(L) = 0$), a somatic influx of mRNAs and protein synthesis taking place at a site x in proportion to $m(x)$, both at the soma and the dendrite. The total amount of mRNAs and proteins is set such that the minimal copy number of $\phi \cdot \eta_p$ at each synapse is met, here ϕ is the filling ratio and η_p maximal spine capacity for proteins, for more mathematical details see supplemental section ‘Obtaining spatial distributions of mRNAs and proteins.’” To help the mathematically interested readers understand the model, we have also released the code underlying our results, which allows to have a quick overview of the parameters and algorithms we used.

Following the suggestion of the referee we have now added a table shown here as Reply Table 1 (Table S6) summarizing the parameter choices we used to generate the key model predictions shown in Figure 2 of our manuscript. Due to the space limits in the article, we could not add it to the main manuscript but instead provide a brief discussion of each parameter and its value in text form on page 3 of the manuscript. To balance the biological clarity needed for our biological readers with the mathematical detail requested by our computational readers, we have placed the derivations of the active transport model, parameter sampling from experimental data etc. in the supplemental information, and present them in the main manuscript the core dynamical equations we simulate. Any details necessary to arrive at a particular parameter (e.g., active transport model) are in the supplement. We hope this format helps enhance the clarity of the presentation.

Parameters	Sampled in Figure 2	Citation
mRNA half-life [hours]	2, 8, 20	7–10
mRNA length [#Non-coding/coding nucleotides]	(3, 6, 15) * aa [0/3, 3/3, 12/3]	11–13
Protein copy number	10, 200, 5000	14
Protein half-life [days]	2, 8, 20	15–18
Protein length [amino acids]	100, 500, 2000	16
mRNA diffusion [$\mu\text{m}^2/\text{s}$]	10^{-4} , 10^{-3} , 10^{-2}	2,3,19–30
Protein diffusion [$\mu\text{m}^2/\text{s}$]	$10^{-2.5}$, $10^{-1.5}$, $10^{-0.5}$	Table S1

Reply Table 1: (added as Table S6) Sampled parameter space range considered for simulating our predictions in Figures 2, 4. See Figure S1 for parameter distribution.

4. *Fig 5 is presented as in line with the model, but I didn’t see how it is related to energy. I think it simply verifies the diffusion model (which is useful of course). The data might restrict the amount of drift mentioned above.*

Answer: Thank you for this comment, it is correct that energy is not the main focus of Figure 5 but since it verifies the protein dynamics time scales (how fast spines are filled), we thought it is helpful to cross-check the model predictions also from this angle. In the main Figures 1-4, our model predictions are contrasted with ‘global’ variables like mRNA preferences or abundances, wherefore we felt it’s necessary to show that our model captures dynamics at lower scales (diffusion, spine-dendrite coupling etc.), too. To clarify that the findings presented in Figure 5 are not in direct connection to energy, we now write in the last paragraph of the ‘Results’ section: “Therefore, we wanted to test the dynamic aspects of our model and chose a paradigm focusing on the spine integration of somatically synthesised Shank3 proteins at varying distances from the soma³¹. Energy consumption is not directly assessed in this paradigm; yet, it is a valuable test for the underlying dynamics used in

our model, e.g., the spine-dendrite coupling, which are fundamental for the energy calculations presented in the previous paragraphs. The availability of quantitative measurements of photoactivated Shank3 proteins [...].”

5. *The discussion mentions that the total energy for protein synthesis and transport is comparable the spiking cost. This is an important claim and would be worth mentioning in the abstract.*

Answer: Following the suggestion of the referee we now mention it in the abstract. The new sentence reads: “Surprisingly, our results show that the energy for molecular turnover is a significant cellular expense, en par with spiking cost, and which requires energy-saving strategies.”.

Minor comments

1. *Data analysis (Fig 2) is a bit odd with this middle bar. Why not simply split the simulation data in two?*

Answer: To cover the whole range of parameter values, we sample three values for each parameter considered in Figure 2. These values are drawn from the distributions in Figure S1. We chose to sample 3 values to cover the low, medium and high ranges of each parameter and keep the number of drawn parameter combinations as low as possible. In addition, we can nicely reproduce the overall distribution of mRNA and protein levels in neurons (Figure 4). For visualization in Figure 2, we chose to follow the obvious split into 3 categories following the 3 values per parameter. To make trends more easily accessible, we decided to show how the preference for somatic vs dendritic mRNA localization changes for the low and high parameter values compared to the middle one. An equal split in two parts is not possible with our sampling strategy.

2. *Is there evidence that protein synthesis wears out the mrna, speeding up degradation? If so, that would be worth mentioning and perhaps including.*

Answer: Some studies have shown that translation results in the destabilization of mRNAs (e.g.³²). However, other works indicate that loading mRNAs with ribosomes protects them, by blocking the access of degradation factors³³. Therefore, we feel that this question cannot yet be thoroughly answered, based on the existing literature.

3. *p2 per mRNA and second => per mRNA per second*

Answer: Thanks for noticing, we changed it accordingly.

4. *p2 what is a 'cargo'?*

Answer: Thanks for pointing out that we never clearly defined this term. In the main text, we now write “[...] the ratio of transported mRNAs and proteins, we mathematically [...]” instead of “[...] the ratio of transported cargoes, we mathematically [...]” and “Finally, transporting one cargo, i.e, one mRNA granule or protein vesicle, along the dendrite [...]” instead of “Finally, transporting one cargo along the dendrite [...]”. We also added a definition in the first paragraph of the Supplement “Capturing active mRNA transport in our model” by extending “These transport-competent granules can bind to kinesin and dynein motors, which move along microtubules and carry the attached cargo.” with “[...], which can be, f.e, organelles, mRNA transport granules or protein vesicles. Here, we focus on mRNA granules and protein vesicles.”

5. *p2 the prominent => a prominent*

Answer: We changed this accordingly.

2 Reviewer #2 (Remarks to the Author)

Neurons are cells with extensive branched processes which form synaptic connections. The proteins within synapses determine their function and it is known that synapses are molecularly complex and diverse. It is therefore of interest to understand the basic cellular mechanisms that regulate the spatial distribution of the mRNAs and synaptic proteins in the dendrites and axons of neurons. Numerous studies (cited by the authors) have isolated mRNAs and proteins from either the soma or dendritic fraction and shown differential molecular compositions, lifetimes and other parameters. In the current paper the authors address the issue of energy constraints on the spatial distribution of mRNAs and proteins in neurons using a computational modelling approach. The authors then draw up on previously published datasets for use in their models and for validation. In general, I like this paper as it addresses the role of energy metabolism and generates interesting hypotheses that at some future time could be tested.

Issues that need to be addressed:

1. *A major claim of this paper is “We found that energy minimisation principles are able to explain experimentally reported exponential distributions of mRNA and protein copy numbers and their spatial profiles”. This is an overstatement that needs to be fixed. The “exponential distributions” the authors present are shown as values plotted as a function of distance from the soma (see for example, Figs 1, 4, 5). What these figures have in common is that they present gradients from soma to distal dendrite of molecular features. The authors repeatedly claim to have validated these gradients and refer to publication numbers 3, 13-18. I have read these papers and in none of them do I find any quantification of mRNAs or proteins along dendrites. These papers all contain data from either soma or dendrites, not gradients. So how is it possible to use these data to claim that the predicted gradients have been validated. To remedy this situation the authors can take one of two courses of action: (i) make it very clear in the text that they have not validated their predicted gradients, (ii) use datasets from the published literature that show real gradients of molecular distributions along dendrites.*

Answer: Let us clarify what we mean by the exponential distributions and modify our text accordingly. We thank the referee for pointing out that the term ‘exponential’ can be highly misleading in the way we used it. Here, we did not want to make any claims about spatial distributions being exponential, we were rather referring to the exponential profile of the mRNA and protein rank distribution (a non-spatial quantity) as shown, f.e., in Figure 4. To avoid confusion we now write “We found that energy minimisation principles explain the experimentally-reported exponential rank distributions of mRNA and protein copy numbers.” This refers to Figure 4A-B. As an example, we show below Reply Figure 6 with a superimposed exponential distribution; here, ‘exponential’ refers to the rank distribution of copy numbers. To stress this point, we adapted the x- and y-axis labels and the captions of Figure 4 (Reply Figure 7). Following the advice of the referee, we now clarified in the manuscript that our main focus is not spatial distributions reported in these papers but statistics on mRNA localisation and physical mRNA and protein properties (e.g., half-lives, length). In the ‘Introduction’ section, we state that “[...] we verified eight major predictions using simulations, our data and data from five further different proteomics and genomics screens conducted in neurons and an experimental measurement of spatiotemporal protein dynamics. We propose a computational framework to explain the spatial localisation profiles and copy number distributions of mRNAs and proteins in neurons.”

We hope that the improved wording in the abstract and the axes labels of Figure 4 make it clear that we are not claiming that all distributions we find are exponential and that the distributions shown in Figure 4 are not showing spatial profiles as a function of distance from soma, but rather copy number distributions.

Let us stress that, we nevertheless also validated the spatial profiles corresponding to our minimal energy solutions for two representative proteins using available experimental data from^{34,35}. For this comparison, we considered a prominent protein with a lot of dendritic mRNAs (CamKII α) and a predominantly somatically synthesized protein (GluA1). Considering the physical properties of these proteins (half-live, diffusion coefficient, amino acid chain length etc.), we predicted the spatial mRNA localisation profile for these proteins: our minimal energy solution favored dendritic mRNAs for the CamKII α parameters but somatic mRNA for GluA1 parameters. We then superimposed (this is not a fit) the predicted spatial CamKII mRNA and protein distributions onto the experimental data³⁴ (Figure 1C), and respectively for GluA1, in Figure S14³⁵ (Reply Figure 9). Here, it is important to note that these spatial distributions are not exponential in our model. The spatial profiles predicted by our model match quite precisely the experimentally reported data for these proteins^{34,35}.

Reply Figure 6: The experimentally reported mRNA copy number distribution in Zeisel et al., 2015³⁶ (left), and protein copy number distribution in Helm et al., 2021¹⁴ (right) can be described by an exponential function (dark blue line). This exponential function can be explained by our minimal energy model.

Reply Figure 7: (new version of Figure 4) Energy-based model predictions explain experimentally reported mRNA and protein copy numbers in neurons. **A** Predicted total mRNA (left) and protein (right) copy numbers across different dendrite lengths, colour-coded for length. mRNA and protein species are rank-sorted for abundance in each panel independently. Insets: Corresponding copy number statistics vs. dendritic length. **B** Validation of predicted abundances on a 250 μm long dendrite with experimental data. Simulated model results are shown alongside mRNA abundance data (left) published by ^{36,37} and protein abundance data (right) reported by us¹⁴. Experimental protein and mRNA data were sorted independently based on abundance. **C** Predicted mRNA (left) and protein (right) abundances for synthetic protein species grouped along the energetically preferred localisation pathway, either dendritic mRNA with mRNA transport (filled box) or somatic mRNA (empty box). Simulations were performed on a 250 μm long dendrite. **D** Validation of C with experimental data. mRNA (left) and protein (right) abundance data were extracted from ³⁸ and labelled as neurite-enriched (filled boxes) or somata-enriched (empty boxes) based on their mRNA enrichment (see Methods). Synthetic protein species use their respective energetically cheapest localisation pathway in A and B. Boxplots indicate median, quartiles and 1.5x interquartile ranges. ****($p < 0.0001$), two-sided Wilcoxon ranksum test.

2. *It would help if the authors could refer to and describe manuscripts where dendritic gradients have been measured at the single-synapse resolution level and note whether they are exponential or not. Furthermore, it is my understanding that some molecular gradients from soma to the distal dendrite can be either diminishing (as modelled in this paper) or increasing (e.g. AMPA receptors). How can their model accommodate gradients where the protein concentration increases as a function of distance from the soma? It is also the case that observed gradients of synapse proteins do not follow an exponential and have more complex distributions, such as input-specific differences. How does the model deal with the observation that certain synaptic proteins are more homogeneously expressed in synapses across the dendrite while others have very heterogeneous patterns of expression?*

Answer: Thank you for the suggestion to discuss the following topics in more detail in our manuscript. First, the need to stress that the spatial protein profile along the dendrite is in general not a simple exponential but can feature much more diverse distributions. Second, we now better highlight that in our model protein and mRNA concentrations do not necessarily decay as a function of distance from soma, but can also follow more complex distributions, e.g. increase, decrease or exhibit maxima along the dendrite. In the following, we will reply to each of these sub-points and describe how we incorporated them into our revised manuscript.

First, let us comment on the spatial profile along the dendrite. We now clarified in the manuscript that the resulting dendritic profile of proteins in our model is in general not exponential. We now state this more clearly in the following sentence: “Depending on parameters, the mRNA and protein distributions can cover a broad functional range (Fonkeu et al., 2019³⁴), for example, they can increase, decrease, exhibit local extrema or be close to constant across the dendritic space and are thus able to capture the variety of experimentally reported molecular distributions, see Figure 1C and Figure S14.” in the ‘Results’ section. For example, in Figure 1 we show the spatial profile of the CamKII α protein and its mRNA. We found that the functional form of the protein profile can be, for example, a double exponential distribution while the mRNA profile is a single exponential, see Figure 1C showing the overlap between theory and experimental CamKII protein data. To verify that dendritic mRNA profiles can be modelled as exponentially decaying from the soma (because the soma is the only source of mRNAs) we superimposed the theoretically predicted CamKII α mRNA profile on the experimental data in Figure 1C and we found them to be in good agreement.

Second, let us comment on the spatial profile of the protein distributions and, more specifically, on protein profiles increasing with distance from soma. In the simulations presented in our manuscript so far, protein levels were generally decreasing with distance. Our model can, however, reproduce protein profiles that are increasing towards the dendritic tips. One way to implement this is by assuming an increasing spine copy number with distance from the soma. In Reply Figure 8A-E we show that our main predictions (Figure 2) hold in this case too. An example of an increasing protein distribution is shown in Reply Figure 8F. For completeness, let us note that our main results (Figure 2) also hold with a decreasing spine protein copy numbers and we obtained results equivalent to Figure 2 also in this case. This illustrates how our model is capable of representing more complex protein profiles along dendrites and that our main predictions (Figure 2) hold for these, too.

Inspired by the comment of the reviewer concerning the increasing AMPA densities, we studied the distribution across the dendrite of the GluA1 subunit, one of the two dominant AMPA subunits and found that, considering the half-live, diffusion coefficient and other parameters of GluA1-containing AMPA receptors, the experimentally measured distribution of GluA1 across the dendrite matches with our theoretically predicted distribution (see Reply Figure 9, included as Figure S14). This suggests that the experimentally observed increase in AMPA currents which is mentioned by the referee and discussed for example in³⁹ cannot be explained by the GluA1-containing AMPA receptors, but may continue to be a topic of current research for the next years. Since GluA1-containing AMPA channels, both homomers and heteromers, are some of the more abundant AMPA channels, it is currently an open question how the increasing AMPA current profile along the dendrite could emerge: 1) it could be due to a superposition of different AMPA subtypes⁴⁰, 2) but it is equally possible that an increase of the individual channel efficacy underlies the observed increase of EPSC size with distance and not an increase in the copy number of AMPA receptors. In summary, our model can cover both decreasing and increasing spatial protein profiles along dendrites and reproduces the observed spatial distributions of a primarily somatically (GluA1) and a highly dendritically synthesized protein (CaMK2A). We also want to mention here that, in line with experimental evidence that GluA1 mRNA is highly somatic, our model predicts that somatic mRNA localization is energetically beneficial for GluA1.

Reply Figure 8: (version of Figure 2 with increasing spine size with distance from soma) For panels A-E we ran the same simulations as in Figure 2 under the assumption that the maximal spine copy number of a protein rises from 80% of the average at the soma to 120% at the dendritic tip at 1000 μm . Note that we still require all spines to be filled to 95% of their maximal capacity, which now varies along the dendrite (the filling rate does not). In panel F, we show an exemplary spatial distribution that increases with distance from the soma due to an increase of maximal protein per spine from 80% of the average at the soma to 120% at the dendritic tip.

Reply Figure 9: (added as Figure S14) Our model reproduces GluA1 and CaMK2 α distributions with distance from soma when provided with realistic parameters from literature (no fits). Predicted dendritic distribution of GluA1 protein (left), and CamKII α protein (right) compared to experimental data^{34,35}. The dendritic distribution profile of GluA1 (left) was captured from cultured mouse hippocampal neurons. The simulated dendritic protein profile was obtained on a 500 μm long dendrite and normalised to the integrated density within the first 70 μm to allow comparison with the data that are limited to this range. The predicted protein profile matches the experimentally reported density with distance from the soma³⁵. Here, we used the following GluA1 parameters: protein half-life 8.46 days¹⁶, mRNA half-life 21.45 hours¹⁰, amino acids per protein 907, non-coding nucleotides per transcript 2400¹⁰, average copy number per spine 279.5¹⁴ and diffusion constant 0.05 $\mu\text{m}^2/\text{s}$ ⁴¹. The mRNA of GluA1 is assumed to be primarily somatic, following the literature, e.g., ref.¹⁰. Interestingly, our model also predicts somatic mRNA localization to be energetically beneficial for GluA1. On the right, we show the experimentally obtained CamKII protein distribution³⁴ together with our corresponding model prediction from Figure 1C for comparison.

3. *Relevant to the above points, in Figure 5 the title reads “Simulations reproduce experimentally observed etc”. Figure 5A-C have no experimental data for support, but Figure 5E does show experimental data for comparison. But Figure 5E only shows fluorescence increase within 4 spines over time and not “spatiotemporal spread of proteins in dendrites” that the title of the figure states. It would be desirable to have experimental data accompanying Figure 5A and B to convince the reader that their model indeed reproduces protein distributions in dendrites and spines, and to show the comparison between the model and the experimental data.*

Answer: Thank you for suggesting to improve Figure 5 and its discussion in the results section. We followed the referee’s recommendation and modified the title to: “Predicted model dynamics reproduces experimentally observed spread of somatic proteins in space and time” and modified the corresponding figure (see Reply Figure 10) to highlight the spatial aspect of these data sets. Since our energy optimal distributions are the result of molecular dynamics, we wanted to verify aspects of the temporal dynamics directly and cross-check whether our estimates of the synaptic uptake speed from the dendrite and the movement of proteins along the dendrite match experimental data. To this end, we studied the data by ³¹ which contains both spatial and temporal dynamics and we now show the measured and predicted dynamics as a function of space (Figure 4A-C) and time (Figure 4D-F) next to each other. Tsuruel et al., 2006³¹ labelled only somatic proteins, therefore any changes in fluorescence in single spines are due to the spread of somatic Shank3 through the dendrites and spines with time (and not due to local synthesis). Tracking the fluorescence of spines at four different distances from the soma allowed us to validate if our model correctly simulates the time course of spine fluorescence given Shank3 parameters from the literature. To address this referee comment we replaced panels A and B of Figure 5 with 3 new panels (Reply Figure 10) showing the spatial distributions of Shank3. These highlight the spatial spread of somatically labelled Shank3 proteins across dendritic spines at various distances from the soma in our simulations (new panel A) and in the data from Tsuruel et al., 2006³¹, (new panel B) and a combination of both (new panel C). We adjusted the description and the figure references in the main text accordingly.

Reply Figure 10: (new version of Figure 5) Simulations reproduce experimentally observed spatiotemporal spread of proteins. **A** Simulated spatial spread of somatically photoactivated Shank3 proteins in dendritic spines along a linear dendrite after 0.5, 3, 7h. Proteins are continuously photoactivated at the soma. The first 150 μm of the dendrite are shown and the spatial locations of four spines of interest (25, 50, 75, 105 μm distance from soma) are highlighted in green. **B** Experimentally observed increase in fluorescence after 0.5, 3, 7h at the imaged spines (on average 25, 50, 75, 105 μm distance from soma) as reported from ³¹. **C** Overlay of predicted (A) and experimentally observed (B) spatial spread of fluorescence across spines. Data were normalised with those of the proximal-most spine to allow comparison of simulated and experimentally observed data. **D** Predicted temporal integration of photoactivated Shank3 in the four spines of interest. **E** Experimentally observed time course of fluorescence during the whole imaging period of 8h at the imaged spines as reported from ³¹. **F** Overlay of predicted (D) and experimentally observed (E) time course of fluorescence in spines of interest. Data were normalised with those of the proximal-most spine to allow comparison of simulated and experimentally observed data.

4. The authors use data from a 100 μm segment of a dendrite to model distributions up to 1000 μm (Figure 1C). The authors should make it clear that this is the assumption behind their model and note that dendrites have more complex gradients and distributions that will not be accommodated by their model.

Answer: Thank you for the suggestion to clarify the spatial range of the available protein data in comparison to the spatial range of the model prediction. Our model does not have any built-in length limitations and can make predictions for both short and long dendrites. However, to be able to make predictions that are experimentally testable we needed first to evaluate the dendritic lengths in studies providing parameter values for our model such as mRNA soma-to-dendrite ratios, half-lives etc. Therefore, we are showing in Figure S11 the dendritic length distributions in the studies we considered works by ⁴²⁻⁴⁵. Similarly, the experimentally measured CamKII profile is limited to 100 μm for technical reasons (field of view vs resolution trade-off). In Figure 1C, we simulated a 500 μm long dendrite (previously in the insets) and showed the first 100 μm thereof (i.e., where we have data to compare) in the main panel. However, to avoid confusion we have now removed these insets. Following the referee's suggestion, we have also added the following sentence to discuss more broadly possible shapes of protein distributions along the dendrite, and how model predictions may look for different parameter settings: "For completeness, let us note that, in general, the dendritic proteins may exhibit more complex spatial gradients and distributions than those of CamKII α in Figure 1C and GluA1 in the Figure S14. As more data becomes available quantifying the long-range spatial profile of protein distributions, our model assumptions and model predictions about the synaptic and dendritic copy number profiles may need to be generalized."

5. The authors state: "Functionally, keeping the protein copy number in synapses within a specific range for each molecular species is essential (Wang et al)". Therefore, we required in our model that all spines be filled to $\geq 95\%$ of their capacity for a protein of interest". Reading Wang et al I could not find reference to the statement about specific ranges being essential. It also doesn't make sense because labelling of synaptic proteins using immunolabelling typically shows considerable heterogeneity (as shown in papers cited in this manuscript including those from Dr Rizzoli). Furthermore, what does it mean to say that spines are filled to 95% capacity? Does that mean any one protein occupies 95% and other proteins the remaining 5%?

Answer: Thank you for the suggestion to clarify what we mean by the parameter filling rate (η_p) and the suggestion to replace the Wang et al citation with more specific statements. First, let us note that in our implementation of synaptic copy numbers, we followed the convention established by other models, e.g. Triesch and Hafner et al., 2018⁴⁶. These models build on the observation that synaptic receptors are present in specific copy numbers. Our prior work (including Helm et al., 2021¹⁴) has revealed that synaptic copy numbers for a given protein of interest can be characterized by a mean value and variability around this mean, for example 1082.8 ± 98.5 for GluA2, or 255.8 ± 23.6 for PSD93, see Supplementary Table of Helm et al., 2021¹⁴. Let us mention here that the protein copy variability of mature mushroom spines, compared to the more immature stubby spines, was particularly low (Helm et al., 2021¹⁴). To include this in the model, we considered a target number of proteins per spine and required that at least 95% of that target number needed to be present in each synapse and that no synapse contained more than η_p copies of this protein (upper bound). In other words, synaptic copy numbers across all synapses were in the range $[0.95\eta_p, \eta]$. We also varied the range to be $[0.7\eta_p, \eta_p]$ or $[0.85\eta_p, \eta_p]$ and obtained equivalent results for Figure 2, see reply Figure 5. Following the referee's suggestion, we now write: "To keep the synaptic protein copy number in our model around the experimentally reported mean values, we set the copy numbers to be within the range $[0.95\eta_p, \eta_p]$ at each synapse, η_p sampled from the range of synaptic copy numbers by ¹⁴. When we varied the copy number range to be $[0.7\eta_p, \eta_p]$ or $[0.85\eta_p, \eta_p]$ we obtained equivalent results, see Figure S16."

6. The authors use protein abundance data from Figure 4C of Helm et al for the copy number of proteins in individual dendritic spines and in particular the copy of number for CamKII of 32,336. The same figure notes that PSD95 copy number is 1,000 and NMDA GluN2B subunit is several hundred. It is widely thought, based on electrophysiological data, that there is 10 NMDARs/synapses which is about 20 copies of GluN2B (see work of Roger Nicoll). Similarly, the estimates for PSD95 based on MS and super-resolution studies is about 10-fold lower than those in Helm. Indeed, in Helm et al Extended Data Figure 8 the graph indicates their estimated copy number is up to 10-100 fold higher than in other studies. It should also be noted that Helm et al did not make direct measurements of copy number per synapse. Furthermore, they reported only two types of spines and not the whole range of spines (which will have implications for the computational modelling). While the current manuscript is not one that might settle these differences in the datasets used for the model, the authors should present results using these lower numbers alongside their existing models based on Helm et al., 2021.

Answer: Thanks for the interesting comment. Let us answer this comment from a computational and an experimental di-

reaction. Concerning the influence of absolute copy numbers per spine on the model predictions we have found that model predictions are fully invariant to the copy numbers in the spine (see Figures 2C and 3C) therefore the theoretical predictions will be identical for those lower copy numbers per spine. We highlight that protein copy numbers per spine are not crucial for the somatic vs. dendritic localisation decision (Figure 2C) and provide an intuitive explanation in the following sentence: “The reason is that the energy budget for each protein species depends critically on the *relative* energy expenditures between translation, transcription, and active transport, and evaluates whether shifting between categories makes sense energetically. However, the value of the total energy budget (which is shaped by the synaptic numbers) is not essential for determining the optimal solution. Therefore, two proteins with the same model parameters differing only in their spine count will have the same energy optimal localisation strategy.”

On the experimental side let us first note that the numbers reported by Helm et al., 2021¹⁴ (Table S2) for PSD95 per spine are 550 ± 80 (mushroom) and 539 ± 78 (stubby); per PSD, the copy numbers are 240 ± 35 and 261 ± 38 , respectively. The latter is in good accordance with the widely used value of 300 PSD95 molecules per PSD (for example, in ⁴⁷, which cites the ^{48–50} for this) and has also been used by MS studies like ⁵¹.

Second, the spine copy numbers for GluN2B are 319 ± 36 (mushroom) and 433 ± 49 (stubby); in PSD, the copy numbers are 152 ± 17 and 253 ± 29 , respectively (Helm et al., 2021¹⁴), implying about 75 channels per PSD for the mushroom spines, the ones that are most abundant in the brain. The MS study⁵¹ found about 35 per PSD, purified from brains (without taking into account the possibility of PSD damage during purification, which will lower the copy number estimates). Therefore, we conclude that in the cases the referee mentioned, the numbers presented by Helm et al., 2021¹⁴, do not differ strongly from the literature.

Third, we are mainly interested in the overall distribution of protein levels per spine, not specific protein identities (as presented in Ext. Data Figure 8 of Helm et al., 2021¹⁴). In combination with the previous arguments, we think that, for our purpose, Helm et al., 2021¹⁴, is a useful and reliable source for synaptic protein levels across the proteome.

Fourth, since our model is created to be applicable across multiple contexts, we think that having data across protein species for two of the most abundant spine types (mushroom and stubby) is an acceptable representation of the biological reality and not a restriction. In conclusion, we think that the data from Helm et al., 2021¹⁴, cover the range of protein levels per synapse to a good level that matches the needs of our modelling approach.

7. What type of neuron (excitatory/inhibitory, pyramidal/purkinje/other neuron type) is being used as a model for this study as the cells have largely different morphologies, spine densities, number of main branches stemming from the soma and all of these would significantly influence the derived mRNA and protein distribution calculations.

Answer: Thank you for the suggestion to clarify the neuronal type that served as an example for our model. Most of the available data on molecular parameters entering our model came from hippocampal neuronal cell cultures (^{10,14,37}, Table S1). For example, mRNA localisation scores and protein half-lives were measured in induced mouse neurons and brain cortex synaptosome (Table S2), and the dendritic lengths we considered in our model are inspired by the reports in cultured hippocampal pyramidal cells see Figure S11. Therefore, our parameter choices fit best with pyramidal cells in the hippocampus. To make this clear, we have added the following sentences to the ‘Discussion’ section “Let us note that the majority of the available data on molecular parameters entering our model came from hippocampal pyramidal cells. Since this is a commonly studied cell type in the context of molecular turnover, we drew on this available experimental literature to build our model and test its predictions. However, future datasets characterizing molecular turnover and trafficking across cell types could help generalize our model findings across the diversity of neural cell types”.

8. Please justify (in main text) why the passive diffusion-mediated molecule movement and microtubule-mediated active transport are reduced into a single variable. The authors should justify this by providing examples of experimental data that demonstrate the validity of this approach. Can the authors justify why they can treat microtubule-mediated active transport as random walk and refer to experimental data.

Answer: Thank you for the suggestion to add more discussion to the results section to justify why microtubule-mediated active transport could be reduced to a 1D diffusion. Let us clarify that we are not introducing this concept; we rely on prior seminal studies from the 1990’s showing that the “tug of war” motion of dynein and kinesin is pulling the cargo in different directions. Seminal work by Paul Bressloff and others has been able to show that the ‘tug of war’ motion can be mathematically described by a 1D diffusion; for a comprehensive discussion see work by ⁵². More recent experimental studies have shown that this motion is remarkably stable to intracellular perturbations (e.g., ATP concentration and roadblock density) and that these have no significant effect on this process⁵³.

9. The Methods section contains a single subheading “Experimental data analysis”. This only really refers to the datasets that were used and provides very little detailed methods information and certainly not enough to replicate the study. Some of this information is in the supplemental information section and should be in methods.

Answer: We thank the referee for this suggestion and followed it by adding a concise summary of the model to the Methods section that allows the informed reader to repeat our model simulations which are the core of our study. The comprehensive and detailed discussion of the parameters and the mathematical details are still covered in the Supplement. For the code with all its details on the model implementation and data analysis pipelines, we refer to the provided GitHub repository.

10. In results section (page 2) the authors refer to “using experimentally reported average run duration, run velocities, and the ratio of transported cargoes” to mathematically map the transport process to a diffusion process. Please create and, in main text, refer to a table outlining the references for the “experimentally reported” measurements, the values associated with each publication and whether the values refer to mRNA or protein species.

Answer: We follow the referee’s suggestion by creating a new Supplementary Table (shown below for convenience, Reply Table 2) that summarizes the parameters, their associated values and the references we considered for each. The relevant parameters are discussed in detail in the sections ‘Fraction of transported mRNA granules θ ’, ‘mRNA granule velocity v during the run phase’, and ‘Transport state exit rate β ’ in the Supplement. Here, we considered publications from almost three decades, facilitating different experimental paradigms and we sometimes derived values from observations made by the authors that we integrated with direct measurements. Therefore, these values have to be discussed in context. We think this is much easier to access for readers in plain text than in an extensively commented table, which would furthermore comprise more than 30 references. We clearly see the advantage of showing individual values in a table that are easier to compare directly, which we hence did for the protein diffusion coefficients in Table S1. To further improve the accessibility of the parameter discussions, we added references in the main text that refer to the specific sections of the Supplement.

mRNA transport model parameter	Value	References
Fraction of transported mRNA granules θ	10%	3, 4, 19, 26–28, 54–57
mRNA granule velocity v during the run phase	1 $\mu\text{m/s}$	2, 3, 19, 26, 54–64
Transport state exit rate β	1s	1, 3, 5, 30, 59, 64–66

Reply Table 2: (added as Table S5) Summary of the parameters used for the mRNA transport model. The references are discussed in context in the corresponding Supplementary Text sections of identical names. We use the same values for the protein transport parameters, for which we argue in the section ‘Capturing active protein transport in our model’ based on work by 67–72.

11. How different are the calculated transcription costs for the following scenarios: (a) most mRNA is soma-localised and protein is synthesized in soma and then transported to target synapses, (b) mRNA is preferentially localised in dendrites and most protein is synthesized locally. Please discuss in main text.

Answer: First, transcription costs are always lower if mRNAs are trafficked into the dendrite to perform local synthesis. In this case, fewer proteins are ‘lost’ on the way along the dendrite and, consequently, fewer mRNAs are required, leading to lower transcription and hence costs. This is exemplified in Figure 1B for CaMK2a and an overview of the transcription cost reduction across protein species is presented in Figure S5. In Figure S5A, we show that the median decrease in transcription costs across our parameter space is $\sim 15\%$, ranging from less than 1% to close to 100%, depending on the actual parameters of mRNA and protein.

We thank the referee for the suggestion and make the actual numbers more accessible to the reader by expanding the main text: “On the other hand, trafficking mRNAs from the soma into the dendrites reduces the spatial distances proteins have to cross, thereby lowering the cost of transcription and translation (the median reduction across our sampled parameters is $\sim 15\%$) because fewer mRNA and protein copies need to be produced, but longitudinal mRNA transport along dendritic microtubules will contribute to the energy budget (Figure S5A, B right).”

12. On page 3, the authors state that “active mRNA transport is more energy effective compared to active protein transport”.

In the ‘Transport cost’ section of the ‘Supplemental Information’ the authors do not refer to any differences or factors that influence the costs of mRNA vs protein trafficking differently. The supplementary figures S3 and S4 they refer to show stacked bar plots and box plots with no context and no intuitive way to help the reader understand how their statement is supported by data. In the same paragraph on page 3, it says that “trafficking mRNAs from the soma into the dendrites can reduce the cost of transcription and translation because fewer mRNAs and protein copies need to be produced”. Please explain where this statement is coming from and, if it aids the naïve reader, add an illustration.

Answer: This interesting question has multiple parts, which we will answer one by one. We will 1) comment on the modalities of mRNA and protein transport costs in the context of our model and the available literature; 2) reply to the comments made on Figures S3, S4, and how this relates to data; and 3) elaborate on the sentence “trafficking mRNAs from the soma into the dendrites can reduce the cost of transcription and translation because fewer mRNAs and protein copies need to be produced”.

First, let us comment on how we integrate mRNA and protein transport costs in our model. To this end, we use a 3-state model (a resting state, a retrograde transport state, and an anterograde transport state) to capture the active transport of mRNAs and proteins in our model. To compute the associated costs, we consider the number of steps molecular motors per time since experimental reports indicate that each step motors take costs ATPs^{73–76}. In our framework, the number of motor steps taken depends on the number of cargoes (i.e., mRNA or protein) and their transport speed. Based on the available literature reporting similar transport properties regardless of cargo (as discussed in the subsections of ‘Capturing active mRNA transport in our model’ and the section ‘Capturing active protein transport in our model’ in the Supplement), we take similar transport parameters for mRNAs and proteins. To make this more clear, we added the sentence to the Supplement “Therefore, we applied our mRNA transport model as described above to protein transport and used the same parameter set for both. The same applies to the associated transport costs.”

Some transport-competent granules could contain more than one mRNA (e.g., 2 mRNAs). When testing this scenario in our model we found that increasing the number of mRNAs per granule led to the same results (see Reply Figure 15). Furthermore, we also tested the scenario where anterograde protein transport bias was present or multiple proteins were trafficked via a single granule and in both scenarios found similar results (see Reply Figure 11). In summary, across all scenarios tested, we found that the energy optimal configurations favoring protein transport were present, but were a relative minority, were present primarily for very long dendrites (1000 μ m) and were most effective in combination with dendritic mRNA localisation.

Second, let us comment on Figures S3, S4. Here we need to mention that we replaced Figure S3 with an updated version which we show below (Reply Figure 11) to explore transport bias, particles per granule and other alternatives next to each other. Here, we show our model prediction on how many of our sampled synthetic protein species energetically prefer each of the possible localization strategies for various dendritic lengths, inspired by the range of experimentally measured dendrites (Figure S11). In short dendrites, somatic mRNA (empty red bar) and dendritic mRNA (filled red bar) are the only energy-optimal strategies. We found that protein transport, when considered without dendritic mRNA, is rarely energetically more beneficial than mRNA transport except for large transport-competent granules containing \sim 90 proteins that are trafficked along very long dendrites. Protein transport as an ‘add-on’ to mRNA transport is predicted to be energy-efficient mainly in long dendrites (purple bar). We followed the referee’s suggestion and expanded the caption of this figure.

In Figure S4, we now show a breakdown of our model energy distribution as a function of dendritic distance. Protein transport (bottom row) shifts the cost substantially further into the dendrite, much more energy is needed beyond 100 μ m from the soma. This now requires a higher local energy supply, rendering it energetically overall less favourable, given the decreasing mitochondria density observed by⁷⁷ (see also Figure 1F).

In summary, Figures S3, S4 highlight our prediction that protein transport is energetically much less efficient than mRNA transport in most parameter regimes. To make our arguments more accessible and specific, we added the following sentence to the ‘Discussion’ section: “Our model also offers an interesting perspective on active protein transport in dendrites. In comparison to mRNA transport, which has been researched quite extensively, reports on active protein trafficking in dendrites are rather sparse (e.g.,^{67–72}). In our modelling framework, mRNA transport is predicted to be almost always energetically more efficient than protein transport to maintain the baseline protein distribution across spines (Figures S3, S4), particularly in shorter dendrites. Future experiments targeting the population of actively trafficked proteins can shed light on the statistics of actively trafficked proteins in relation to the locally synthesized protein group.”

Third, let us comment on the sentence “trafficking mRNAs from the soma into the dendrites can reduce the cost of transcription and translation because fewer mRNAs and protein copies need to be produced”. If all proteins of a given species

are synthesized in the soma, they have to move along the dendritic shaft to be integrated into the dendritic spines. Ongoing degradation causes a substantial loss of proteins on the way. Local synthesis reduces this loss because the spatial distance the proteins have to cross is reduced. This advantage comes at the cost of transporting mRNAs along the dendrite. The decisive question is how the additional transport costs relate to the reduction in biosynthesis of mRNA and protein. The corresponding illustrations are Figure 1B for an exemplary calculation and Figure S5 for the analysis across species. To improve our explanations, we added the following line to the ‘Results’ section: “trafficking mRNAs from the soma into the dendrites reduces the spatial distances proteins have to cross, thereby lowering the cost of transcription and translation because fewer mRNAs and protein copies need to be produced”

Reply Figure 11: (new version of Figure S3) Protein transport in small granules is energy efficient only as an ‘add-on’ to mRNA transport and solely in long dendrites. For each synthetic protein species within our parameter space we computed the total cost for four trafficking options: 1) somatic mRNA with no active transport at all (red, empty), 2) somatic mRNA with protein transport in absence of mRNA transport (blue, empty), 3) dendritic mRNA with mRNA transport but no protein transport (red, filled), and 4) dendritic mRNA with mRNA and protein transport (violet, filled). In dendrites of increasing length, we then computed which of the four possible transport schemes was energetically optimal for each synthetic protein species. In **A** we show the summary distribution per dendrite length. Somatic mRNA with protein transport but no mRNA transport was never optimal, therefore we omit the associated bar (blue, empty). To evaluate the effect of anterogradely biased protein transport, we added a net velocity of $\nu_p = 0.001 \mu\text{m}/\text{s}$ to our simulations and found that the preference for strategies remains largely unchanged (right bar). In **B**, we show that transporting more proteins in each transport-competent granule (5 proteins/granule) reduces the transport cost but leaves our model predictions largely the same. For completeness, let us note that when increasing the number of proteins per granule (to $\approx 60\text{-}90$) and growing the net anterograde protein transport bias such that the net protein velocity is $\geq \nu_p = 0.001 \mu\text{m}/\text{s}$, an additional (4.) optimal strategy can occur where protein transport in the presence of somatic mRNA is favoured by a very small fraction of proteins.

13. The “cell’s general ‘infrastructure’ costs” (page 3). These costs will scale with the location in the cell (close or in the soma vs distal dendrites). This is especially crucial for protein translation-related infrastructure. A calculation of the average costs for such infrastructure in different parts of the cell should be estimated and then the workload these infrastructure components get could be used to split the costs per protein species and show how substantial the costs end up being. If they are minor, they can stay as a supplementary figure, if major, they should be included and integrated into the main model.

Answer: Thank you for suggesting clarifying the infrastructure cost and specifically commenting on the influence of ribosome localisation cost as the key element of the protein translation infrastructure. We understand the comment of the reviewer as follows: the localisation cost for translation-related infrastructure could be different in distal dendrites because the ribosomes would need to be trafficked to these locations before they could be employed. This would mean that the infrastructure cost could be different for somatically synthesized proteins vs dendritically synthesized proteins, a difference that would need to be potentially accounted for in the model. This is an interesting question that we have investigated as follows. The trafficking of ribosomes from the soma to the dendrite could follow two different scenarios. First, the ribosomes move by free diffusion in the cytosol. Second, the ribosomes would be transported along microtubular tracks (either together with mRNAs or separately). Interestingly, there is experimental evidence suggesting that ribosomal subunits are freely diffusing in the cytoplasm. For example, it has been argued that the recycling of ribosomes through Brownian diffusion in the cytosol plays an important role in the control or regulation of translation⁷⁸. Importantly, there is also evidence that, when ribosomes are transported actively with motor proteins along the microtubular tracks, they are often transported together with mRNA molecules⁷⁹. This implies that their transport does not incur a separate energy cost, since the energy necessary for mRNA transport is sufficient also for the progression of ribosomes to synapses. Overall, while much of the biology surrounding the assembly and trafficking of ribosomes is still to be understood, at this point it is not possible to determine the additional ribosome localisation cost that is incurred separately from mRNA transport. At any rate, current evidence points to the idea that the ribosome localisation cost is already contained in the mRNA active trafficking cost, and is further minimized by ribosome diffusion. We have added a sentence to the results section “We excluded these costs because they are already included in the cost estimates we use (e.g. co-transport of ribosomes and mRNA in a single granule does not incur additional cost⁷⁹) or can be excluded because they are not unique to the biosynthesis or transportation of a specific transcript but instead can be considered general ‘infrastructure’ costs of a cell.”

Reply Figure 12: Excerpt of Figure 2 of Dastidar et al., 2022⁷⁹ showing that ribosomes are transported ‘on top of’ the mRNA as it is actively transported.

14. For the statement “CaMKIIalpha proteins are not actively transported in dendrites”, I can’t find in the Shen et al, 1998 referenced paper any data supporting this statement. Shen only refers to CamKIIalpha capture and transfer from dendrite to the dendritic spine and not the trafficking from soma to and along the dendrites. Moreover, the reference to Hirokawa and Takemura, 2005 paper only speaks about CaMKIIalpha mRNA trafficking and not protein.

Answer: Thank you for suggesting replacing Shen et al., 1998, and Hirokawa and Takemura, 2005, with more direct experimental evidence that CamKIIa protein is not actively transported longitudinally along the dendrite. To this end, we have now replaced these citations with⁸⁰. This study employed single-molecule tracking via PALM of individual CamKIIa molecules

within a dendrite and has shown that their motion in 2D is consistent with a non-directed two-dimensional diffusion. Mathematically, a non-directed diffusive motion is characterized by a square root behaviour of the mean square displacement (MSD). Lu et al., 2014⁸⁰, have shown that in the dendrites of living neurons, CamKII proteins exhibit exclusively 2D diffusive motion that is characterized by a square root behaviour of the 2D MSD, which shows no traces of active transport along microtubules. Below is Figure 1 from Lu et al., 2014⁸⁰ (Reply Figure 13). The corresponding sentence in the ‘Results’ section now reads: “CamKII α proteins have been shown to perform a rapid, non-directed two-dimensional diffusive motion in the dendritic cytoplasm without signatures of active transport⁸⁰.”

Reply Figure 13: Figure 1 of Lu et al., 2014⁸⁰. Panel F reproduced here shows that CamKII protein shows the mean square displacement vs time that is consistent with a diffusive motion inside the dendrite. Panels H and N show the experimentally measured distribution of diffusion coefficients that are derived from following individually particles.

15. For the sentence “Remarkably, we found that the experimentally measured mitochondria distribution in dendrites is consistent with our model predictions.” Please clarify which exact predictions. To claim ‘consistency’ you would require support from a statistical test.

Answer: Our model allows us to compute the total cost to localize proteins as a function of distance from the soma. In Figure 1F, we show the predicted spatial distribution of the energy costs incurred in the dendrites (as a cumulative density, integrated from soma (position zero) to the distance of interest shown on the x-axis). Superimposed on the theory line, we show the experimentally measured cumulative density of dendritic mitochondria, which are a source of local ATP in dendrites⁷⁷. This figure shows that the theory and experiment overlap to a large degree. We appreciate your noticing that our statement is not precise enough and thus made it more quantitative.

Consequently, we now write “Remarkably, we found that the local energy need predicted by our model is within 1.4σ interval of the experimentally measured mitochondria distribution in hippocampal pyramidal dendrites of mice⁷⁷, see Figure 1F.”

16. In Figure 1F, the graph of energy needs indicates that beyond 200 μm there is very little need for energy, which is implausible. Please clarify.

Answer: This is a cumulative distribution indicating that the energy needed for protein turnover (translation and degradation) and mRNA transport diminishes as a function of dendritic distance from the soma. It does not imply that the energy need is zero beyond 200 μm , since it is a relative measure comparing the need close to the soma to that of more distal locations. Additionally, the theoretical measure considers only the energy demand of molecular turnover, while energy is needed also for many other functions, e.g., opening and closing of ion channels or operating chloride pumps⁸¹. We agree that it seems counter-intuitive to observe fewer mitochondria with distance even though the synaptic density stays constant. Therefore, it is particularly notable that the experimentally reported mitochondrial profile in the dendrite and our model simulations are consistent with each other (for a quantification see our reply to the previous comment). In the hope that more data can help with the intuition, we would like to point out that other experimental studies including⁸² (their Figure 2C, D) and⁸³ have also arrived at very similar mitochondrial distributions to the one we show in Figure 1F, and Chavan et al., 2015⁸³ even write in their abstract “Confocal imaging analysis on neuronal cultures revealed that most neuronal mitochondria are either somatic or distributed in the proximal part of major dendrites.” which is in line with our Figure 1F. Finally, let us note that while mitochondria are an important ATP source, they are not the only energy source. It is conceivable that different cellular processes rely on different energy sources. For example, selected energy demands (e.g., synaptic plasticity) may be at least partially served by alternative ATP sources including anaerobic glycolysis⁸⁴. While the mitochondrial oxidative phosphorylation (OXPHOS) system is the final biochemical pathway in the production of ATP, aerobic glycolysis can also make substantial contributions to biosynthesis and has been proposed as a marker of synaptic plasticity⁸⁵ and autonomous vesicle transport⁸⁶.

17. Figure 4a. The graphs show that the most abundant mRNA has a \log_{10} value of 15, which is 1.176 mRNAs for a dendrite of 1000 μm . For the right panel showing protein, for the most abundant protein, the total predicted copy number is less than \log_{10} 20, which equals 1.3 proteins across the entire dendrite. How can that be correct?

Answer: The scale of the y-axis in Figure 4A is \log_{10} , i.e., a value of x on it corresponds to an actual value (in linear scale) of 10^x . Regarding the values we compute for the total mRNA and protein numbers in a neuron, we compare our predictions with experimental data^{14,36,37}, finding them to be in good agreement (Figure 4B). Here, the number of mRNAs per species varies between ~ 1 and $\sim 10^5$ per neuron, and the number of proteins per species is in the range of $\sim 10^4$ to $\sim 10^8$ copies per neuron, both in our simulations and the experimental data.

18. To estimate and predict a single neuron’s total protein and mRNA copy number (in absolute numbers), you may have to know how big your model neuron is, the total length of its dendrites, and the number of spines/synapses as all those measurements directly influence how much protein and mRNA will be needed to fill up the relevant synapses. Please discuss and provide some numbers.

Answer: In our model, we consider both the length of the dendrite and the density of spines. Here, we varied the dendritic length between 250 and 1000 μm to cover most dendrites present in rodent brains (Figure S10). Longer dendrites lead to higher total mRNA and protein levels, their total counts scale exponentially with length (Figure 4A). For the exact numbers, we refer the referee to Figure 4A and its insets. We keep the spine density at 1 spine per μm , which we find to be a reasonable

and general value (for a comprehensive discussion see the section ‘Spine density’ in the Supplement). It aligns well with the values reported in the experimental databases we use, e.g.¹⁴.

19. *Figure S12: the correlation between total mRNA and protein numbers appears unrealistically high given previous literature which typically shows lower values (see Liu et al, 2016 cited by the authors). The authors should discuss why their values appear high and the implications of that for their model.*

Answer: Thank you for this comment. Figure S12 shows the predicted correlations between mRNA and protein levels that follow from the assumptions we made when constructing our model. As the referee correctly points out, the correlations reviewed by Liu et al, 2016⁸⁷, are somewhat lower than the ones we compute from our simulations. Liu et al, 2016⁸⁷, themselves point towards many factors affecting the underlying protein and mRNA levels, and hence the measured correlation. Many of these noise factors originate either in the measurement imprecision or in the variable cell states, in fact the authors list effects such as cell states, activity, or experimental error in measuring either mRNA or protein levels as possible error sources. We are aware that our statement “We found that the model-predicted correlation between baseline mRNA and protein levels has also been observed experimentally (reviewed in ⁸⁷)” might imply that we reproduce exactly the values they report. Following your recommendation, we therefore modified this sentence and now write “Experimentally-measured baseline mRNA and protein levels are known to correlate (reviewed in ⁸⁷). Because mRNA and protein levels are subject to variability, e.g. cell state or activity fluctuations, the experimentally observed correlations can be lower than those predicted by our model.”

20. *In the first paragraph of the ‘Results’ section the authors refer to their model capturing the mRNA and protein density inside the soma and other parts of the neuron. Are you referring to absolute number of molecules or number of molecules per some unit area? If per unit area, what units are used? Please specify. Straight after they mention protein concentration. Is it meant to be the same as protein density? Or is this a different measure? What are the units for concentration? Is it 2D or 3D spine/shaft/soma?*

Answer: Our terminology in the first paragraph of the ‘Results’ section was indeed not perfectly clean, thank you for bringing this up. In the soma, our model gives only absolute molecule counts. In the dendrites and spines, concentrations are to be understood per micrometer dendrite length, i.e., along the dendritic shaft. We improved our writing by removing the word “density” from this paragraph and adding the unit of concentrations “[...] concentration per μ dendritic length of dendritic mRNAs, [...]”.

21. *Throughout the text the authors refer to “Supplemental Information” without directing the reader to the actual part of the Supplemental information, which contains 13 figures, 4 tables and 14 pages of “Additional Details on the Mathematical Model and Parameters”. The authors must specify the precise figure/table/section of the Supplemental information at each point in the text.*

Answer: Thank you for this comment, we have now added the specific supplemental references throughout the manuscript.

22. *The synthesis and degradation rates are denoted by a single variable. Given existent evidence that protein synthesis and lifetime are synapse-specific, how accurately does a single variable per cellular process capture the diversity? Please discuss.*

Answer: Thank you for this interesting question. Experiments measuring the life-times or degradation rates frequently consider the fraction of labelled molecules that are present after a given time and these fractions are most commonly fitted by a single exponential, e.g.,^{88,89}. Experimental variability in relative protein count measurements across time can be large generating variability in measured half-lives. Challenges in determining the precise difference in half-lives as a function of synaptic activity or synapse-to-synapse variability have been the topic of several prominent reviews, for example,⁹⁰.

To rule out the existence of any population-level effects due to a local heterogeneity in synthesis or degradation, we re-simulated Figure 2 (see Reply Figure 14 below) while adding a spatial synthesis or degradation profile (gaussian, 10% peak increase) on top of the constant rates. Here, we found the same predictions as we show in Figure 2 in the main manuscript indicating that possible experimentally reported variability of half-lives across the dendrite does not alter our model predictions. Therefore, we decided to use in our model constant synthesis and degradation rates to represent the endogenous protein distributions.

Reply Figure 14: Local variations of translation and degradation do not affect the predicted energy efficiency along parameter dimensions. Here, we re-simulated Figure 2 for the same parameter space and added a local variation in the translation (**A1-E1**) or the degradation (**A2-E2**). Local variations are established by adding Gaussian kernels, positioned at $200\mu\text{m}$ with standard deviation $10\mu\text{m}$ and peak height 10% of the baseline value. The findings of Figure 2 are preserved under these variations.

23. *Since the proteins are not uniformly distributed among all synapses along the dendrite, please discuss the known mechanisms involved in active capture of the protein into specific spines and their associated costs. Are these costs substantial to affect the calculations in the model?*

Answer: Thank you for the suggestion to discuss the costs and mechanisms associated with the transfer of proteins from the dendrite into spines in more depth. Overall, the entry of proteins into the spines is mediated by diffusion through the spine neck. Several previous modelling and experimental studies corroborated the prominent contribution of spine neck diffusion and visualized experimentally the motion of proteins through the spine neck including^{91,92}) and modelled this computationally⁹³. To cross-check further whether the diffusive spine-dendrite coupling in our model can reproduce the protein intake as a function of time and dendritic distance from the soma, we decided to use the Shank3 data³¹ (Figure 5) to validate these dynamics in our model. Here, we found that the temporal spine-filling dynamics are captured by our model. Therefore, we did not associate any cost (diffusion is energetically cost-free) with the entry of proteins into the spine. However, just for completeness let us calculate the cost associated with active transport if the spines were filled via active transport assuming a distance of $1\mu\text{m}$ to enter a spine from the dendrite underneath, 100 proteins per spine (motivated by Figure S1C) and 100 spines along a dendrite of $100\mu\text{m}$ length (assuming 1 spine per μm) and assuming a protein half-life of 10 days (motivated by Figure S1A, middle point). One would need 1.250.000 ATPs to fill these spines since only half of the proteins would need to be replenished after an average of 864.000 seconds (10 days), which translates to a cost of 0.72 ATP per second. This is negligible compared to the 10^5 average cost needed to maintain a typical protein distribution across a dendrite of this length, see for example Figure 1B or Figure 4.

24. *The authors should discuss how the model will be affected by branching in the dendrites which obviously could affect the distributions of mRNA and proteins in more distal parts of the neuron and will likely significantly depend on the neuron type.*

Answer: Thank you for the suggestion to discuss the potential influence of branching. At a branch point, the molecules can decide to go either back into the mother branch or into one of the daughter branches. The probabilities for each of these three actions can depend either on the ratios of circumferences or on the ratios of the cross-sectional areas, according to Sartori et al., 2020⁹⁴ or on other factors that are yet to be discovered. For example, how branching affects active transport is less well understood. In particular, it is an open question whether the number of microtubular tracks is preserved at a branch point, how the available microtubular tracks are distributed across the three directions of a branch point, or how the area/circumference ratios are determining the active motor movement. As this information becomes available, influences of branching concerning active mRNA and protein transport could be studied in our model. We have now added to the discussion the following sentences “Additionally, our model could serve as a powerful platform addressing how differences in branching translate into molecular localisation preferences and different energy demands as more experimental evidence becomes available about which cell types do or do not preserve the number of microtubular tracks between the mother and daughter branches, on how the area/circumference ratios at branch points⁹⁴ determine the distribution of microtubules, mRNAs and proteins between the mother and daughter branches.”

25. *The datasets that the authors refer to are derived from many different sources including primary neuronal cultures, whole brain homogenates, different species, different brain areas, different cell types and different parts of the dendritic tree. This information could be added to Table S1 and S2. The authors need to specify which dataset and its source material is being referred to in the main text at the point the data is being used/described. For example, Figure 1C shows data from Fonkeu et al but it is not clear if the measure is from a single dendrite or population, which part of a dendrite, from primary cell culture or in vivo, and what cell type.*

Answer: Following the reviewer’s suggestion we have added this information to Tables S1 and S2. We have now expanded the description of Figure 1C in the main text to give more details: “the experimentally measured cell- and dendrite averaged profile in cultured rat hippocampal pyramidal cells reported in³⁴ (Figure 1C).” and concerning Figure 1F we now write “Remarkably, we found that the local energy need predicted by our model is within the 1.4σ interval of the experimentally measured mitochondria distribution in dendrites of hippocampal pyramidal cells of mice⁷⁷, see Figure 1F.” We took care to provide as much information about the tissue preparation methods as possible and for more details we refer the interested reader to the methods sections of the corresponding papers. Regarding GluA1, we added the description of cell type and measurement to the caption of Figure S14.

26. *The “protein diffusion constants” presented in Table S1: are they passive transport, microtubule-associated transport or*

a mix? Should be specified where possible.

Answer: All values in Table S1 are protein diffusion constants measured for the free diffusion inside the cytoplasm, we have added this to the information to the caption of Table S1: “Summary of curated passive protein diffusion constants matched with their corresponding mRNA enrichment scores”.

27. *On page 3, where the authors explain what they consider when calculating the “energy cost (in ATP per second) necessary to maintain each protein distribution”, they do not mention protein degradation. Protein degradation is an essential component of maintenance of stable protein distribution and proteasome-mediated degradation does cost ATP and really should be included in the considerations and should be well explained in main text.*

Answer: Thank you for this remark. Protein degradation is indeed contributing to the energy budget, as shown by¹². We did include protein degradation costs in our model. Briefly, we consider costs per amino acid of 4 ATP for protein biosynthesis and 1 ATP for degradation, totaling 5 ATP per amino acid. Following this referee suggestion we made this more precise by writing “[...] the cost for protein biosynthesis and degradation together amounts to 5 ATP per amino acid^{12,95,96}. We collectively refer to these costs as ‘translation’ cost [...]”.

28. *When introducing the calculations of energy costs in ATP for the different processes, including transcription, translation and active transport, it would be helpful to include a figure or a table with values for an example protein and its distribution in an example dendrite of some unit length. It would help put those numbers into perspective and more intuitively visualise the most costly processes.*

Answer: The first section of the ‘Results’ section and Figure 1B is where we provide an exemplary calculation of costs for the prominent postsynaptic protein CaMK2a, Figure 1C, where we show the associated distributions of mRNA and protein, and Figure S5, where we summarize the relative size of cost factors.

29. *For fig. 1A, please expand the description and remove the passive diffusion from the illustration since it does not require energy.*

Answer: We followed the referee’s advice and expanded the description of Figure 1A, detailing which processes we consider and which of them consume energy. We now write “Overview of the processes considered in our model. Yellow coins denote energy-consuming processes, processes lacking a coin are energetically free (e.g. diffusion).”

We acknowledge that the previous panel’s description emphasized the energy-consuming processes. Now, the caption points to the main purpose of Figure 1A, which is to summarize all processes that we included in our model and also show which of them are associated with energy consumption and which are not.

30. *On page 4, the authors say that they “defined three groups along each of the seven dimensions: small, medium, and large”. Were the three groups defined by percentile or other measure of the distribution? Since the 3 groups are defined not by a range but by a single point value, how was the range for the 3 groups defined? Please discuss and clarify.*

Answer: Thank you for the suggestion to clarify how we determined the small, medium, and large values for each parameter. In Figure S1 we show the experimentally measured distributions we considered for each parameter, and the sampling strategy is explained in greater detail in the section ‘Sampling of synthetic protein species’ in the Supplement. In summary, we wanted to apply the same sampling strategy to all parameters simultaneously to avoid biases. Yet, for some of the parameters, e.g., protein half-life, multiple resources were available that featured different sample sizes and different ranges of values. Because we wanted to take into account as many of the resources as possible, we decided to go for a manual positioning considering three points that are representative of the ‘center’ and the ‘lower’ and ‘upper’ variability of the measurements, we corroborated this choice with experimental experts in the field. Let us stress that our optimal energy predictions do not depend on the precise values of the three exact parameter values per parameter dimension, we rather used them to span the range of plausible parameter values and study how specific directions (e.g., longer vs shorter half-lives) influence the molecular localisation preferences.

To make the parameter information more accessible, we expanded the caption of Figure S1 and added Table S6 (shown here for convenience as Reply Table 3) containing a parameter overview.

Parameters	Sampled in Figure 2 Figure S1	Citation
mRNA half-life [hours]	2, 8, 20	7-10
mRNA length [#Non-coding/coding nucleotides]	(3, 6, 15) * aa [0/3, 3/3, 12/3]	11-13
Protein copy number	10, 200, 5000	14
Protein half-life [days]	2, 8, 20	15-18
Protein length [amino acids]	100, 500, 2000	16
mRNA diffusion [$\mu m^2/s$]	10^{-4} , 10^{-3} , 10^{-2}	2,3,19-30
Protein diffusion [$\mu m^2/s$]	$10^{-2.5}$, $10^{-1.5}$, $10^{-0.5}$	Table S1

Reply Table 3: (added as Table S6) Sampled parameter space range considered for our main predictions in Figures 1, 2, 4. See Figure S1 for the actual parameter distributions.

31. *In Fig. 2 the authors present how the “preference for dendritic vs. somatic mRNA changes as a function of the parameter of interest” and present the plots for 5 parameters out of the 7 that were mentioned earlier in the paragraph. Where are the other two and why not include and discuss them?*

Answer: Thank you for the suggestion to clarify this point. In Figure 2 we present model predictions along the five parameter dimensions (five panels in Figure 2) that currently have enough data to be quantitatively tested. Currently available datasets on five parameters contain hundreds or more entries, allowing us to perform reliable statistics on them. The model predictions for two parameters (out of seven contained in the model, which are the diffusion coefficients of mRNAs and proteins) could be tested when more values are measured by future studies. For example, investigating the available protein diffusion coefficients we found 32 different values from a variety of experimental paradigms. While the currently available data shows a trend which aligns with the model prediction (Figure S8A) that proteins with larger diffusion coefficients energetically prefer somatic mRNA localisation (corresponding to a smaller neurite/soma mRNA ratio), $N = 32$ is not enough to achieve a level of significance which is comparable to that of the five parameters shown in Figure 3. To discuss the remaining model predictions that go beyond the five we presented in Figures 2, 3, we write in the discussion section “Interestingly, our simulations also suggested that proteins with higher diffusion constants preferentially localise mRNA in the soma (Figure S8A). We curated and expanded a previously published list of diffusion constants⁹⁴ matching each reported protein with the corresponding soma/dendrite mRNA enrichment scores³⁸ (Table S1) revealing a trend in line with our prediction (Figure S8B). As future studies provide more diffusion records, this model-predicted relation could be further tested.”

32. *Page 4-5, authors refer to Figure S8 saying that their “results hold across different dendritic lengths”. Fig. S8 shows nothing to do with dendritic lengths. Please fix.*

Answer: Thank you for noticing this. We fixed the references to the Supplementary Figures. In this particular case we now write “When the dendrite grows, it will acquire more synapses and exponentially increase its total protein need with dendrite length (Figures S9, S10). However, we found the model predictions discussed above to be largely independent of dendritic length (Figure S3).”

33. *Page 5, authors refer to ‘somata-enriched’ and ‘neurite-enriched’ mRNA species that were used in their analysis but it is unclear what study the data was taken from (please reference in main text), whether the data is from intact tissue or cell culture, what parts of neurites (i.e. apical/basal/proximal/distal) were included in the analysis and what cell type was analysed. Please clearly state in text.*

Answer: The required information has now been added to Table S2.

34. *On page 5, authors state “we found that species with neurite-enriched mRNAs indeed exhibit significantly lower protein half-lives (Figure 3C)”. Figure 3C shows data for protein copies per spine and there is no mention of protein half-lives in this figure. Please fix.*

Answer: We went through the text and corrected the order of panels being mentioned related to Figure 3.

35. *For ‘soma-enriched’ and ‘neurite-enriched’ mRNA species that are discussed and analyzed in the paper, what fraction of all those mRNA species code for synaptic proteins? Did the authors filter to only analyze the synaptic proteins? Please clarify.*

Answer: Thank you for raising this question. Since we gathered our data from various resources and different studies, all mentioned in Reply Table 4, we did not apply this filter to the data. Because not all of the datasets include reported synaptic protein-coding mRNA, we would lose a considerable amount of our cross-matching data through such filters. Thus we only used filters explained in the manuscript.

36. *In text, the authors refer to figure 3E when saying the following: “We could also confirm our third prediction that the protein copy number per spine is independent of the preference of mRNAs for soma or neurites”. Figure 3E refers to protein length in amino acids and has no mention of protein copy numbers. Please fix.*

Answer: Thank you for noticing. We corrected the panel references of Figure 3 in the text.

37. *On page 5, where authors aim to “predict a single neuron’s total protein and mRNA copy number per transcript” and use the synthetic model proteins for this, I can see how you can aim to predict the relative localization preferences of mRNAs and proteins across the whole dendritic tree, but how do you work out the total numbers of mRNAs and proteins per cell? Do we know what determines the total numbers of any given mRNA or protein species per any given cell? Please expand.*

Answer: Since all of our parameters come from experimental measurements, we obtain ‘actual’ numbers of mRNAs and proteins across our model neuron (and not, for example, arbitrary units), i.e., a resulting density of 1 protein per μm along a $100\mu\text{m}$ long dendrite can be integrated to 100 proteins in total. Similarly, we obtain total mRNA and protein counts for any distribution simply by integration along the dendritic distance. The resulting distributions nicely align with experimental quantifications of mRNA and protein copy numbers per neuron (Figure 4).

38. *Figure S3. The legend refers to “Blue, empty” bars but none are visible in the figure.*

Answer: The bars denote the share of synthetic protein species for which a given strategy is optimal. For example, the empty blue bar represents those energetically favouring protein transport in combination with somatic mRNA. Since no protein species energetically favoured this strategy, the corresponding blue bar is not visible. We make this more clear by adapting the caption of Figure S3: “For each synthetic protein species within our parameter space we computed the total cost for four trafficking options: 1) somatic mRNA with no active transport at all (red, empty), 2) somatic mRNA with protein transport in absence of mRNA transport (blue, empty)[...], wherefore we omit the associated bar (blue, empty).”

39. *Figure S6. There is no scale for Y-axis.*

Answer: Thanks for noticing, we added it.

3 Reviewer #3 (Remarks to the Author)

The manuscript by Bergmann et al. "How energy determines spatial localization and copy number of molecules in neurons," addresses important questions in the field concerning the parameters that affect the localization of RNAs and proteins in neuronal cells. It presents a computational framework to explain the spatial localization profiles and copy number distributions of mRNAs and proteins in neurons. The core proposition of the manuscript is that molecular species that generate high energy costs benefit more from dendritic mRNA localization. I found reading the manuscript enjoyable. The conclusion that RNA transport and local translation can be energetically more efficient than protein transport is insightful and holds significant importance for the field.

I have a few comments and questions:

1. *What literature data are available on how many proteins are localized primarily through RNA transport and local translation compared to those localized primarily through protein transport? A summary would be useful.*

Answer: Thank you for this interesting comment. We have carefully researched this topic and consulted with leading experts on active transport and local translation and found that there is currently no systematic overview of proteins that are actively transported. The identities and statistics of proteins that have dendritic mRNAs can be derived from mRNA localisation scores from ^{10,38}, which we use throughout our manuscript (see also Reply Table 4). The mRNA localisation scores from these studies report the ratio (number of dendritic mRNA)/(number of somatic mRNAs). By filtering the gene names that correspond to larger quantities of dendritic compared to somatic mRNA, one can identify the names of the corresponding proteins. However, it is currently not possible to construct similar statistics for proteins that are actively trafficked along the microtubules. We are not aware of data that would allow us to tell if there is an overlap between the group of locally synthesized proteins and those that are actively transported, or if these groups are distinct. What is known is that active transport has been observed for certain receptor complexes such as AMPAR, NMDAR, GABAR, and GlyR, and that it is considered unlikely that membrane proteins in general, including receptors, will reach synapses by pure diffusion. Despite this, it is difficult to provide an exact percentage for the actively transported protein class. Typical neuronal proteins have a maximum length of about 2,000 amino acids. Assuming an average mass of 118.9 g/mol (Da) per amino acid, this results in an approximate mass of 236 kDa. Referring to the statement by ⁹⁷, which can be cited as follows: "while 500 kDa molecules can diffuse freely across a cell⁹⁸, complexes larger than 2 MDa are confined and effectively immotile. Thus, cargoes for dynein are typically large objects, such as organelles and ribonucleoprotein (RNP) or protein complexes."

Additionally, ⁹⁸ states: "Our results indicate relatively free and rapid diffusion of macromolecule-sized solutes up to approximately 500 kD in cytoplasm and nucleus." This implies that most dendritic proteins could engage cytoplasmic diffusion and not be actively transported via dynein or kinesin. Notable exceptions include larger complexes or organelles such as mitochondria and receptor complexes like AMPAR and NMDAR.

While it is theoretically possible to determine the percentage of proteins exceeding 500 kDa based on amino acid length distribution (it is relatively small), it cannot be used as an exclusion criterion for active protein transport, because whether or not a given protein can be actively transported is a multi-factorial question and may depend on 1) whether a protein is part of a membrane-bound complex, 2) whether it can bind to a transport-competent protein, 3) whether it can or will be sorted into a transport-competent vesicle, (4) whether it is transiently associated to a cell organelle (Golgi, autophagosome, amphisome, lysosome, MVB, recycling endosome etc.), (5) whether it forms large complexes with other molecules, (6) and many other factors that are yet to be understood.

Therefore, taking the currently sparse knowledge on actively transported proteins into account, we cannot directly compare our model predictions on active protein transport to data (in the same way as we did for dendritic vs somatic mRNA localisation in Figure 3). Notably, our theory offers predictions that can be tested experimentally in future studies. In Figure S3 we show that actively transporting proteins is only an energy-optimal solution to maintain baseline protein levels across spines if the corresponding mRNA is also transported into the dendrites and if the dendrites are very long ($\geq 750\mu\text{m}$). Most studies quantifying molecular transport and mRNA localisation work with in vitro models whose dendrites are less than $\approx 750\mu\text{m}$ long (see Figure S11) which is a regime where our model predicts active protein transport to be rare. Our model further predicts that some proteins benefit energetically from both dendritic mRNA localisation and active protein transport (purple bar in Figure S3). These are interesting predictions to be tested as we now write in the discussion "In our modelling framework, mRNA transport is predicted to be almost always energetically more efficient than protein transport to maintain the baseline protein distribution across spines (Figures S3, S4), particularly in shorter dendrites. Future experiments targeting the population of actively trafficked proteins can shed light on the statistics of actively trafficked proteins in relation to the locally

synthesized protein group.”

2. In calculating transport costs, is it assumed that each molecule is transported individually? Recent studies suggest that mRNAs are localized as parts of larger granules with multiple molecules co-transported together. How would this affect the calculation of energy costs and the model?

Answer: Thanks for pointing this out. Based on our review of the existing literature we concluded that most mRNAs do indeed travel alone along microtubules (see section ‘mRNA granule content’ in the Supplement). Still, following the referee’s suggestion, we wanted to check the effect of bigger granules on our main findings. We did so by assuming that always two mRNAs are trafficked together per granule and obtained similar results as shown in Figure 2 (Reply Figure 15). We here assume 2 mRNAs per granule, because, to our knowledge, bigger granules are observed only extremely sparsely.

Reply Figure 15: Re-simulation of Figure 2 with 2 mRNAs per granule.

3. Figure 2 demonstrates that longer proteins favour dendritic RNA localization (2E), but lower non-coding/coding RNA ratios favour somatic localization (2B). How were the energy costs for the data in 2B calculated? Is the difference in energy costs for different non-coding/coding ratios due to varying protein lengths (e.g., longer proteins lead to a lower non-coding/coding ratio) or other factors? How does the plot (2B) look when UTR length is analyzed instead of different non-coding/coding ratios?

Answer: Thanks for this comment. We will answer this comment in two ways. First, we follow the referee's suggestion and show a version of Figure 2 using different UTR lengths per transcript and not the non-coding/coding nucleotide ratio (Reply Figure 16). Second, we will argue that the non-coding/coding nucleotide ratio represents the same effect compared to adding UTRs of different lengths.

First, we re-simulated Figure 2 using UTR lengths of 100, 1000, 5000 (Reply Figure 16). This means that, for a given protein, we have three transcript lengths. Their lengths are given by three times the amino acid number (the coding sequence) plus the UTR, which is either 100, 1000, or 5000 nucleotides long. Our findings (Figure 2) are preserved.

Second, we want to argue that our sampling approach based on the non-coding/coding nucleotide ratio is conceptually the same as sampling additive UTR lengths. For a given protein with a given number of amino acids, we sample three non-coding/coding nucleotide ratios. This gives three total transcript lengths for this protein, which contain the same coding region (three times the amino acid number) and three differently sized non-coding regions. The referee is correct insofar as bigger proteins have larger non-coding regions than small ones - but in Figure 2B we directly compare the localization of the three transcript lengths *per underlying protein*, so there should not be a distorting effect of the protein size itself. We acknowledge that this might not be clear at first glance, wherefore we visualized the results in Reply Figure 16 below.

Reply Figure 16: Re-simulation of Figure 2 with additive UTR lengths (as shown in B). For every parameter combination, we therefore use a nucleotide number of three times the amino acid number (the coding sequence) plus either 100, 1000, or 5000 nucleotides representing the UTR.

4. *The labeling of panels in Figure 3 does not match the text, and the panels are not presented in the same order as mentioned in the text.*

Answer: Thank you for spotting this. We now went through the manuscript text and cross-checked and corrected references to panel labels across all figures, including Figure 3.

5. *There is more than a dozen datasets from the literature pertaining to RNA localization and stability, yet the manuscript uses only a few for model validation. Other datasets should be included or an explanation for their exclusion should be provided. Moreover, data from all relevant datasets should be applied to test multiple parameters, i.e., in different panels of Figure 3.*

Answer: Thank you for this comment. In selecting the databases and matching proteins and mRNAs across databases, we have indeed evaluated multiple database pairings whereby some database pairings had much larger overlaps than others. In selecting databases for Figure 3 we gave preference to the databases which met two criteria: 1) had a large number of entries and 2) had the largest overlap with other databases, e.g., a database reporting half-lives vs a database reporting soma/dendrite localisation ratios needed to have a large overlap in molecular species. A summary of the datasets we considered can be found in Reply Table 4. We have cross-matched genes from datasets reporting one of the parameters of interest with datasets that reported other parameters of interest for the same gene (e.g., mRNA neurite-to-soma enrichment scores vs half-lives). To maintain consistency, we have used the same dataset for analysis whenever possible and gave preference to dataset combinations with the highest overlap. Shown in grey in Reply Table 4 are the cross-matched datasets we selected for Figure 3 based on the database overlap and total sample size.

Parameter combination	Main data	Matched data	Number of matches
mRNA half-life vs Enrichment scores	Tushev et al. 2018 ¹⁰	Tushev et al. 2018 ¹⁰	542
		Zappulo et al. 2017 ³⁸	542
		Glock et al. 2021 ⁹⁹	440
mRNA length vs Enrichment score	Zappulo et al. 2017 ³⁸	Zappulo et al. 2017 ³⁸	7029
Protein copy number vs Enrichment scores	Helm et al. 2021 ¹⁴	Zappulo et al. 2017 ³⁸	83
		Glock et al. 2021 ⁹⁹	59
		Tushev et al. 2018 ¹⁰	17
Protein half-life vs Enrichment scores	Fornasiero et al. 2018 ¹⁶	Zappulo et al. 2017 ³⁸	578
		Tushev et al. 2018 ¹⁰	410
		Glock et al. 2021 ⁹⁹	248
	Price et al. 2010 ¹⁵	Zappulo et al. 2017 ³⁸	255
		Tushev et al. 2018 ¹⁰	212
		Glock et al. 2021 ⁹⁹	114
Protein length vs Enrichment scores	Zappulo et al. 2017 ³⁸	Zappulo et al. 2017 ³⁸	7029
		Tushev et al. 2018 ¹⁰	1049
		Glock et al. 2021 ⁹⁹	987

Reply Table 4: Overview of the databases used to verify our model's predictions. For each parameter combination, we have cross-matched two databases, one with a reported parameter of interest and the other with mRNA localization score, and demonstrated in Figure 3, are the matched databases in gray that we have decided to present due to the higher number of matches. In the rare cases where two database combinations gave the same number of matches we gave preference to matches from the same dataset.

6. The authors discuss previous studies that show transcripts with shorter 3' UTRs tend to remain in the soma, while those with longer 3' UTRs are preferentially localized in dendrites, in relation to Figure 3B. However, Figure 3B shows mRNA length. What are the findings when 3' UTR length is considered?

Answer: We thank the referee for mentioning this point. In case the referee is interested in the model predictions when considering 3'-UTR length instead of mRNA length, we kindly refer to our answer to comment 3.3.

Here, we want to comment on the experimental findings concerning the localization of transcripts with different 3'-UTR lengths. To clarify that our statements on 3'-UTRs are indeed valid, we want to point to the findings of Erin Schuman's lab on the localization of transcripts in neurons. Tushev et al., 2018¹⁰ showed that transcripts localizing in dendrites are generally longer than those in the soma (Reply Figure 17A), and, when comparing transcripts for a given gene (which is equivalent to keeping the coding sequence constant and varying the UTR length), the transcript lengths in the dendrite are longer than those in the soma (see Reply Figure 17B). In addition, similar results have been independently obtained by¹⁰⁰ for BDNF. To highlight this in our manuscript, we added the following lines to the 'Discussion' section: "Interestingly, this model prediction can be observed not only at the population level (Figures 2, 3) but also in mouse hippocampal neurons within the mRNAs coding for the same protein. An et al., 2008¹⁰⁰ reported that the mRNAs of the prominent synaptic protein BDNF preferentially localise to the soma if they have a short 3'-UTR sequence. In contrast, BDNF mRNAs with long 3' UTR sequences preferentially localise in dendrites, confirming our model predictions at the level of individual protein species. This finding has been further corroborated across genes¹⁰."

Reply Figure 17: Figure 3D (Left) and 5B (right) of Tushev et al., 2018¹⁰

References

- ¹ Bauer, K. E. *et al.* Live cell imaging reveals 3-UTR dependent mRNA sorting to synapses. *Nature Communications* **10** (2019).
- ² Dynes, J. L. & Steward, O. Dynamics of bidirectional transport of Arc mRNA in neuronal dendrites. *The Journal of Comparative Neurology* **500**, 433–447 (2006).
- ³ Park, H. Y. *et al.* Visualization of Dynamics of Single Endogenous mRNA Labeled in Live Mouse. *Science* **343**, 422–424 (2014).
- ⁴ Rook, M. S., Lu, M. & Kosik, K. S. CaMKII α 3 Untranslated Region-Directed mRNA Translocation in Living Neurons: Visualization by GFP Linkage. *The Journal of Neuroscience* **20**, 6385–6393 (2000).
- ⁵ Zimyanin, V. L. *et al.* In Vivo Imaging of oskar mRNA Transport Reveals the Mechanism of Posterior Localization. *Cell* **134**, 843–853 (2008).

- ⁶ Glock, C., Heumüller, M. & Schuman, E. M. mRNA transport & local translation in neurons. *Current Opinion in Neurobiology* **45**, 169–177 (2017).
- ⁷ Schwanhäusser, B. *et al.* Global quantification of mammalian gene expression control. *Nature* **473**, 337–342 (2011).
- ⁸ Yang, E. *et al.* Decay Rates of Human mRNAs: Correlation With Functional Characteristics and Sequence Attributes. *Genome Research* **13**, 1863–1872 (2003).
- ⁹ Sharova, L. V. *et al.* Database for mRNA Half-Life of 19 977 Genes Obtained by DNA Microarray Analysis of Pluripotent and Differentiating Mouse Embryonic Stem Cells. *DNA Research* **16**, 45–58 (2009).
- ¹⁰ Tushev, G. *et al.* Alternative 3' UTRs Modify the Localization, Regulatory Potential, Stability, and Plasticity of mRNAs in Neuronal Compartments. *Neuron* **98**, 495–511 (2018).
- ¹¹ Hawkin, J. D. A survey on intron and exon lengths. *Nucleic Acids Research* **16**, 9893–9908 (1988).
- ¹² Lynch, M. & Marinov, G. K. The bioenergetic costs of a gene. *Proceedings of the National Academy of Sciences* **112**, 15690–15695 (2015).
- ¹³ Roy, M., Kim, N., Xing, Y. & Lee, C. The effect of intron length on exon creation ratios during the evolution of mammalian genomes. *RNA* **14**, 2261–2273 (2008).
- ¹⁴ Helm, M. S. *et al.* A large-scale nanoscopy and biochemistry analysis of postsynaptic dendritic spines. *Nature Neuroscience* **24**, 1151–1162 (2021).
- ¹⁵ Price, J. C., Guan, S., Burlingame, A., Prusiner, S. B. & Ghaemmaghami, S. Analysis of proteome dynamics in the mouse brain. *Proceedings of the National Academy of Sciences* **107**, 14508–14513 (2010).
- ¹⁶ Fornasiero, E. F. *et al.* Precisely measured protein lifetimes in the mouse brain reveal differences across tissues and subcellular fractions. *Nature Communications* **9**, 4230 (2018).
- ¹⁷ Dörrbaum, A. R., Kochen, L., Langer, J. D. & Schuman, E. M. Local and global influences on protein turnover in neurons and glia. *eLife* **7**, e34202 (2018).
- ¹⁸ Mathieson, T. *et al.* Systematic analysis of protein turnover in primary cells. *Nature Communications* **9**, 689 (2018).
- ¹⁹ Elvira, G. *et al.* Characterization of an RNA Granule from Developing Brain. *Molecular & Cellular Proteomics* **5**, 635–651 (2006).
- ²⁰ Batish, M., van den Bogaard, P., Kramer, F. R. & Tyagi, S. Neuronal mRNAs travel singly into dendrites. *Proceedings of the National Academy of Sciences* **109**, 4645–4650 (2012).
- ²¹ Farris, S., Lewandowski, G., Cox, C. D. & Steward, O. Selective Localization of Arc mRNA in Dendrites Involves Activity- and Translation-Dependent mRNA Degradation. *Journal of Neuroscience* **34**, 4481–4493 (2014).
- ²² Krichevsky, A. M. & Kosik, K. S. Neuronal RNA Granules. *Neuron* **32**, 683–696 (2001).
- ²³ Barbarese, E. *et al.* Protein translation components are colocalized in granules in oligodendrocytes. *Journal of Cell Science* **108**, 2781–2790 (1995).
- ²⁴ Obashi, K., Matsuda, A., Inoue, Y. & Okabe, S. Precise Temporal Regulation of Molecular Diffusion within Dendritic Spines by Actin Polymers during Structural Plasticity. *Cell Reports* **27**, 1503–1515.e8 (2019).
- ²⁵ Buxbaum, A. R., Haimovich, G. & Singer, R. H. In the right place at the right time: visualizing and understanding mRNA localization. *Nature Reviews Molecular Cell Biology* **16**, 95–109 (2014).
- ²⁶ Knowles, R. B. *et al.* Translocation of RNA Granules in Living Neurons. *The Journal of Neuroscience* **16**, 7812–7820 (1996).
- ²⁷ Tubing, F. *et al.* Dendritically Localized Transcripts Are Sorted into Distinct Ribonucleoprotein Particles That Display Fast Directional Motility along Dendrites of Hippocampal Neurons. *Journal of Neuroscience* **30**, 4160–4170 (2010).

- ²⁸ Mitsumori, K., Takei, Y. & Hirokawa, N. Components of RNA granules affect their localization and dynamics in neuronal dendrites. *Molecular Biology of the Cell* **28**, 1412–1417 (2017).
- ²⁹ Yan, X., Hoek, T. A., Vale, R. D. & Tanenbaum, M. E. Dynamics of Translation of Single mRNA Molecules In Vivo. *Cell* **165**, 976–989 (2016).
- ³⁰ Monnier, N. *et al.* Inferring transient particle transport dynamics in live cells. *Nature Methods* **12**, 838–840 (2015).
- ³¹ Tsuriel, S. *et al.* Local Sharing as a Predominant Determinant of Synaptic Matrix Molecular Dynamics. *PLoS Biology* **4**, e271 (2006).
- ³² Dave, P. *et al.* Single-molecule imaging reveals translation-dependent destabilization of mRNAs. *Molecular Cell* **83**, 589–606 (2023).
- ³³ Blake, L. A., Watkins, L., Liu, Y., Inoue, T. & Wu, B. A rapid inducible RNA decay system reveals fast mRNA decay in p-bodies. *Nature communications* **15**, 2720 (2024).
- ³⁴ Fonkeu, Y. *et al.* How mRNA Localization and Protein Synthesis Sites Influence Dendritic Protein Distribution and Dynamics. *Neuron* **103**, 1109–1122.e7 (2019).
- ³⁵ Harb, A. *et al.* Auxiliary subunits regulate the dendritic turnover of AMPA receptors in mouse hippocampal neurons. *Frontiers in molecular neuroscience* **14**, 728498 (2021).
- ³⁶ Zeisel, A. *et al.* Cell types in the mouse cortex and hippocampus revealed by single-cell RNA-seq. *Science* **347**, 1138–1142 (2015).
- ³⁷ Perez, J. D. *et al.* Subcellular sequencing of single neurons reveals the dendritic transcriptome of GABAergic interneurons. *eLife* **10**, e63092 (2021).
- ³⁸ Zappulo, A. *et al.* RNA localization is a key determinant of neurite-enriched proteome. *Nature Communications* **8**, 583 (2017).
- ³⁹ Shipman, S. L., Herring, B. E., Suh, Y. H., Roche, K. W. & Nicoll, R. A. Distance-dependent scaling of AMPARs is cell-autonomous and *gluA2* dependent. *Journal of Neuroscience* **33**, 13312–13319 (2013).
- ⁴⁰ Cull-Candy, S., Kelly, L. & Farrant, M. Regulation of Ca²⁺-permeable AMPA receptors: synaptic plasticity and beyond. *Current Opinion in Neurobiology* **16**, 288–297 (2006). Signalling mechanisms.
- ⁴¹ Mikasova, L. *et al.* Disrupted surface cross-talk between NMDA and Ephrin-B2 receptors in anti-NMDA encephalitis. *Brain* **135**, 1606–1621 (2012).
- ⁴² Kumari, P., Srinivasan, B. & Banerjee, S. Modulation of hippocampal synapse maturation by activity-regulated E3 ligase via non-canonical pathway. *Neuroscience* **364**, 226–241 (2017).
- ⁴³ Andrae, L. C. & Burrone, J. Spontaneous Neurotransmitter Release Shapes Dendritic Arbors via Long-Range Activation of NMDA Receptors. *Cell Reports* **10**, 873–882 (2015).
- ⁴⁴ Shirinpour, S. *et al.* Multi-scale modeling toolbox for single neuron and subcellular activity under Transcranial Magnetic Stimulation. *Brain Stimulation* **14**, 1470–1482 (2021).
- ⁴⁵ Chapleau, C. A. *et al.* Dendritic spine pathologies in hippocampal pyramidal neurons from Rett syndrome brain and after expression of Rett-associated MECP2 mutations. *Neurobiology of Disease* **35**, 219–233 (2009).
- ⁴⁶ Triesch, J., Vo, A. D. & Hafner, A.-S. Competition for synaptic building blocks shapes synaptic plasticity. *eLife* **7**, e37836 (2018).
- ⁴⁷ Broadhead, M. J. *et al.* PSD95 nanoclusters are postsynaptic building blocks in hippocampus circuits. *Scientific Reports* **6**, 24626 (2016).
- ⁴⁸ Chen, X. *et al.* Mass of the postsynaptic density and enumeration of three key molecules. *Proceedings of the National Academy of Sciences* **102**, 11551–11556 (2005).

- ⁴⁹ MacGillavry, H. D., Song, Y., Raghavachari, S. & Blanpied, T. A. Nanoscale Scaffolding Domains within the Postsynaptic Density Concentrate Synaptic AMPA Receptors. *Neuron* **78**, 615–622 (2013).
- ⁵⁰ Nair, D. *et al.* Super-Resolution Imaging Reveals That AMPA Receptors Inside Synapses Are Dynamically Organized in Nanodomains Regulated by PSD95. *Journal of Neuroscience* **33**, 13204–13224 (2013).
- ⁵¹ Lowenthal, M. S., Markey, S. P. & Dosemeci, A. Quantitative Mass Spectrometry Measurements Reveal Stoichiometry of Principal Postsynaptic Density Proteins. *Journal of Proteome Research* **14**, 2528–2538 (2015).
- ⁵² Bressloff, P. C. & Newby, J. M. Stochastic models of intracellular transport. *Reviews of Modern Physics* **85**, 135–196 (2013).
- ⁵³ Monzon, G. A. *et al.* Stable tug-of-war between kinesin-1 and cytoplasmic dynein upon different atp and roadblock concentrations. *Journal of cell science* **133**, jcs249938 (2020).
- ⁵⁴ Fusco, D. *et al.* Single mRNA Molecules Demonstrate Probabilistic Movement in Living Mammalian Cells. *Current Biology* **13**, 161–167 (2003).
- ⁵⁵ Tiruchinapalli, D. M. *et al.* Activity-Dependent Trafficking and Dynamic Localization of Zipcode Binding Protein 1 and β -Actin mRNA in Dendrites and Spines of Hippocampal Neurons. *The Journal of Neuroscience* **23**, 3251–3261 (2003).
- ⁵⁶ McKenney, R. J., Huynh, W., Tanenbaum, M. E., Bhabha, G. & Vale, R. D. Activation of cytoplasmic dynein motility by dynactin-cargo adapter complexes. *Science* **345**, 337–341 (2014).
- ⁵⁷ Donlin-Asp, P. G., Polisseni, C., Klimek, R., Heckel, A. & Schuman, E. M. Differential regulation of local mRNA dynamics and translation following long-term potentiation and depression. *Proceedings of the National Academy of Sciences* **118**, e2017578118 (2021).
- ⁵⁸ Köhrmann, M. *et al.* Microtubule-dependent Recruitment of Staufen-Green Fluorescent Protein into Large RNA-containing Granules and Subsequent Dendritic Transport in Living Hippocampal Neurons. *Molecular Biology of the Cell* **10**, 2945–2953 (1999).
- ⁵⁹ Yoon, Y. J. *et al.* Glutamate-induced RNA localization and translation in neurons. *Proceedings of the National Academy of Sciences* **113**, E6877–E6886 (2016).
- ⁶⁰ Kon, T., Nishiura, M., Ohkura, R., Toyoshima, Y. Y. & Sutoh, K. Distinct Functions of Nucleotide-Binding/Hydrolysis Sites in the Four AAA Modules of Cytoplasmic Dynein. *Biochemistry* **43**, 11266–11274 (2004).
- ⁶¹ Kural, C. *et al.* Kinesin and Dynein Move a Peroxisome in Vivo: A Tug-of-War or Coordinated Movement? *Science* **308**, 1469–1472 (2005).
- ⁶² Lakadamyali, M., Rust, M. J., Babcock, H. P. & Zhuang, X. Visualizing infection of individual influenza viruses. *Proceedings of the National Academy of Sciences* **100**, 9280–9285 (2003).
- ⁶³ Ma, S. & Chisholm, R. L. Cytoplasmic dynein-associated structures move bidirectionally in vivo. *Journal of Cell Science* **115**, 1453–1460 (2002).
- ⁶⁴ Ori-McKenney, K. M., Xu, J., Gross, S. P. & Vallee, R. B. A cytoplasmic dynein tail mutation impairs motor processivity. *Nature Cell Biology* **12**, 1228–1234 (2010).
- ⁶⁵ Das, S., Moon, H. C., Singer, R. H. & Park, H. Y. A transgenic mouse for imaging activity-dependent dynamics of endogenous Arc mRNA in live neurons. *Science Advances* **4**, eaar3448 (2018).
- ⁶⁶ Amrute-Nayak, M. & Bullock, S. L. Single-molecule assays reveal that RNA localization signals regulate dyneindynactin copy number on individual transcript cargoes. *Nature Cell Biology* **14**, 416–423 (2012).
- ⁶⁷ Guillaud, L., Setou, M. & Hirokawa, N. KIF17 Dynamics and Regulation of NR2B Trafficking in Hippocampal Neurons. *The Journal of Neuroscience* **23**, 131–140 (2003).
- ⁶⁸ Hangen, E., Cordelières, F. P., Petersen, J. D., Choquet, D. & Coussen, F. Neuronal Activity and Intracellular Calcium Levels Regulate Intracellular Transport of Newly Synthesized AMPAR. *Cell Reports* **24**, 1001–1012.e3 (2018).

- ⁶⁹ Neupert, C. *et al.* Regulated Dynamic Trafficking of Neurexins Inside and Outside of Synaptic Terminals. *The Journal of Neuroscience* **35**, 13629–13647 (2015).
- ⁷⁰ Setou, M., Nakagawa, T., Seog, D.-H. & Hirokawa, N. Kinesin Superfamily Motor Protein KIF17 and mLin-10 in NMDA Receptor-Containing Vesicle Transport. *Science* **288**, 1796–1802 (2000).
- ⁷¹ Song, A.-h. *et al.* A Selective Filter for Cytoplasmic Transport at the Axon Initial Segment. *Cell* **136**, 1148–1160 (2009).
- ⁷² Jaqaman, K. *et al.* Cytoskeletal Control of CD36 Diffusion Promotes Its Receptor and Signaling Function. *Cell* **146**, 593–606 (2011).
- ⁷³ Hua, W., Young, E. C., Fleming, M. L. & Gelles, J. Coupling of kinesin steps to ATP hydrolysis. *Nature* **388**, 390–393 (1997).
- ⁷⁴ Schnitzer, M. J. & Block, S. M. Kinesin hydrolyses one ATP per 8-nm step. *Nature* **388**, 386–390 (1997).
- ⁷⁵ Vale, R. D. & Milligan, R. A. The Way Things Move: Looking Under the Hood of Molecular Motor Proteins. *Science* **288**, 88–95 (2000).
- ⁷⁶ Yildiz, A., Tomishige, M., Vale, R. D. & Selvin, P. R. Kinesin Walks Hand-Over-Hand. *Science* **303**, 676–678 (2004).
- ⁷⁷ López-Doménech, G. *et al.* Loss of Dendritic Complexity Precedes Neurodegeneration in a Mouse Model with Disrupted Mitochondrial Distribution in Mature Dendrites. *Cell Reports* **17**, 317–327 (2016).
- ⁷⁸ Chou, T. Ribosome recycling, diffusion, and mrna loop formation in translational regulation. *Biophysical Journal* **85**, 755–773 (2003).
- ⁷⁹ Dastidar, S. G. & Nair, D. A ribosomal perspective on neuronal local protein synthesis. *Frontiers in Molecular Neuroscience* **15**, 823135 (2022).
- ⁸⁰ Lu, H. E., MacGillavry, H. D., Frost, N. A. & Blanpied, T. A. Multiple Spatial and Kinetic Subpopulations of CaMKII in Spines and Dendrites as Resolved by Single-Molecule Tracking PALM. *The Journal of Neuroscience* **34**, 7600–7610 (2014).
- ⁸¹ Attwell, D. & Laughlin, S. B. An Energy Budget for Signaling in the Grey Matter of the Brain. *Journal of Cerebral Blood Flow & Metabolism* **21**, 1133–1145 (2001).
- ⁸² Calkins, M. J. & Reddy, P. H. Assessment of newly synthesized mitochondrial dna using brdu labeling in primary neurons from alzheimer’s disease mice: Implications for impaired mitochondrial biogenesis and synaptic damage. *Biochimica et Biophysica Acta (BBA) - Molecular Basis of Disease* **1812**, 1182–1189 (2011).
- ⁸³ Chavan, V. *et al.* Central presynaptic terminals are enriched in atp but the majority lack mitochondria. *PLOS ONE* **10**, 1–19 (2015).
- ⁸⁴ Rumpf, S., Sanal, N. & Marzano, M. Energy metabolic pathways in neuronal development and function. *Oxford Open Neuroscience* **2** (2023).
- ⁸⁵ Shannon, B. J. *et al.* Brain aerobic glycolysis and motor adaptation learning. *Proceedings of the National Academy of Sciences* **113**, E3782–E3791 (2016).
- ⁸⁶ Hinckelmann, M.-V. *et al.* Self-propelling vesicles define glycolysis as the minimal energy machinery for neuronal transport. *Nature Communications* **7**, 13233 (2016).
- ⁸⁷ Liu, Y., Beyer, A. & Aebersold, R. On the Dependency of Cellular Protein Levels on mRNA Abundance. *Cell* **165**, 535–550 (2016).
- ⁸⁸ Wang, Y. *et al.* Precision and functional specificity in mrna decay. *Proceedings of the National Academy of Sciences* **99**, 5860–5865 (2002). <https://www.pnas.org/doi/pdf/10.1073/pnas.092538799>.
- ⁸⁹ Chan, L. Y., Mugler, C. F., Heinrich, S., Vallotton, P. & Weis, K. Non-invasive measurement of mrna decay reveals translation initiation as the major determinant of mrna stability. *eLife* **7**, e32536 (2018).

- ⁹⁰ Cohen, L. D. & Ziv, N. E. Recent insights on principles of synaptic protein degradation. *F1000Research* **6** (2017).
- ⁹¹ Obashi, K., Taraska, J. W. & Okabe, S. The role of molecular diffusion within dendritic spines in synaptic function. *Journal of General Physiology* **153**, e202012814 (2021).
- ⁹² Bloodgood, B. L. & Sabatini, B. L. Neuronal Activity Regulates Diffusion Across the Neck of Dendritic Spines. *Science* **310**, 866–869 (2005).
- ⁹³ Holcman, D. & Schuss, Z. Diffusion laws in dendritic spines. *The Journal of Mathematical Neuroscience* **1**, 10 (2011).
- ⁹⁴ Sartori, F. *et al.* Statistical Laws of Protein Motion in Neuronal Dendritic Trees. *Cell Reports* **33**, 108391 (2020).
- ⁹⁵ Hu, X.-P., Dourado, H., Schubert, P. & Lercher, M. J. The protein translation machinery is expressed for maximal efficiency in *Escherichia coli*. *Nature Communications* **11**, 5260 (2020).
- ⁹⁶ Moldave, K. Eukaryotic Protein Synthesis. *Annual Review of Biochemistry* **54**, 1109–1149 (1985).
- ⁹⁷ Reck-Peterson, S. L., Redwine, W. B., Vale, R. D. & Carter, A. P. The cytoplasmic dynein transport machinery and its many cargoes. *Nature reviews Molecular cell biology* **19**, 382–398 (2018).
- ⁹⁸ Seksek, O., Biwersi, J. & Verkman, A. Translational Diffusion of Macromolecule-sized Solutes in Cytoplasm and Nucleus. *The Journal of Cell Biology* **138**, 131–142 (1997).
- ⁹⁹ Glock, C. *et al.* The translome of neuronal cell bodies, dendrites, and axons. *Proceedings of the National Academy of Sciences* **118**, e2113929118 (2021).
- ¹⁰⁰ An, J. J. *et al.* Distinct Role of Long 3 UTR BDNF mRNA in Spine Morphology and Synaptic Plasticity in Hippocampal Neurons. *Cell* **134**, 175–187 (2008).

Dear Reviewers,

thank you for forwarding us the many encouraging and helpful comments which helped us improve the accessibility of our work. Below we present a point-by-point reply. In *italics* are the comments of the referees followed by our respective answers.

1 Reviewer #1

I'm satisfied with the corrections. Nice paper!

Answer: Thank you very much for the many helpful comments and the appreciation of our work.

2 Reviewer #2

We thank the authors for their careful and systematic responses to our questions and for the corrections to the manuscript. This is a very interesting and timely study.

Answer: Thank you very much for the many helpful comments and the appreciation of our work.

3 Reviewer #3

I thank the authors for addressing my comments. This is an important manuscript that could make better use of the available datasets. Please see my specific comments below.

Previous referee comment 2. *In calculating transport costs, is it assumed that each molecule is transported individually? Recent studies suggest that mRNAs are localized as parts of larger granules with multiple molecules co-transported together. How would this affect the calculation of energy costs and the model?*

Our previous reply: Thanks for pointing this out. Based on our review of the existing literature we concluded that most mRNAs do indeed travel alone along microtubules (see section ‘mRNA granule content’ in the Supplementary Text). Still, following the referee’s suggestion, we wanted to check the effect of bigger granules on our main findings. We did so by assuming that always two mRNAs are trafficked together per granule and obtained similar results as shown in Figure 2 (Reply Figure 15). We here assume 2 mRNAs per granule, because, to our knowledge, bigger granules are observed only extremely sparsely.

New referee comment: *I feel the manuscript presents a somewhat selective interpretation of published data and a narrow assumption that mRNA granules contain only one or at most two mRNA molecules, despite studies both supporting and opposing this view. The authors should at least discuss this possibility in the text.*

Our new answer: Thank you for this interesting suggestion, which we have now incorporated in the manuscript. In addition to the scenarios where the transport-competent granules contained 1 or 2 mRNAs we now considered 10 mRNAs per granule. All of the predictions we show in Figure 2 continue to hold, see Reply Figure 1 below which is now Figure S18. We added a discussion of this variability to the results section “For completeness, let us also note that in Figure 2 we considered 1 mRNA per granule and confirmed in Figure S18 that our results hold if 2 or 10 mRNAs are transported in each granule. Since the number of mRNAs per granule is variable across the literature (see section ‘mRNA granule content’ in the Supplementary Text), we confirmed that our results hold across a broad experimentally plausible range.”. For completeness let us mention here the corresponding paragraph in the Supplementary Text ‘mRNA granule content’ section: “[...] mRNAs are known to be transported as part of specialised granules¹⁻³. Various reports showed that mRNA granules can contain exclusively one mRNA species⁴⁻⁷ and recent studies observed that many granules may carry only one mRNA molecule^{4,7-11}. Other experimental studies indicated the possibility that more than one mRNA could be present in a transport competent granule^{12,13}, e.g. 2 mRNAs (CamKII and Arc) were reported in a single granule¹⁴. As recent reviews indicate, the current technological advances are only beginning to tackle the molecular complexity, the sub-types, the assembly and the possible mRNA cargo inside the transport competent granules¹⁵ such that the number of mRNAs per granule and its upper bound is not yet known

but is a topic of active research^{15,16}. For our model here, we thus considered one mRNA per granule throughout the main manuscript and confirmed that our main results hold when considering 2 and 10 mRNAs per granule in Fig S18.”

Reply Figure 1: (Added as Figure S18) Results in Figure 2 remain valid if a granule contains multiple mRNAs. We re-simulated the mRNA and protein distributions with their associated energetic costs assuming 2 (A1-E1) and 10 (A2-E2) mRNAs per granule. All other parameters were chosen as in Figure 2 in the main text. This illustrates that the results shown in Figure 2 can be obtained equivalently in the presence of larger transport-competent mRNA granules.

Previous referee comment 3. *Figure 2 demonstrates that longer proteins favor dendritic RNA localization (2E), but lower non-coding/coding RNA ratios favor somatic localization (2B). How were the energy costs for the data in 2B calculated? Is the difference in energy costs for different non-coding/coding ratios due to varying protein lengths (e.g., longer proteins lead to a lower non-coding/coding ratio) or other factors? How does the plot (2B) look when UTR length is analyzed instead of different non-coding/coding ratios?*

Our previous reply: Thanks for this comment. We will answer this comment in two ways. First, we follow the referee's suggestion and show a version of Figure 2 using different UTR lengths per transcript and not the non-coding/coding nucleotide ratio (Reply Figure 16). Second, we will argue that the non-coding/coding nucleotide ratio represents the same effect compared to adding UTRs of different lengths. First, we re-simulated Figure 2 using UTR lengths of 100, 1000, 5000 (Reply Figure 16). This means that, for a given protein, we have three transcript lengths. Their lengths are given by three times the amino acid number (the coding sequence) plus the UTR, which is either 100, 1000, or 5000 nucleotides long. Our findings (Figure 2) are preserved. Second, we want to argue that our sampling approach based on the non-coding/coding nucleotide ratio is conceptually the same as sampling additive UTR lengths. For a given protein with a given number of amino acids, we sample three non-coding/coding nucleotide ratios. This gives three total transcript lengths for this protein, which contain the same coding region (three times the amino acid number) and three differently sized non-coding regions. The referee is correct insofar as bigger proteins have larger non-coding regions than small ones - but in Figure 2B we directly compare the localization of the three transcript lengths per underlying protein, so there should not be a distorting effect of the protein size itself. We acknowledge that this might not be clear at first glance, wherefore we visualized the results in Reply Figure 16 below.

New referee comment: *The new figure suggests that mRNAs with longer half-lives (16A) also have longer UTRs (16B). Is that actually the case? Which dataset(s) were used to generate this figure? All prior research in this field has shown the opposite: transcripts with longer 3' UTRs tend to contain more destabilizing elements and are generally less stable.*

Our new answer: Thank you for the interesting comment suggesting that a shared dendritic mRNA preference of two parameters could be mistaken for a correlation between these parameters and that we should take care to clarify this issue. In the previous Reply Figure 16 (shown below for convenience as Reply Figure 2) panels A and B show that both longer mRNA length and mRNA half-lives favor dendritic mRNAs. Each of these parameters has this effect independently. In other words, our model does not predict or use any correlation between mRNA length and half-life variables. To highlight this we now describe the parameter sampling procedure in the results section as follows: "we generated synthetic protein species by drawing uncorrelated samples from the biologically plausible range for each model parameter individually (N=3⁷=2187; Figure S1 for experimental sources, Table S6 for parameter overview and the Supplementary Text section 'Sampling of synthetic protein species' for more details)."

Reply Figure 16A shows that longer-lived mRNAs prefer dendritic mRNA localisation, and mRNAs with longer UTR lengths also prefer dendritic mRNA localisation, but one is not causally related to the other because mRNA half-life and UTR length are independent parameters in our model. To arrive at this result we generated more than 2000 synthetic proteins for our model whereby we sampled mRNA half-life, UTR length and other parameters without assuming any correlation between them. Across these proteins, there are all possible combinations of half-lives and UTR lengths and for each parameter combination, we calculated whether dendritic or somatic mRNA is energetically cheaper and then assigned the cheaper strategy for this particular parameter combination. To find out whether mRNA localisation was influenced by mRNA half-life we sorted these species into buckets based on the half-life (or another model parameter in B-E) while keeping all values for the other parameters random (e.g., for A we varied mRNA half-life while allowing the UTR length to be 100, 1000, or 5000). Reply Figure 2 (as well as Figure 2 in the main manuscript) indicates that in a group of mRNAs that have all possible half-lives and all possible UTR lengths without knowledge of any correlation between these values the transcripts that have longer half-lives or those with longer UTR lengths will tend to prefer dendritic localisation. Therefore, when comparing compartments soma vs dendrites we expect to see a separation such that longer UTRs and/or longer mRNA half-lives (regardless of whether or not half-lives and UTR lengths are correlated when considering the whole neuron) will be found in the dendrites.

For completeness, we decided to investigate if a correlation between mRNA length and half-life would affect our main predictions (Figure 2). We introduced a correlation between these variables, either a positive (0.2) or a negative correlation (-0.2). Doing so, we find that our results remain unchanged (Reply Figure 4A1-E1 (positive correlation) and A2-E2 (negative correlation)).

From the cell physiological perspective, it means that correlations between variables need to be carefully evaluated because even in the presence of a cell-wide parameter correlation local compartments could prefer a local parameter configuration leading to an apparent local correlation. For completeness, let us mention here Tushev et al., 2018¹⁷, who showed that longer mRNA half-life and longer 3'-UTRs are features of dendritically localized transcripts (but Tushev et al., 2018, report this not

for the whole cell, only for dendrites). For convenience, we show the corresponding figures from Tushev et al., 2018, below (Reply Figure 3). Let us mention that correlated parameters and sub-compartments may be an interesting direction of future research we have therefore expanded our Discussion accordingly: “Future studies can also adapt our model framework to investigate local and global correlations between individual parameters and integrate additional intracellular processes, e.g. temporal and spatial dynamics and metabolic costs associated with the temporal changes in synaptic copy numbers during plasticity, opening doors for a comprehensive view on molecular dynamics.”

Reply Figure 2: (Previous Reply Figure 16) Re-simulation of Figure 2 with additive UTR lengths (as shown in B). For every parameter combination, we therefore use a nucleotide number of three times the amino acid number (the coding sequence) plus either 100, 1000, or 5000 nucleotides representing the UTR.

Reply Figure 3: Figure adapted from Tushev et al., 2018¹⁷. Panel A shows that average UTR length is longer for dendritic mRNAs (red) vs somatic mRNAs (blackpoints). Figure adapted from Figure 3D in Tushev et al. Panel B also indicates that mRNA UTR length is longer for dendritic transcripts (adapted from Figure 5B in Tushev et al). Panel C shows that dendritic transcripts have longer UTRs. Let us note that this observation does not statistically imply that longer UTR length is correlated with longer half-life, it only shows that both parameters individually differentiate between soma and dendrite, if either UTR length or half-life is longer then the transcript would prefer dendritic localisation. At the level of a cell both parameters can be either positively or negatively correlated while still leading to the same mRNA localisation outcome (see Reply Figure 4).

Reply Figure 4: Results in Figure 2 remain valid in the presence of positively or negatively correlated mRNA half-life and mRNA length. We re-simulated the mRNA and protein distributions with their associated energetic costs on a restricted parameter space, artificially creating a positive (0.2 in **A1-E1**) and an negative correlation (-0.2 in **A2-E2**) of mRNA half-life and length (see text for description). All other parameters were chosen as in Figure 2 in the main text. This shows that the results shown in Figure 2 can be obtained equivalently in the presence of positively or negatively correlated mRNA half-life and mRNA length.

New referee comment 4. *The labeling of panels in Figure 3 does not match the text, and the panels are not presented in the same order as mentioned in the text.*

Our new answer: Thank you so much for spotting that we described the individual panels of Figure 3 not in the order they appear in the figure. Now we have edited the main text to match the order of the panels in Figure 3.

Previous referee comment 5. *There is more than a dozen datasets from the literature pertaining to RNA localization and stability, yet the manuscript uses only a few for model validation. Other datasets should be included or an explanation for their exclusion should be provided. Moreover, data from all relevant datasets should be applied to test multiple parameters, i.e., in different panels of Figure 3.*

Our previous reply: Thank you for this comment. In selecting the databases and matching proteins and mRNAs across databases, we have indeed evaluated multiple database pairings whereby some database pairings had much larger overlaps than others. In selecting databases for Figure 3 we gave preference to the databases which met two criteria: 1) had a large number of entries and 2) had the largest overlap with other databases, e.g., a database reporting half-lives vs a database reporting soma/dendrite localisation ratios needed to have a large overlap in molecular species. A summary of the datasets we considered can be found in Reply Table 4. We have cross-matched genes from datasets reporting one of the parameters of interest with datasets that reported other parameters of interest for the same gene (e.g., mRNA neurite-to-soma enrichment scores vs half-lives). To maintain consistency, we have used the same dataset for analysis whenever possible and gave preference to dataset combinations with the highest overlap. Shown in grey in Reply Table 4 are the cross-matched datasets we selected for Figure 3 based on the database overlap and total sample size.

New referee comment: *I thank the authors for their explanation. First, this explanation should be included in the manuscript. Second, some of the presented datasets show equal or almost equal overlap with multiple other datasets. In such cases, the authors should conduct an analysis of these multiple overlaps. Drawing conclusions based on multiple datasets will strengthen the reliability of those conclusions. Third, the list of datasets available for analysis in Table 4 is incomplete.*

Our new answer: We have now added this information about dataset selection to the Supplementary Text section ‘Dataset selection for validation of model predictions’. Furthermore we re-computed Figure 3 with different pairs of cross-matched datasets for each panel and show that our main results still hold when alternative data sources are considered (Reply Figure 5 and Figure S17). The Reply Table 3 and Table S7 now contain the detailed information of eligible cross-matched entries for each dataset used for various panels of Figure 3, and Figure S17 (Reply Figure 5). Let us mention that we decided to expand the Table S7 and now list the number of eligible cross-matched entries between any two datasets for the somata-enriched and neurite-enriched categories separately, instead of listing the total gene name overlap between these databases. We needed to evaluate the gene overlap separately for somata-enriched and neurite-enriched while applying author-provided filters (e.g. ignoring gene names with missing values, non-significant values etc.). Previously we have listed the gene overlap without the consideration of additional author-provided filters because depending on the parameter considered very different filters applied, but now we provide both numbers in Tab. S2 and S7. We now provide the eligible matches we used to compute the statistical tests for each parameter dimension.

We have made sure that all the datasets used we analyse are listed in the tables we provide, including Table S7. Tables S2 and S7 (Reply Table 1) show an overview of datasets with their full size (Table S2), and cross-matched samples, and the number of species categorized in somata- or neurite-enriched groups as used in Figures 3, 4 and S17 (Reply Figure 5) depended on the particular databases-to-database match and is now provided in Table S7, in two versions one match (dark gray) is used in Figure 3 and one alternative match (light gray) is shown in Figure S17. In other words, Figure 3 now has five panels for which Table S7 has five dedicated sections (listed under ‘Parameter combination’) and corresponding rows in Table S2 describe the corresponding databases, to which we have added additional data entries as suggested by referee (e.g. Price et al., 2010, data set). For completeness, the datasets used in Figure 4 are also listed in the last two rows of Table S2.

Previous referee comment 6. *The authors discuss previous studies that show transcripts with shorter 3' UTRs tend to remain in the soma, while those with longer 3' UTRs are preferentially localized in dendrites, in relation to Figure 3B. However, Figure 3B shows mRNA length. What are the findings when 3' UTR length is considered?*

Our previous reply: We thank the referee for mentioning this point. In case the referee is interested in the model predictions when considering 3'-UTR length instead of mRNA length, we kindly refer to our answer to comment 3.3. Here, we want to comment on the experimental findings concerning the localization of transcripts with different 3'-UTR lengths. To clarify

that our statements on 3'-UTRs are indeed valid, we want to point to the findings of Erin Schuman's lab on the localization of transcripts in neurons. Tushev et al., 2018¹⁰ showed that transcripts localizing in dendrites are generally longer than those in the soma (Reply Figure 17A), and, when comparing transcripts for a given gene (which is equivalent to keeping the coding sequence constant and varying the UTR length), the transcript lengths in the dendrite are longer than those in the soma (see Reply Figure 17B). In addition, similar results have been independently obtained by 100 for BDNF. To highlight this in our manuscript, we added the following lines to the 'Discussion' section: "Interestingly, this model prediction can be observed not only at the population level (Figures 2, 3) but also in mouse hippocampal neurons within the mRNAs coding for the same protein. An et al., 2008¹⁰⁰ reported that the mRNAs of the prominent synaptic protein BDNF preferentially localise to the soma if they have a short 3'-UTR sequence. In contrast, BDNF mRNAs with long 3'UTR sequences preferentially localise in dendrites, confirming our model predictions at the level of individual protein species. This finding has been further corroborated across genes¹⁰."

New referee comment: *The authors are working with multiple RNA localization datasets, and for each of these, the length of 3' UTRs is an easily calculable parameter. The analysis should not be restricted to just one study.*

Our new answer Thank you for this interesting suggestion. We re-analyzed Figure 3 with different pairs of cross-matched datasets for each panel presented, and added Reply Figure 5 as Figure S17. For each panel in Figure S17, we show that the results of Figure 3 remain valid, even when different pairs of datasets and cross-matched proteins and mRNAs are considered. Following referee advice we now provide for the mRNA length three alternatives and re-compute our main result with two of them (see Figure S17). Furthermore, we added the corresponding characterisation of these datasets to Table S7 (shown here as Reply Table 1); highlighted in dark grey are dataset combinations used in Figure 3, and highlighted in light grey are dataset combinations used in Figure S17 (Reply Figure 5) which is a cross-check for Figure 3.

Reply Figure 5: (Added as Figure S17) Results presented in Figure 3 remain valid when considering different database pairs. We confirmed the results presented in Figure 3 with different cross-matched datasets, these are dataset pairs indicated in light grey in Table S7 (dark grey dataset pairs are shown in Figure 3). Analogously to Figure 3, we employ the following colour code: mRNA (red) and protein (blue) parameters and their corresponding mRNA enrichment in neurites vs. somata were extracted from published datasets. In each panel, the x-axis label denotes the corresponding parameter. According to mRNA enrichment, we labelled database entries as neurite-enriched (filled boxes) or somata-enriched (empty boxes). **A** 3'-UTR isoform half-lives from¹⁷ and corresponding enrichment scores from¹⁸; **B** Transcript length in nucleotide by¹⁹ and mRNA enrichment of¹⁸; **C** Protein counts per spine from our data²⁰ and mRNA enrichment from¹⁸; **D** Protein half-lives from²¹ matched with mRNA enrichment scores of¹⁹; **E** Protein length in amino acids and mRNA enrichment scores from¹⁹ matched with mRNA enrichment scores by¹⁸. Boxplots indicate median, quartiles and 1.5x interquartile ranges. *($p < 0.05$), **($p < 0.01$), ***($p < 0.001$), two-sided Wilcoxon ranksum test.

Parameter combination	Main data	Matched data	Eligible matches	
			Somata-enriched	Neurite-enriched
mRNA half-life vs Enrichment scores	Tushev et al. 2018 ¹⁷	Tushev et al. 2018 ¹⁷	341	201
		Glock et al. 2021 ¹⁸	319	94
		Zappulo et al. 2017 ¹⁹	210	22
mRNA length vs Enrichment scores	Zappulo et al. 2017 ¹⁹	Zappulo et al. 2017 ¹⁹	5623	1406
		Glock et al. 2021 ¹⁸	2597	704
		Tushev et al. 2018 ¹⁷	314	191
Protein copy number vs Enrichment scores	Helm et al. 2021 ²⁰	Zappulo et al. 2017 ¹⁹	58	25
		Glock et al. 2021 ¹⁸	43	15
		Tushev et al. 2018 ¹⁷	8	3
Protein half-life vs Enrichment scores	Fornasiero et al. 2018 ²²	Zappulo et al. 2017 ¹⁹	418	151
		Tushev et al. 2018 ¹⁷	313	200
		Glock et al. 2021 ¹⁸	221	27
	Price et al. 2010 ²¹	Zappulo et al. 2017 ¹⁹	194	61
		Glock et al. 2021 ¹⁸	95	14
		Tushev et al. 2018 ¹⁷	31	17
Protein length vs Enrichment scores	Zappulo et al. 2017 ¹⁹	Zappulo et al. 2017 ¹⁹	5623	1406
		Glock et al. 2021 ¹⁸	726	261
		Tushev et al. 2018 ¹⁷	314	191

Reply Table 1: (Added as Table S7) Database cross-match overview. Updated overview of the databases used to verify our model predictions. For each parameter combination, we have cross-matched two databases, one with a reported parameter of interest and the other with mRNA localization score. The matched databases used in Figure 3 are indicated in dark grey, we have selected these based due to the higher number of eligible matches. Similarly in light grey are the alternative cross-matches used for a cross-check in Reply Figure 5.

References

- ¹ Glock, C., Heumüller, M. & Schuman, E. M. mRNA transport & local translation in neurons. *Current Opinion in Neurobiology* **45**, 169–177 (2017).
- ² Kiebler, M. A. & Bassell, G. J. Neuronal RNA Granules: Movers and Makers. *Neuron* **51**, 685–690 (2006).
- ³ Zeitelhofer, M., Macchi, P. & Dahm, R. Perplexing bodies: The putative roles of P-bodies in neurons. *RNA Biology* **5**, 244–248 (2008).
- ⁴ Batish, M., van den Bogaard, P., Kramer, F. R. & Tyagi, S. Neuronal mRNAs travel singly into dendrites. *Proceedings of the National Academy of Sciences* **109**, 4645–4650 (2012).
- ⁵ Elvira, G. *et al.* Characterization of an RNA Granule from Developing Brain. *Molecular & Cellular Proteomics* **5**, 635–651 (2006).
- ⁶ Farris, S., Lewandowski, G., Cox, C. D. & Steward, O. Selective Localization of Arc mRNA in Dendrites Involves Activity- and Translation-Dependent mRNA Degradation. *Journal of Neuroscience* **34**, 4481–4493 (2014).
- ⁷ Mikl, M., Vendra, G. & Kiebler, M. A. Independent localization of MAP2, CaMKII α and β -actin RNAs in low copy numbers. *EMBO reports* **12**, 1077–1084 (2011).
- ⁸ Amrute-Nayak, M. & Bullock, S. L. Single-molecule assays reveal that RNA localization signals regulate dyneindynactin copy number on individual transcript cargoes. *Nature Cell Biology* **14**, 416–423 (2012).
- ⁹ Donlin-Asp, P. G., Polisseni, C., Klimek, R., Heckel, A. & Schuman, E. M. Differential regulation of local mRNA dynamics and translation following long-term potentiation and depression. *Proceedings of the National Academy of Sciences* **118**, e2017578118 (2021).
- ¹⁰ Fusco, D. *et al.* Single mRNA Molecules Demonstrate Probabilistic Movement in Living Mammalian Cells. *Current Biology* **13**, 161–167 (2003).
- ¹¹ Park, H. Y. *et al.* Visualization of Dynamics of Single Endogenous mRNA Labeled in Live Mouse. *Science* **343**, 422–424 (2014).
- ¹² Bertrand, E. *et al.* Localization of ASH1 mRNA Particles in Living Yeast. *Molecular Cell* **2**, 437–445 (1998).
- ¹³ Cha, B.-J., Koppetsch, B. S. & Theurkauf, W. E. In Vivo Analysis of Drosophila bicoid mRNA Localization Reveals a Novel Microtubule-Dependent Axis Specification Pathway. *Cell* **106**, 35–46 (2001).
- ¹⁴ Kanai, Y., Dohmae, N. & Hirokawa, N. Kinesin transports rna: Isolation and characterization of an rna-transporting granule. *Neuron* **43**, 513–525 (2004).
- ¹⁵ Bauer, K. E., de Queiroz, B. R., Kiebler, M. A. & Besse, F. Rna granules in neuronal plasticity and disease. *Trends in Neurosciences* **46**, 525–538 (2023).
- ¹⁶ Ripin, N. & Parker, R. Formation, function, and pathology of rnp granules. *Cell* **186**, 4737–4756 (2023).
- ¹⁷ Tushev, G. *et al.* Alternative 3' UTRs Modify the Localization, Regulatory Potential, Stability, and Plasticity of mRNAs in Neuronal Compartments. *Neuron* **98**, 495–511 (2018).
- ¹⁸ Glock, C. *et al.* The translome of neuronal cell bodies, dendrites, and axons. *Proceedings of the National Academy of Sciences* **118**, e2113929118 (2021).
- ¹⁹ Zappulo, A. *et al.* RNA localization is a key determinant of neurite-enriched proteome. *Nature Communications* **8**, 583 (2017).
- ²⁰ Helm, M. S. *et al.* A large-scale nanoscopy and biochemistry analysis of postsynaptic dendritic spines. *Nature Neuroscience* **24**, 1151–1162 (2021).
- ²¹ Price, J. C., Guan, S., Burlingame, A., Prusiner, S. B. & Ghaemmaghami, S. Analysis of proteome dynamics in the mouse brain. *Proceedings of the National Academy of Sciences* **107**, 14508–14513 (2010).
- ²² Fornasiero, E. F. *et al.* Precisely measured protein lifetimes in the mouse brain reveal differences across tissues and subcellular fractions. *Nature Communications* **9**, 4230 (2018).

Dear reviewers,

thank you very much for your support with our manuscript entitled “How energy determines spatial localisation and copy number of molecules in neurons”.

We are pleased to learn that reviewer 1 and reviewer 2 support the publication of our manuscript. Below we answer the remaining two comments of reviewer 3. The reviewer comments are shown in *italic* and our corresponding answer below.

Thank you for your support and the many helpful comments.

Reviewer #3

The dataset usage remains limited, and the authors combine data from different models in a suboptimal way. Despite the availability of over a dozen recent neuronal RNA localization datasets with high coverage, the manuscript relies on only three (Table S7: Zappulo 2017, Glock 2021, and Tushev 2018). Additionally, data from different models are combined. For example, the authors use mRNA localization scores from mESC-derived neurons but pair them with mRNA half-lives from rat hippocampal neurons. This limits the overall conclusions of the paper. The authors should address the limited dataset usage and its impact on their model in the manuscript.

Answer: Thank you for the suggestion to show that our model predictions hold for more datasets. We have now added Figure S19 (Reply Figure 1 below) which shows that at least two database pairs in rats and mice support each of our five model predictions from Figure 2. In each of these pairs, we combined mRNA localisation data and the values for the parameters of interest (e.g., half-life or length of mRNAs or proteins) from neurons of the same species. For completeness, we also introduced Reply Table 1 (added as Table S8) detailing the corresponding citations, animal models, and cell types from which the data were obtained. Both are shown below for convenience. Let us stress that we tested all five model predictions on mRNA localisation scores and the parameters of interest that were measured in the same species, either mouse or rat. We show that our predictions hold across various dataset pairs for either mice or for rats in A, B, D, and E. Details are listed in the Reply Table 1 (Table S8). In panel C we show three database pairings exclusively from rat neurons because synaptic protein count profiles across many different protein species are currently only available for rat tissue (Helm et al., 2021¹).

Finally, let us highlight that more studies than we have space to present here have reported experimental results in line with our predictions and we cite these throughout our Results section. Below we show for convenience two Reply Figures 2 and 3. For example, Hale et al., 2021² have shown that mRNA coding sequence lengths and 3'-UTR lengths are significantly longer in dendritic compared to somatic compartments, which is a result that directly aligns with our prediction in panels B and E because amino acid chain length is equivalent to coding sequence length divided by three.

Let us also mention that the Chekulaeva lab has shown very recently in Loedige et al., 2023³, that neurite-enriched mRNAs have longer half-lives and that not only the binary classification into soma/dendrite categories reveals this trend but also the soma/dendrite localisation ratio itself is directly proportional to the transcript half-life (top panel reply Figure 2). Also, at least 6 studies from rats and mice which we discuss in the reply to the referee's second question have reported that dendritic mRNA transcripts are longer than those found in the soma. We show below all these additional results from Hale et al., 2021², and Loedige et al., 2023³, and others in the Reply Figures 3 and 2.

We highlighted the new data in the Results section and added the following sentence “For completeness, let us mention that we confirmed all five experimental confirmations shown in Figure 3 using at least three different database pairings per result considering only data from the same species in each database-pair (Figure S19).”

Taken together, the currently available experimental data across different studies in mice and rats separately strongly indicate that our optimal energy theory can unite many different experimental findings under a single energy-optimal framework and thus makes it a valuable reference frame to understand current and future experimental findings. In the discussion we have modified one of the last sentences to read as follows “[...]future research addressing to what degree energy optimisation strategies play a role at the molecular level across the evolutionary, developmental, and cell type-specific dimensions will help us gain a deeper understanding of the functional architecture of a cell at the subcellular level and understand the role of the neuronal metabolism in disease⁴⁻⁷.”

Reply Figure 1: Extended analysis for Figure 3 focusing on multiple database pairs from the same species

Results presented in Figure 3 remain valid across numerous dataset pairings considering pairing with the same species, rat or mouse. By using the same species from an individual dataset or matching it with the dataset of the same species from another study (Reply Table 1 and Table S8), we show that the results presented in Figure 3 and Figure S17 hold across different dataset pairings. Analogously to Figure 3, we employ the following color code: mRNA (red) and protein (blue) parameters, and their corresponding mRNA enrichment in neurites vs. somata was extracted from published datasets. In each panel, the Y-axis label denotes the corresponding parameter. According to mRNA enrichment, we labelled database entries as neurite-enriched (filled boxes) or somata-enriched (empty boxes). **A1-A4** 3'-UTR isoform half-lives and corresponding enrichment scores^{3,8,9}; **B1-B4** 3'-UTR length in nucleotide and mRNA enrichments^{3,8,10,11}; **C1-C3** Protein counts per spine from our data¹ and mRNA enrichments from^{8,9,12}; **D1-D4** Protein half-lives matched with corresponding mRNA enrichment scores^{9-11,13-15}; **E1-E4** Protein length in amino acids matched with corresponding mRNA enrichment scores⁸⁻¹¹ and Ensemble BioMart tool¹⁶. Boxplots indicate median, quartiles and 1.5x interquartile ranges. *($p < 0.05$), **($p < 0.01$), ***($p < 0.001$), two-sided Wilcoxon ranksum test. A p-value of x % means that the hypothesis that the two datasets have a statistically indistinguishable mean value holds with (1-x)% certainty.

Related to model prediction in panel A in reply Figure 1 that dendritic mRNAs have a longer half-life

Loedige et al 2023

Related to model prediction in panel E in reply Figure 1 that aminoacid sequence/CDS is longer for dendritic mRNAs

Hale et al 2021

Reply Figure 2: Additional data from cultured primary cortical neurons and mESC-derived neurons³ and mouse hippocampal slices² confirming our predictions in Figure S19A (A1-A4) and Figure S19E (E1-E4) (which is Reply Figure 1).

The question of how 3'-UTR length affects localization is also presented in a limited manner. The figures show mRNA lengths for somata-enriched and neurite-enriched transcripts, but the text attributes the difference in mRNA length to variations in the length of the 3'-UTRs, implying this is a general trend. However, the authors do not provide an analysis of multiple datasets to support this claim, such as distributions of 3'-UTR lengths for somata-enriched and neurite-enriched transcripts within individual localization datasets, which is essential to back up the authors' claim. A reference to the conclusions of the Tushev et al. manuscript is insufficient, as it is a literature reference, not a finding of the current study, and it represents a single study.

Answer: Thank you for the suggestion to not only cite but directly include more datasets supporting our model prediction that longer 3'-UTRs are more abundant in the dendrites than the soma. We now include in our manuscript 4 different datasets (all obtained using database pairings from the same species (either from mice or from rats)) and all four support our model conclusion that longer 3'UTRs are more abundant in the dendrites than the soma, see Reply Figure 3 (left) below. In addition to our own data analysis, we show below two other published results that also confirm our predictions, these are from An et al., 2008¹⁷, and Hale et al., 2021². This brings the number of datasets supporting this particular model prediction to six and each of these six represents independently conducted studies (Figure 3, right). One result (An et al., 2008¹⁷) has been obtained within the population of mRNAs coding for the same gene (BDNF). We discuss the An et al., 2008¹⁷, dataset in the results section and cite the corresponding study. For the sake of readability, we opted to show one of these results in the main manuscript (Figure 3B) and the remaining in the new Figure S19 (Reply Figure 1). The results from An et al., 2008¹⁷, Hale et al., 2021², studies are now cited in the results section.

Our own reanalysis of available experimental data from four studies, three in mice (B1,B2, B4) and one in rats (B3), now part of suppl. Fig. 19

An et al., 2008

Hale et al., 2021

Reply Figure 3: Additional data supporting our model prediction that 3'UTR sequences are longer in dendritic mRNAs. This represents 6 different database pairs in total that support our model prediction in panel B in Figure 2. (Left) For reference we show panels B1 to B4 from Figure S19 (Reply Figure 1) which represent our data analysis of the corresponding citations. **(Right)** Adapted from Figure 1E-F from An et al., 2008¹⁷ (in cultured rat cortical neurons) and from Figure 1C from Hale et al., 2021² (in mouse hippocampal slices).

Panels	Data	Cell type
A1	Tushev et al. 2018 ⁸	Rat hippocampal slices
A2	Tushev et al. 2018 ⁸ Glock et al. 2021 ⁹	Rat hippocampal slices Rat hippocampal slices
A3	Loedige et al. 2023 ³	Mouse primary cortical neurons
A4	Loedige et al. 2023 ³	Mouse mESC-derived neurons
B1	Zappulo et al. 2017 ¹⁰	Mouse mESC-derived neurons
B2	Zappulo et al. 2017 ¹⁰ Farris et al. 2019 ¹¹	Mouse mESC-derived neurons Mouse hippocampal slices
B3	Tushev et al. 2018 ⁸	Rat hippocampal slices
B4	Loedige et al. 2023 ³	Mouse mESC-derived neurons
C1	Helm et al. 2021 ¹ Tushev et al. 2018 ⁸	Rat cultured hippocampal neurons Rat hippocampal slices
C2	Helm et al. 2021 ¹ Glock et al. 2021 ⁹	Rat cultured hippocampal neurons Rat hippocampal slices
C3	Helm et al. 2021 ¹ Cajigas et al. 2012 ¹²	Rat cultured hippocampal neurons Rat hippocampal slices
D1	Fornasiero et al. 2018 ¹³ Zappulo et al. 2017 ¹⁰	Mouse cortex synaptosomes Mouse mESC-derived neurons
D2	Fornasiero et al. 2018 ¹³ Farris et al. 2019 ¹¹	Mouse cortex synaptosomes Mouse hippocampal slices
D3	Price et al. 2010 ¹⁴ Zappulo et al. 2017 ¹⁰	Mouse brain Mouse mESC-derived neurons
D4	Heo et al. 2018 ¹⁵ Glock et al. 2021 ⁹	Rat cultured neurons Rat hippocampal slices
E1	Zappulo et al. 2017 ¹⁰	Mouse mESC-derived neurons
E2	Zappulo et al. 2017 ¹⁰ Farris et al. 2019 ¹¹	Mouse mESC-derived neurons Mouse hippocampal slices
E3	Ensemble BioMart ¹⁶ Tushev et al. 2018 ⁸	Rat genome assembly Rnor_6.0 (release 104) Rat hippocampal slices
E4	Ensemble BioMart ¹⁶ Glock et al. 2021 ⁹	Rat genome assembly Rnor_6.0 (release 104) Rat hippocampal slices

Reply Table 1: References used in Figure S19 (Reply Figure 1) and the cell type of the used datasets. For each subpanel, we only cross-matched the datasets within the same species.

References

- ¹ Helm, M. S. *et al.* A large-scale nanoscopy and biochemistry analysis of postsynaptic dendritic spines. *Nature Neuroscience* **24**, 1151–1162 (2021).
- ² Hale, C. R. *et al.* FMRP regulates mRNAs encoding distinct functions in the cell body and dendrites of CA1 pyramidal neurons. *eLife* **10** (2021).
- ³ Loedige, I. *et al.* mRNA stability and m6A are major determinants of subcellular mRNA localization in neurons. *Molecular Cell* **83**, 2709–2725 (2023).
- ⁴ Li, S. & Sheng, Z.-H. Energy matters: presynaptic metabolism and the maintenance of synaptic transmission. *Nature Reviews Neuroscience* **23**, 4–22 (2021).
- ⁵ Le˘Masson, G., Przedborski, S. & Abbott, L. F. A Computational Model of Motor Neuron Degeneration. *Neuron* **83**, 975–988 (2014).
- ⁶ Pissadaki, E. K. & Bolam, J. P. The energy cost of action potential propagation in dopamine neurons: clues to susceptibility in Parkinsons disease. *Frontiers in Computational Neuroscience* **7** (2013).
- ⁷ Vandoorne, T., Bock, K. D. & Bosch, L. V. D. Energy metabolism in ALS: an underappreciated opportunity? *Acta Neuropathologica* **135**, 489–509 (2018).
- ⁸ Tushev, G. *et al.* Alternative 3' UTRs Modify the Localization, Regulatory Potential, Stability, and Plasticity of mRNAs in Neuronal Compartments. *Neuron* **98**, 495–511 (2018).
- ⁹ Glock, C. *et al.* The translome of neuronal cell bodies, dendrites, and axons. *Proceedings of the National Academy of Sciences* **118**, e2113929118 (2021).
- ¹⁰ Zappulo, A. *et al.* RNA localization is a key determinant of neurite-enriched proteome. *Nature Communications* **8**, 583 (2017).
- ¹¹ Farris, S. *et al.* Hippocampal subregions express distinct dendritic transcriptomes that reveal differences in mitochondrial function in ca2. *Cell reports* **29**, 522–539 (2019).
- ¹² Cajigas, I. J. *et al.* The Local Transcriptome in the Synaptic Neuropil Revealed by Deep Sequencing and High-Resolution Imaging. *Neuron* **74**, 453–466 (2012).
- ¹³ Fornasiero, E. F. *et al.* Precisely measured protein lifetimes in the mouse brain reveal differences across tissues and subcellular fractions. *Nature Communications* **9**, 4230 (2018).
- ¹⁴ Price, J. C., Guan, S., Burlingame, A., Prusiner, S. B. & Ghaemmaghami, S. Analysis of proteome dynamics in the mouse brain. *Proceedings of the National Academy of Sciences* **107**, 14508–14513 (2010).
- ¹⁵ Heo, S. *et al.* Identification of long-lived synaptic proteins by proteomic analysis of synaptosome protein turnover. *Proceedings of the National Academy of Sciences* **115** (2018).
- ¹⁶ Ensemble BioMart. <https://www.ensembl.org/index.html>.
- ¹⁷ An, J. J. *et al.* Distinct Role of Long 3 UTR BDNF mRNA in Spine Morphology and Synaptic Plasticity in Hippocampal Neurons. *Cell* **134**, 175–187 (2008).

Dear reviewer 4,

thank you for taking the time to review our manuscript and our answer to previous referee comments. We are pleased to hear that all comments are addressed and that the manuscript is now ready for publication. Below we thank you for specific remarks about the strength of our study.

Reviewer #4

*I am not really qualified to evaluate the paper as a whole as the computational methods it uses are outside my expertise. However, the results of the model to me seem to generally agree with the prevailing trends from the published datasets that I am aware of. Particularly that neurite-localized RNAs tend to have longer halflives and longer 3' UTRs. The effect of transcript halflife has been well documented in Loedige et al (as referenced in the manuscript). The effect of 3' UTR length has been hinted at in a few papers including <https://link.springer.com/article/10.1186/s12864-021-07781-1>. This analysis has the advantage of comparing 3' UTR lengths between different isoforms *of the same gene* rather than comparing 3' UTR lengths across different genes. This lets to description of effects more precisely to differences in 3' UTRs rather than differences in the rest of the RNA. Here, the effect remains with longer 3' UTRs preferentially being neurite-enriched compared to their sister isoforms with shorter UTRs.*

Answer: Thank you for recognizing that our isoform analysis ”*of the same gene* rather than comparing 3' UTR lengths across different genes” represents unique strength of our study and thank you for appreciating that our result that the model-predicted 3'UTR length effect is present within the same gene and across different genes. Thank you also for mentioning that the findings by <https://link.springer.com/article/10.1186/s12864-021-07781-1> showing that specific RNA-binding protein expression (that may correlate with U'UTR length) is also enriched in distal dendritic mRNAs. This provides even more indirect experimental evidence supporting our model predictions.

As far as the response to reviewer 3 goes, in my view, the authors' response was adequate. I don't find the fact that the authors use datasets from disparate sources (mESC neurons, rat hippocampal neurons, etc.) as a weakness. Rather, I view it as a strength as it might mean that the conclusions are broadly generalizable.

Answer: Thank you for mentioning this, we thank for the appreciation of our work and the broad across species angle it covers.